

# Comparative internal anatomy of Staurozoa (Cnidaria), with functional and evolutionary inferences

Lucília S. Miranda[1], Allen G. Collins[2], Yayoi M. Hirano[3],
Claudia E. Mills[4] and Antonio C. Marques[1,5]

[1] Department of Zoology, Instituto de Biociências, Universidade de São Paulo, São Paulo, Brazil
[2] National Systematics Laboratory, National Marine Fisheries Service (NMFS), National Museum
of Natural History, Smithsonian Institution, Washington, D.C., United States of America
[3] Coastal Branch of Natural History Museum and Institute, Chiba, Katsuura, Chiba, Japan
[4] Friday Harbor Laboratories and the Department of Biology, University of Washington, Friday
Harbor, Washington, United States of America
[5] Centro de Biologia Marinha, Universidade de São Paulo, São Sebastião, São Paulo, Brazil

Corresponding author
Lucília S. Miranda,
mirandals@ib.usp.br

## ABSTRACT

Comparative efforts to understand the body plan evolution of stalked jellyfishes are scarce. Most characters, and particularly internal anatomy, have neither been explored for the class Staurozoa, nor broadly applied in its taxonomy and classification. Recently, a molecular phylogenetic hypothesis was derived for Staurozoa, allowing for the first broad histological comparative study of staurozoan taxa. This study uses comparative histology to describe the body plans of nine staurozoan species, inferring functional and evolutionary aspects of internal morphology based on the current phylogeny of Staurozoa. We document rarely-studied structures, such as ostia between radial pockets, intertentacular lobules, gametoducts, pad-like adhesive structures, and white spots of nematocysts (the last four newly proposed putative synapomorphies for Staurozoa). Two different regions of nematogenesis are documented. This work falsifies the view that the peduncle region of stauromedusae only retains polypoid characters; metamorphosis from stauropolyp to stauromedusa occurs both at the apical region (calyx) and basal region (peduncle). Intertentacular lobules, observed previously in only a small number of species, are shown to be widespread. Similarly, gametoducts were documented in all analyzed genera, both in males and females, thereby elucidating gamete release. Finally, ostia connecting adjacent gastric radial pockets appear to be universal for Staurozoa. Detailed histological studies of medusozoan polyps and medusae are necessary to further understand the relationships between staurozoan features and those of other medusozoan cnidarians.

## INTRODUCTION

The class Staurozoa of the phylum Cnidaria (*Marques & Collins, 2004*; *Collins et al., 2006*) includes representatives with a peculiar life cycle: creeping larvae settle and develop into

juvenile stauropolyps that later metamorphose into non-free-swimming, adult stauromedusae while still being attached to a substrate by a peduncle (*Wietrzykowski, 1912*; *Kikinger & Salvini-Plawen, 1995*; *Miranda, Collins & Marques, 2010*). In general, the apical half of the metamorphosed stauromedusa (calyx) has characters similar to those of adult scyphomedusae and cubomedusae, such as hollow structures of tentacular origin (rhopalioids/rhopalia), circular coronal muscle, gastric filaments, and gonads (*Collins, 2002*; *Collins et al., 2006*). The basal region (peduncle), on the other hand, retains polypoid characters such as gastric septa associated with four interradial longitudinal muscles (*Collins, 2002*; *Stangl, Salvini-Plawen & Holstein, 2002*). Consequently, understanding the body plan of a stauromedusa is more complex than for other medusozoans because of its dual nature (*Collins et al., 2006*; *Miranda, Collins & Marques, 2013*).

Internal anatomy is an important source of characters used in staurozoan taxonomy (*Miranda, Collins & Marques, 2013*), mainly because stauromedusae have relatively few macromorphological characters useful to differentiate species (*Hirano, 1997*). There are several detailed histological studies (*Clark, 1878*; *Gross, 1900*; *Wietrzykowski, 1912*; *Uchida, 1929*; *Uchida & Hanaoka, 1933*; *Uchida & Hanaoka, 1934*; *Ling, 1939*; *Miranda, Collins & Marques, 2013*), but comparative efforts to understand the evolution of the body plan of staurozoans are scarce and based only on a small number of species (*Berrill, 1963*; *Thiel, 1966*). Comprehensive histological studies are important to establish detailed similarities and differences in character states within Staurozoa and other clades of Cnidaria, providing a basis to infer character evolution in these clades (*Miranda, Collins & Marques, 2013*).

Recently, histological characters used in the taxonomy of Staurozoa were reviewed based on the study of the internal anatomy of *Haliclystus antarcticus* (*Miranda, Collins & Marques, 2013*). Among other features, poorly known structures such as intertentacular lobules, ostia between adjacent gastric radial pockets, and male and female gonadal vesicles were described, and two possible regions of cnida formation were hypothesized (*Miranda, Collins & Marques, 2013*). However, most of these characters have neither been explored for the class, nor broadly applied to its taxonomy and classification.

A microanatomical comparison benefits greatly from the historical context provided by the molecular phylogenetic analysis of Staurozoa (Fig. 1), which has led to an extensive reassessment of the traditional classification of the group (*Miranda et al., 2016*). This analysis corroborated the non-monophyly of the suborders Cleistocarpida and Eleutherocarpida, formerly recognized based on the presence and absence of claustrum, respectively (*Clark, 1863*). Apparently, the claustrum, an internal tissue that divides the gastrovascular cavity (*Clark, 1863*; *Gross, 1900*), is either homoplastic in different groups or was lost several times (*Collins & Daly, 2005*; *Miranda et al., 2016*). In contrast, the interradial longitudinal muscles in the peduncle have a strong phylogenetic signal, supporting the proposal of the new suborders Myostaurida and Amyostaurida, with and without such muscles in peduncle, respectively (*Miranda et al., 2016*).

Many traits employed in the taxonomy of Staurozoa come from incomplete and/or misinterpreted histological studies, leading to their inaccuracy and inefficiency as taxonomic characters (*Miranda, Collins & Marques, 2013*). Hence, a broad histological

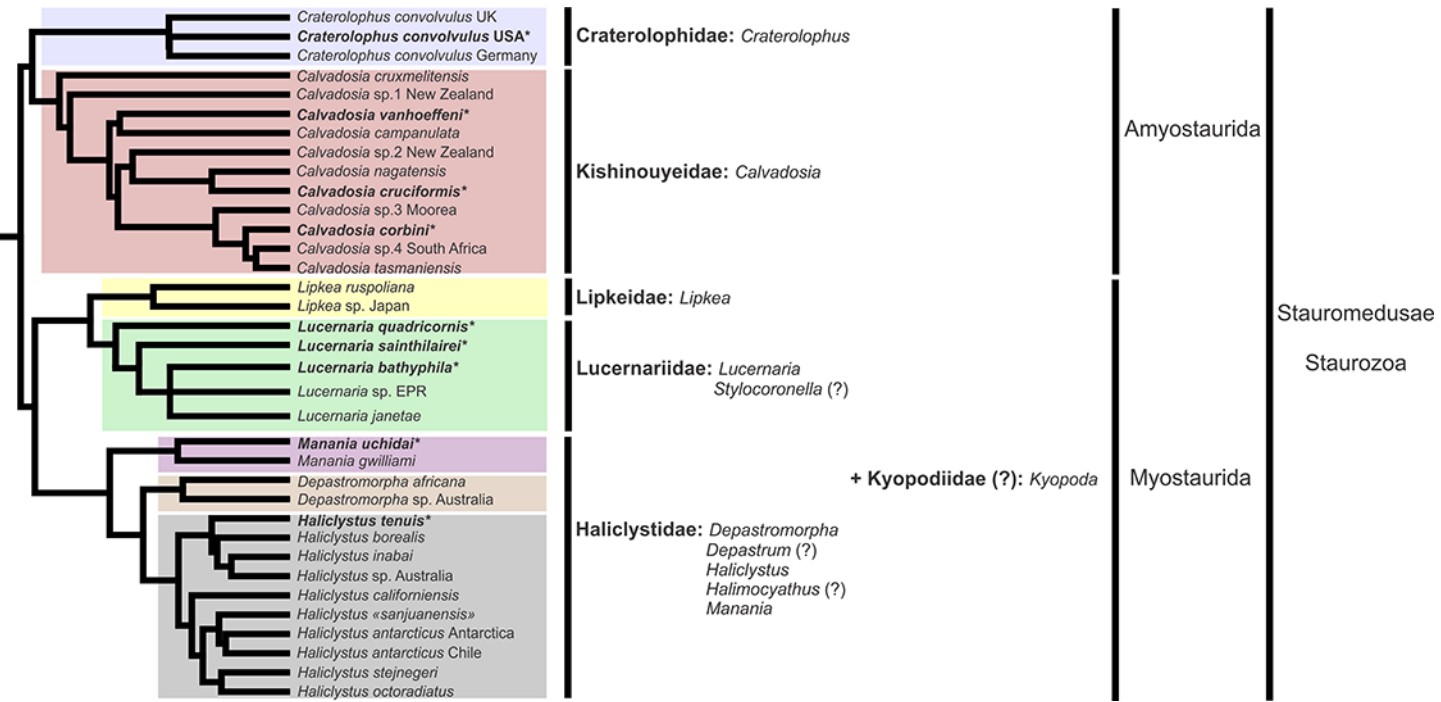

**Figure 1** Molecular phylogenetic hypothesis (based on markers COI, 16S, ITS, 18S, and 28S) of relationships among species of Staurozoa, with its derived classification (*Miranda et al., 2016*). *Species included in this histological study. "?" Groups not included in the molecular analysis, classified according to morphological evidence. EPR, East Pacific Rise; UK, United Kingdom; USA, United States of America.

comparative study of staurozoan taxa is necessary both to allow inferences about the evolution of their body plan as well to add more morphological perspective on the new staurozoan classification (*Miranda et al., 2016*). Therefore, the aim of this study is to use comparative histology to describe the body plan of a broad range of staurozoan species, inferring functional and evolutionary aspects of internal morphology, and reviewing their taxonomic use in the context of the new understanding of the phylogeny of Staurozoa (*Miranda et al., 2016*).

## MATERIAL AND METHODS

We studied nine species of Staurozoa (stauromedusa stage) representing five genera and four families (Table 1), previously fixed directly in 4% formaldehyde solution with seawater, either sampled by us or from museum collections (Table 1). The histological procedures were carried out according to the methods developed for Staurozoa (*Miranda, Collins & Marques, 2013*; modified from *Humason, 1962*; *Mahoney, 1966*). Specimens were cleaned in distilled water; dehydrated in a graded ethanol series (70–100%); cleared in xylene (three steps); infiltrated and embedded in paraffin; serially sectioned transversely (7.0–10.0 μm thick) with a microtome Leica RM2025; cleared in xylene (twice); rehydrated in a graded ethanol series (100–70%); cleaned in distilled water; and stained, using acid fuchsin (15′) (Mallory; *Humason, 1962*: 147) and acetic aniline blue (3′) (Mallory; modified from *Humason, 1962*: 231), intercalated with distilled water to improve the contrast between structures. Prepared slides were observed and

**Table 1 Species of Staurozoa analyzed in this study, with respective localities, voucher catalog numbers, and slides catalog numbers.**

| Family | Species | Locality | Voucher catalog number | Slides catalog number |
|---|---|---|---|---|
| Haliclystidae | *Haliclystus tenuis* | Muroran, Hokkaido, Japan | USNM 1106652 | LEM 09 |
| | *Manania uchidai* | Muroran, Hokkaido, Japan | USNM 1106645 | LEM 10 |
| Lucernariidae | *Lucernaria quadricornis* | Chupa Inlet, Kandalaksha Bay, Russia | USNM 1106240 | LEM 11 |
| | *Lucernaria bathyphila* | Nicolskaya Inlet, Kandalaksha Bay, Russia | USNM 1106643 | LEM 12 |
| | *Lucernaria sainthilairei* | Cross Islands, close to the Biological Station of Moscow State University, Russia | USNM 1102446 | LEM 13 |
| Kishinouyeidae | *Calvadosia corbini* | Aracruz, Espírito Santo, Brazil | MZUSP 1563 | LEM 14 |
| | *Calvadosia cruciformis* | Muroran, Hokkaido, Japan | USNM 1106656 | LEM 15 |
| | *Calvadosia vanhoeffeni* | Janus Island, Palmer Archipelago, Antarctica | USNM 79939 | LEM 16 |
| Craterolophidae | *Craterolophus convolvulus* | Woods Hole, Massachusetts, USA | USNM 54321 | LEM 17 |

**Notes:**
LEM, Laboratory of Marine Evolution of the Institute of Biosciences, University of São Paulo; MZUSP, Museum of Zoology of the University of São Paulo, Brazil; USNM, National Museum of Natural History, Smithsonian Institution, USA.

photographed under a Zeiss microscope AXIO Imager M2. The slides are deposited in the collection of the Laboratory of Marine Evolution of the Institute of Bioscience, University of São Paulo (Table 1; LEM 09-17) and are available for loan. The abbreviations of the morphological structures indicated in Figs. 2–58 are listed in Table 2.

Based on the species examined and on literature information (*Gosse, 1860*; *Clark, 1863*; *Mayer, 1910*; *Uchida, 1929*; *Uchida & Hanaoka, 1933*; *Uchida & Hanaoka, 1934*; *Carlgren, 1935*; *Ling, 1937*; *Kramp, 1961*; *Larson, 1980*; *Larson, 1988*; *Hirano, 1986*; *Hirano, 1997*; *Larson & Fautin, 1989*; *Kikinger & Salvini-Plawen, 1995*; *Marques & Collins, 2004*; *Collins & Daly, 2005*; *Collins et al., 2006*; *Van Iten et al., 2006*; *Pisani et al., 2007*; *Miranda et al., 2016*), we present a matrix of characters for staurozoan genera (Table 3). Some characters have not been investigated in detail for some taxa, especially *Depastrum*, *Halimocyathus*, *Kyopoda*, *Lipkea*, and *Stylocoronella*. Morphological characters were optimized at the generic level by using ACCTRAN (accelerated transformation) in TNT 1.1 (*Goloboff, Farris & Nixon, 2008*), based on the recent staurozoan phylogeny (Figs. 1 and 59) (*Miranda et al., 2016*).

## RESULTS

### General body anatomy

The internal anatomy of nine species of stalked jellyfishes is described below (Figs. 2–58; Table 2). For each species we included detailed information on general body plan, peduncle and septa, gonads and gametoducts, intertentacular lobules, and claustrum (when applicable). The muscular system, manubrium and gastric radial pockets, gastric filaments, perradial and interradial anchors/primary tentacles, ostia, arms delimitation, white spots of nematocysts, batteries of nematocysts, internal subumbrellar layer of nematocysts, secondary tentacles, and pad-like adhesive structures were comparatively analyzed. Therefore, the figures used to illustrate these structures (Figs. 5–7, 9–13, 15–17, 24 and 32) will be mentioned in the descriptions independently of the species.

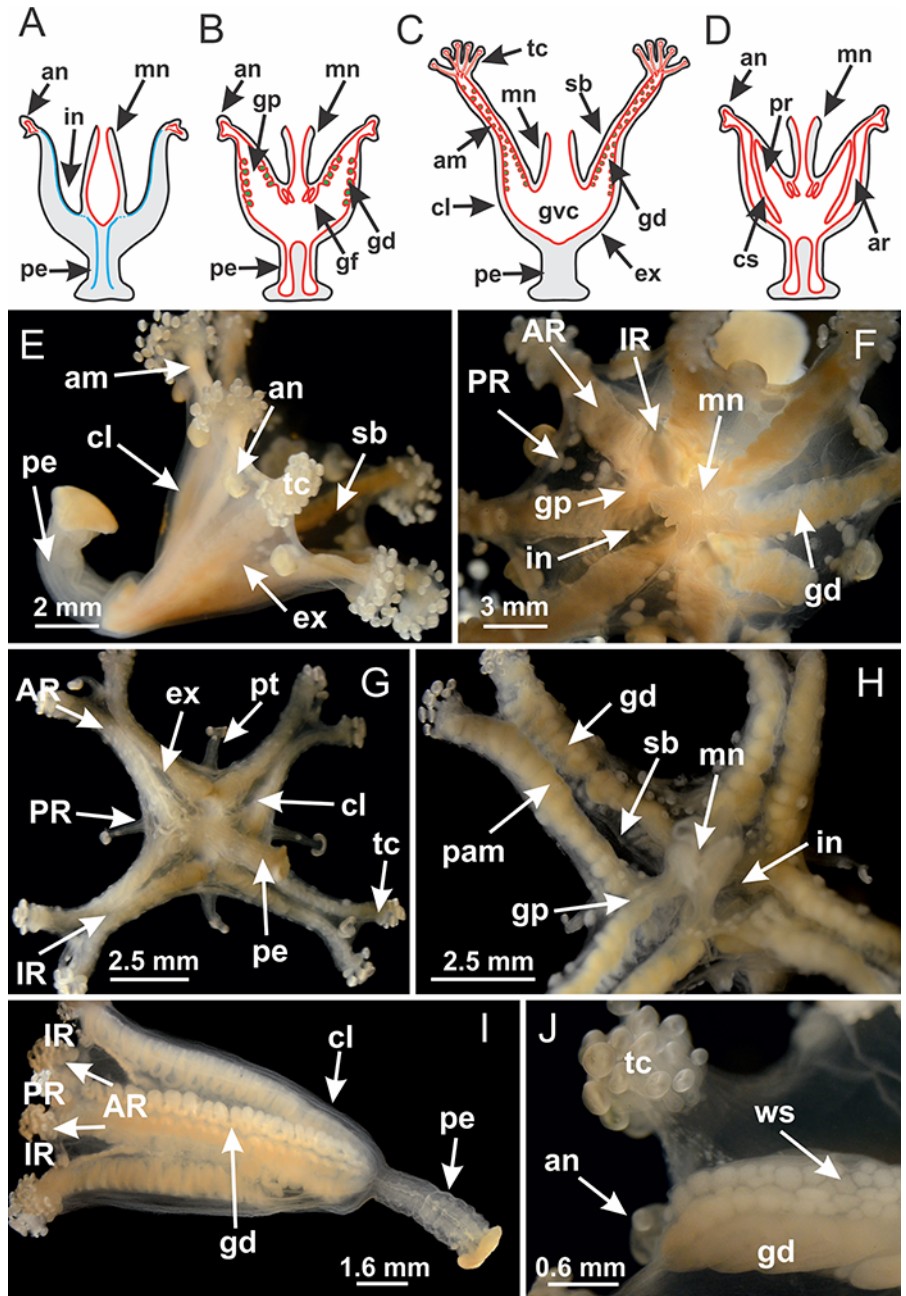

**Figure 2 General organization of the body plan of stalked jellyfishes.** (A) Scheme of a longitudinal section at interradii; (B) scheme of a longitudinal section at perradii (species without claustrum); (C) scheme of a longitudinal section at adradii (species without claustrum; the species with claustra do not have adrarial gonads); (D) scheme of a longitudinal section at perradii (species with claustra) (modified from *Miranda, Collins & Marques, 2013*); *Haliclystus tenuis*: (E) lateral view (exumbrellar); (F) oral view (subumbrellar); *Calvadosia cruciformis*: (G) basal view (exumbrellar); (H) oral view (subumbrellar); *Manania uchidai*: (I) lateral view (exumbrellar); (J) margin of subumbrella. Legend (A–D): black, epidermis; blue, longitudinal muscle; gray, mesoglea; green, gonads; red, gastrodermis. See Table 2 for abbreviations.

**Table 2 Abbreviations of structures reported in the figures.**

| Abbreviations | Structures |
| --- | --- |
| ac | Axial canal |
| am | Arm |
| an | Anchor |
| ar | Accessory radial pocket |
| AR | Adradii |
| ax | Auxiliary radial pocket |
| bn | Battery of nematocysts |
| ci | Cilium |
| cl | Calyx |
| cm | Coronal muscle |
| cs | Claustrum |
| ep | Epidermis |
| ex | Exumbrella |
| fc | Follicle cells |
| ga | Gametoduct |
| gd | Gonad |
| gf | Gastric filament |
| gp | Gastric radial pocket |
| gt | Gastrodermis |
| gvc | Gastrovascular cavity |
| il | Intertentacular lobules |
| in | Infundibulum |
| IR | Interradii |
| ito | Immature oocytes |
| iv | Invagination |
| kb | Knob |
| mc | Manubrial corner |
| mn | Manubrium |
| mnm | Mature nematocysts |
| ms | Mesoglea |
| mto | Mature oocytes |
| mu | Interradial longitudinal muscle/longitudinal muscle |
| nm | Nematocyst |
| nmb | Nematoblast |
| oo | Oocytes |
| ot | Ostia |
| pa | Pads |
| pam | Paired arms |
| pc | Perradial chamber |
| pd | Pedal disk |
| pe | Peduncle |

| Abbreviations | Structures |
| --- | --- |
| pr | Principal radial pocket |
| PR | Perradii |
| pt | Primary tentacle |
| sb | Subumbrella |
| sc | Spermatocytes |
| sp | Septum |
| st | Stem |
| sz | Spermatozoa |
| tc | Tentacles |
| usp | "U-shaped" space |
| vs | Vesicle |
| ws | White spots of nematocysts |

## Suborder Myostaurida *Miranda, Hirano, Mills, Falconer, Fenwick, Marques & Collins, 2016*

Family Haliclystidae *Haeckel, 1879*

Genus *Haliclystus Clark, 1863*

*Haliclystus tenuis Kishinouye, 1910* (Figs. 2E, 2F; 3; 4; 5M, 5Q, 5V; 6; 8; 9A–9I; 10A–10F; 11A–11H; 12A–12C; 13C; 14; 15I–15Q; 17N–17Q; 21A; 57; 58A, 58C and 58D)

Basal region formed by pedal disk of peduncle (stalk) with increased surface area due to invaginations (Figs. 3A and 3B). Peduncle with four perradial chambers (delimited by gastrodermis), alternating with four interradial longitudinal muscle bands (epitheliomuscular cells) embedded in mesoglea (Fig. 4); chambers and muscles developed throughout peduncle (Figs. 4A–4C) except at pedal disk. Perradial chambers fusing at junction of peduncle and calyx (Figs. 4D–4G). Gastrodermis envelops interradial longitudinal muscles at basal region of calyx (Fig. 4H), defining four interradial gastric septa: one thin layer of mesoglea surrounded by two layers of gastrodermis (Figs. 4I and 4J). Four infundibula (peristomal pits) funnel-shaped with blind end, delimited by epidermis, deeply developed down to base of calyx, widening apically, with broad apertures on subumbrella (Figs. 2F and 4J). Gastrovascular cavity without claustrum. At base of infundibula, interradial longitudinal muscle becomes compressed and flattened, being V-shaped in cross section apically (as in other species examined, e.g., Figs. 5C, 5D, 5G, 5H, 5K and 5L), progressively dividing into two adradial bands toward the arms (Figs. 11A and 11B). Adjacent septal gastrodermis merge defining four perradial regions and dividing gastrovascular cavity (Figs. 6B–6D). Fusion of septal gastrodermis forms basal region of manubrium and gastric radial pockets (perradial pockets), i.e., central part of gastrodermis of each septum joins forming four-sided manubrial gastrodermis while lateral parts of adjacent septa join forming gastric radial pockets (Fig. 6). Similarly, each infundibular epidermis also progressively merges apically: central part of each infundibular epidermis becomes manubrial epidermis, and epidermis of adjacent

**Table 3** Matrix of internal anatomy characters for the staurozoan genera. Information from *Gosse (1860)*, *Clark (1863)*, *Mayer (1910)*, *Uchida (1929)*, *Uchida & Hanaoka (1933)*, *Uchida & Hanaoka (1934)*, *Carlgren (1935)*, *Ling (1937)*, *Kramp (1961)*, *Larson (1980)*, *Larson (1988)*, *Hirano (1986)*, *Hirano (1997)*, *Larson & Fautin (1989)*, *Kikinger & Salvini-Plawen (1995)*, *Marques & Collins (2004)*, *Collins & Daly (2005)*, *Collins et al. (2006)*, *Van Iten et al. (2006)*, *Pisani et al. (2007)*, *Miranda et al. (2016)*, and this study. Information that was not available (or not confirmed) was coded as a question mark (?) and non-comparable structures were coded in the matrix as N. Polymorphic characters have states indicated. **1: Longitudinal muscles in peduncle.** (0) present; (1) absent. Four interradial, intramesogleal longitudinal muscles associated with infundibula were considered symplesiomorphic for Staurozoa, shared by staurozoan ancestor with some other medusozoans (*Collins et al., 2006*; *Miranda et al., 2016*). **2: Coronal muscle.** (0) entire; (1) divided. Coronal muscle may be divided into eight sections by the adradial arms or entire (*Miranda et al., 2016*). Coronal muscle entire would be a putative symplesiomorphy of Staurozoa (*Miranda et al., 2016*). *Haliclystus inabai* has been described with entire coronal muscle (*Hirano, 1986*). It is not clear in the original description if the coronal muscle of *Halimocyathus platypus* is entire or divided (*Clark, 1863*). *Stylocoronella* has vestigial coronal muscle (*Kikinger & Salvini-Plawen, 1995*). **3: Number of chambers in peduncle.** (0) peduncle one-chambered; (1) peduncle four-chambered; (2) peduncle 4/1 chambered. The peduncle four-chambered is hypothesized as a synapomorphy of Staurozoa, as the four perradial chambers in the peduncle of stauromedusae are not found in any other cnidarian (*Collins & Daly, 2005*; *Miranda et al., 2016*). The species with 4/1-chambered peduncle have four chambers basally and one chamber at the middle of the peduncle (*Miranda et al., 2016*). Some species of *Manania* have also been described with one chamber throughout the peduncle or one chamber basally and 4 chambers in the middle of the peduncle, and this character should be investigated in detail for the genus (see *Uchida, 1929*; *Hirano, 1986*; and *Miranda et al., 2016* for discussion on changes during development). *Kyopoda* has a unique peduncular anatomy, containing the stomach and gonads (see *Larson, 1988*), and more detailed information is necessary to code this state. **4: Claustrum.** (0) absent; (1) present. The claustrum as a potential symplesiomorphy of Staurozoa (*Thiel, 1966*; *Collins & Daly, 2005*) is equivocal (*Miranda et al., 2016*), and detailed histological studies of the outgroup taxa are need to ground homology hypotheses. **5: White spots of nematocysts.** (0) absent; (1) present. White spots of nematocysts is a putative synapomorphy of Staurozoa (see text for further details). Some species of *Haliclystus* (*H. auricula* and *H. antarcticus*) do not have white spots (see *Hirano, 1997*; *Miranda, Morandini & Marques, 2009*). **6: Internal tips of arms (intertentacular lobules X U-shaped space).** (0) absent; (1) internal tips of arms with intertentacular lobules; (2) internal tips of arms with U-shaped-space. Intertentacular lobules in the internal tips of arms is a putative synapomorphy of Staurozoa (see text). Some species of *Haliclystus* (*H. tenuis*, *H. borealis*, and *H. californiensis*) have a U-shaped space in the internal tips of arms (*Hirano, 1997*). Coding is uncertain for *Depastromorpha*, but according to the illustrations of the original description *D. africana* (single species of the genus) seems to have intertentacular lobules (*Carlgren, 1935*). This character is probably not applicable in *Lipkea* because marginal lobes (or lappets) cannot be undoubtedly homologized with arms or anchors (*Miranda et al., 2016*). **7: Gametoduct.** (0) absent; (1) present. Gametoduct is a putative synapomorphy of Staurozoa (although additional information about the presence of this structure in other cnidarians is still necessary; see *Tiemann & Jarms, 2010*). **8: Anchors.** (0) absent; (1) present. Anchors are metamorphosed eight primary tentacles (perradial and interradial) (*Wietrzykowski, 1912*). Polyp primary tentacles are a potential symplesiomorphy of Staurozoa (*Collins et al., 2006*; *Miranda et al., 2016*). Some species of *Calvadosia* retain primary tentacles, with a modified shape (*C. cruciformis*, *C. tsingtaoensis*; *Ling, 1937*; *Ling, 1939*). Anchors have a swollen, adhesive base, but can retain a remnant of primary tentacle (*Larson & Fautin, 1989*; *Kahn et al., 2010*). **9: Ostia.** (0) present; (1) absent. Stauromedusae have an interradial ostium between two adjacent gastric radial pockets (or accessory radial pockets in species with claustra). Comparable structures, such as the septal ostium or connecting canal, have been observed in Scyphozoa (*Korschelt & Heider, 1895*; *Chapman, 1966*) and Cubozoa (*Conant, 1898*). Although further detailed studies are necessary to assess homology across taxa, we consider ostia between adjacent gastric radial pockets a symplesiomorphy of Staurozoa. **10: Pad-like adhesive structures.** (0) absent; (1) presence of pads in the outermost tentacles; (2) presence of pads on the tips of arms, connected to the stem of secondary tentacles; (3) presence of pads on the tips of arms, externally separated from the stem of the secondary tentacles. The presence of pad-like adhesive structures in the outermost secondary tentacles is a putative synapomorphy of Staurozoa (see text). Some species of *Calvadosia* (*C. capensis*, *C. corbini*, *C. hawaiiensis*, and *C. tasmaniensis*) have broad pads on the tips of arms (*Miranda et al., 2016*). *Calvadosia cruxmelitensis* has a particular adhesive pad-like structure on the tip of each arm, in which the secondary tentacles arise directly from this structure (*Corbin, 1978*; *Miranda et al., 2016*). This character is probably not applicable in *Lipkea* because marginal lobes (or lappets) cannot be undoubtedly homologized with arms or anchors, and capitate secondary tentacles cannot be recognized (*Pisani et al., 2007*; *Miranda et al., 2016*).

| Staurozoan genera | 1 | 2 | 3 | 4 | 5 | 6 | 7 | 8 | 9 | 10 |
|---|---|---|---|---|---|---|---|---|---|---|
| *Calvadosia* | 1 | 1 | 2 | 0 | 1 | 1 | 1 | 0 | 0 | 1 |
| *Craterolophus* | 1 | 1 | 1 | 1 | 1 | 1 | 1 | 0 | 0 | 1 |
| *Depastromorpha* | 0 | 0 | 1 | 1 | 1 | 1(?) | ? | 1 | ? | 1 |
| *Depastrum* | 0 | 0 | 1 | 1 | 1 | ? | ? | 0 | ? | 0 |
| *Haliclystus* | 0 | 0/1 | 1 | 0 | 0/1 | 1/2 | 1 | 1 | 0 | 0/1 |
| *Halimocyathus* | 0 | ? | 1 | 1 | 1 | ? | ? | 1 | ? | 1 |
| *Kyopoda* | 0 | 0 | ? | 0 | 1 | ? | ? | 1 | ? | 1 |
| *Lipkea* | 0 | 0 | 0 | 0 | 1 | N(?) | ? | 0 | ? | N(?) |
| *Lucernaria* | 0 | 1 | 0 | 0 | 1 | 1 | 1 | 0 | 0 | 0 |
| *Manania* | 0 | 0 | 1 | 1 | 1 | 1 | 1 | 1 | 0 | 1 |
| *Stylocoronella* | 0 | ? | 0 | 0 | 1 | ? | ? | 0 | ? | 0 |

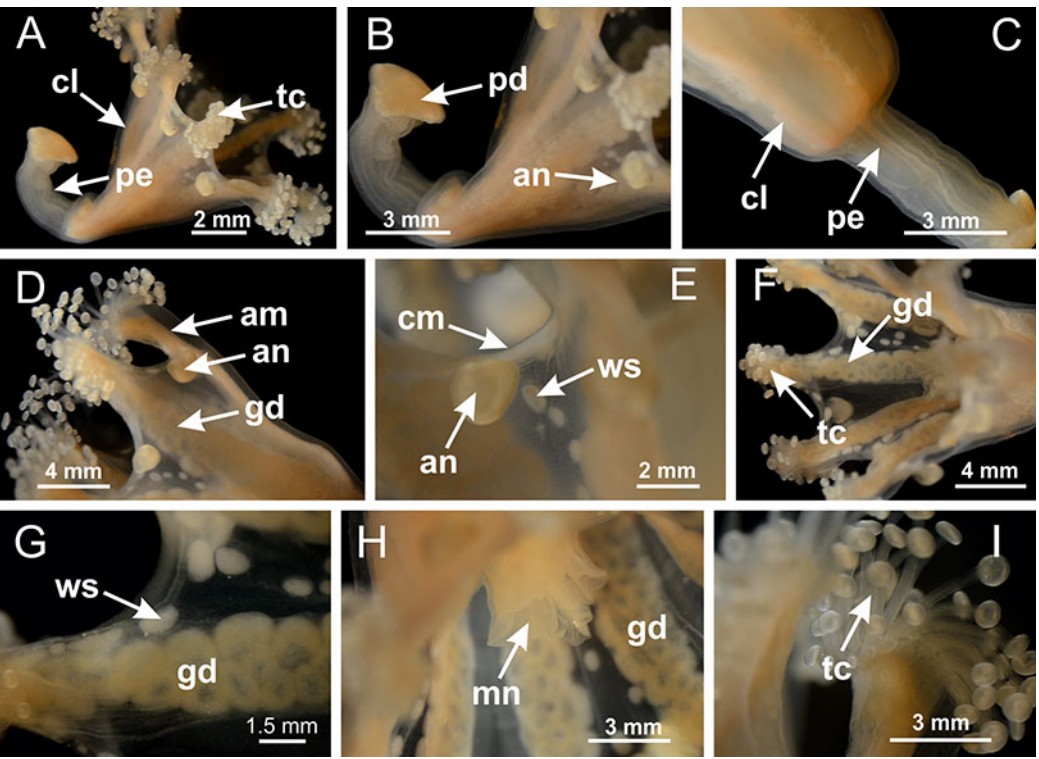

**Figure 3 General view of *Haliclystus tenuis*.** (A) General view of body; (B, C) detail of calyx and peduncle; (D) gonad and arms in calyx; (E) anchor, coronal muscle, and white spots; (F) general view of the subumbrella showing gonads and tentacles; (G) detail of gonads and white spots; (H) detail of manubrium; (I) tentacular cluster. See Table 2 for abbreviations.

infundibula forms epidermis of gastric radial pockets (Figs. 6E–6N). Therefore, each gastric radial pocket is formed by fusion of gastrodermis and epidermis of adjacent septa, and manubrium is formed by fusion of all four septa (Fig. 6). Four gastric radial pockets laterally separated from each other by interradial septa (Figs. 6L and 6N); gastric radial pockets directly connected only by means of small interradial ostia at margin of calyx (Figs. 10A–10F); each gastric radial pocket connected to main gastrovascular cavity. Manubrium (Fig. 3H) internally defined by gastrodermis, externally by epidermis (Figs. 6O and 6P). Gastric filaments composed of one layer of mesoglea surrounded by gastrodermis, formed by lateral evaginations of gastrodermal layer of septa at base of manubrium, concentrated at perradii (as in other species examined; Fig. 7). Gonads (Figs. 3F–3H) with approximately six rows of vesicles (follicles), which are serial gastrodermal evaginations at lateral regions of interradial septa, gastric radial pockets and arms (Fig. 8); vesicles of same gastric radial pocket formed by gastrodermis of two different interradial septa (two adjacent septa) (Fig. 8B). Vesicles composed of an internal layer of gastrodermis, mesoglea, an external (subumbrellar) layer of epidermis, and inner gonadal content (gastrodermal origin) (Figs. 8A, 8D and 8E). Female specimen analyzed (Fig. 8) with ovarian vesicles; gonadal content composed of two main layers: peripheral layer with immature oocytes in different developmental stages (Figs. 8C and 8D), internal layer with mature oocytes with scattered yolk granules (Figs. 8C and 8D).

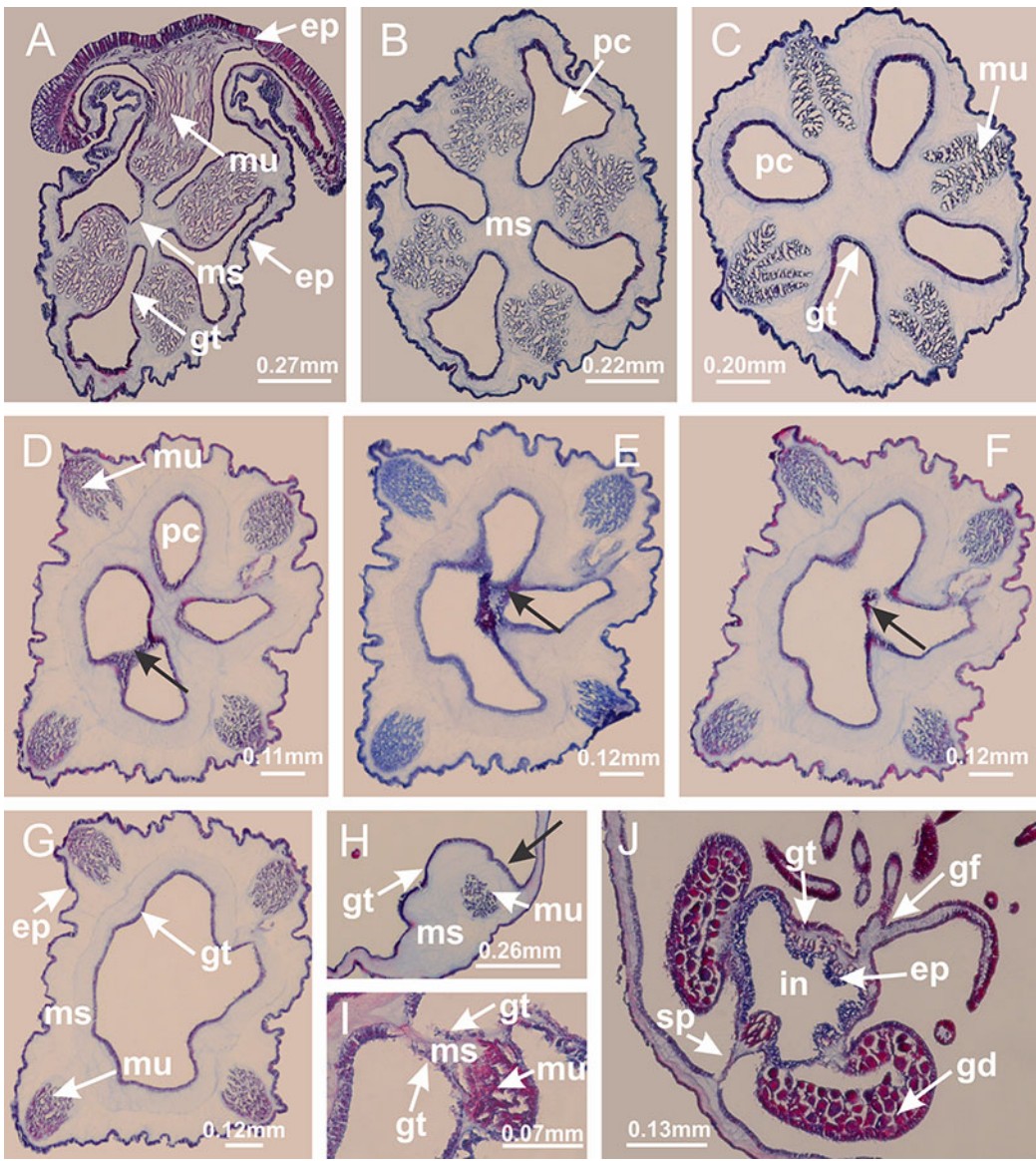

**Figure 4 Peduncle and septa of *Haliclystus tenuis* (from base moving upward in A–J).** (A–C) Organization of four perradial chambers and four interradial longitudinal muscles in peduncle; (D–G) connection of the four perradial chambers in the region between peduncle and calyx, defining one central chamber; (H) gastrodermis envelops interradial longitudinal muscle (indicated by black arrow), defining septum; (I, J) detail of septum, with infundibulum delimited by epidermis, lateral gonads and gastric filaments. (A–J): cross sections. See Table 2 for abbreviations.

Mature oocytes surrounded by cells of gastrodermal origin (probably follicle cells), leading to gastric radial pockets through fusion of these cells with gastrodermis of the ovarian vesicles (Figs. 8D–8M), forming gametoducts (Figs. 8N and 8O). Cilia often associated with gametoducts. Anchors (rhopalioids) (Figs. 3D and 3E) hollow, with hollow stem as evaginations of body surface. Eight large anchors, each located between adjacent arms at calyx margin (Figs. 3D and 3E), four perradial (Figs. 9A–9I) and four interradial (Figs. 10A–10F). Gastrodermis of perradial anchors directly connected to

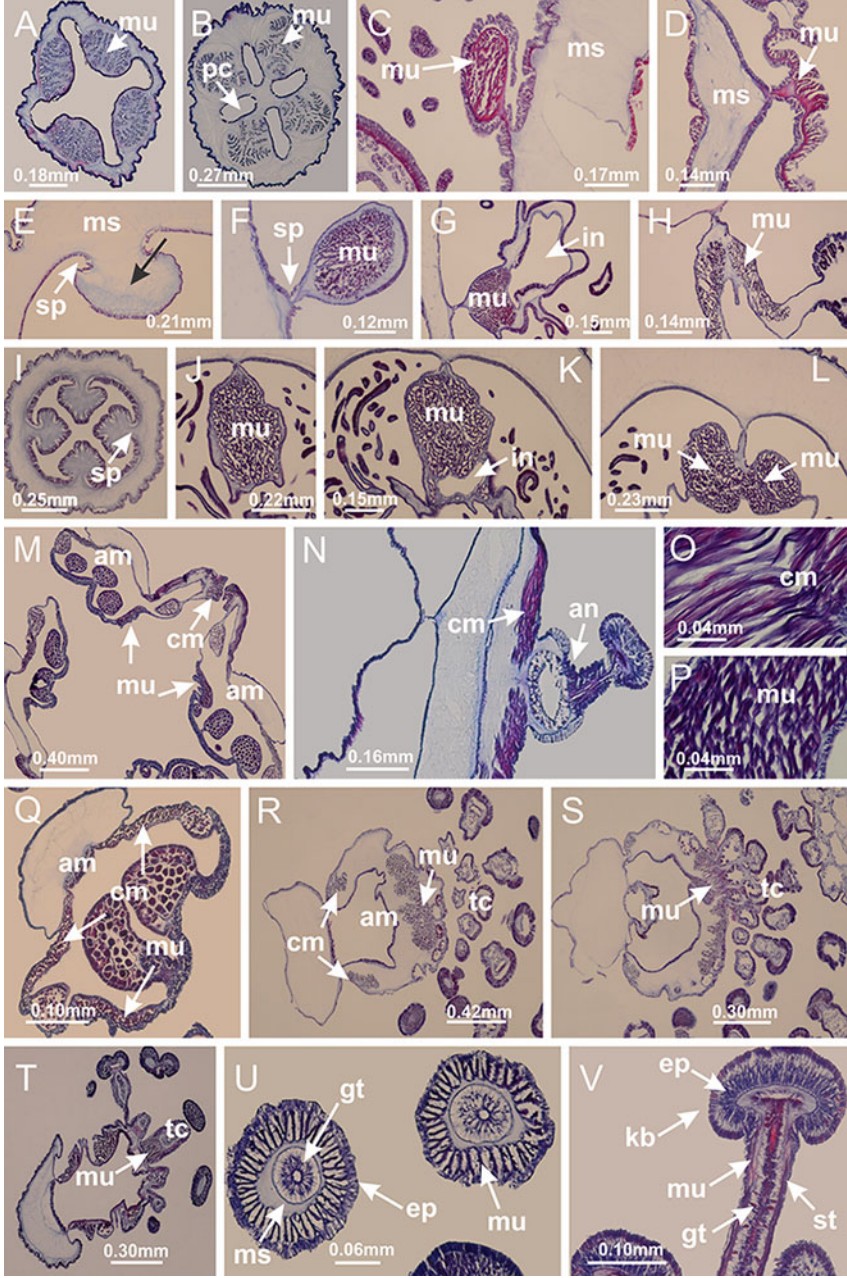

**Figure 5 Muscular system.** *Manania uchidai*: (A, B) interradial longitudinal muscles in peduncle; *Lucernaria sainthilairei*: (C) interradial longitudinal muscle in peduncle; (D) interradial longitudinal muscle in calyx; *Calvadosia vanhoeffeni*: (E) absence of interradial longitudinal muscle in peduncle (indicated by black arrow); (F) interradial longitudinal muscle associated with septum at peduncle/calyx connection; (G) interradial longitudinal muscle associated with septum at base of calyx; (H) interradial longitudinal muscle in calyx, divided into two bands; *Calvadosia corbini*: (I) absence of interradial longitudinal muscle in peduncle; (J) interradial longitudinal muscle associated with septum at peduncle/calyx connection; (K) interradial longitudinal muscle associated with septum at base of calyx; (L) interradial longitudinal muscle in calyx, divided into two bands; *Haliclystus tenuis*: (M) organization of muscular system in the region between calyx and arms (division of arms at perradial region occurs first than at interradial region; one band of longitudinal muscle toward each arm); *M. uchidai*: (N) coronal muscle at the margin of calyx; *C. corbini*: (O) detail of coronal muscle; (P) detail of interradial longitudinal muscle; *H. tenuis*: (Q) muscular organization in arms (one central band of longitudinal muscle, and two lateral bands of coronal muscle); *C. vanhoeffeni*: (R, S) longitudinal muscle toward secondary tentacles; *C. corbini*: (T) longitudinal muscle toward secondary tentacles, (U) longitudinal muscle in the stem of secondary tentacles; *H. tenuis*: (V) longitudinal muscle in the stem of secondary tentacles. (A–U): cross sections; V: longitudinal section. See Table 2 for abbreviations.

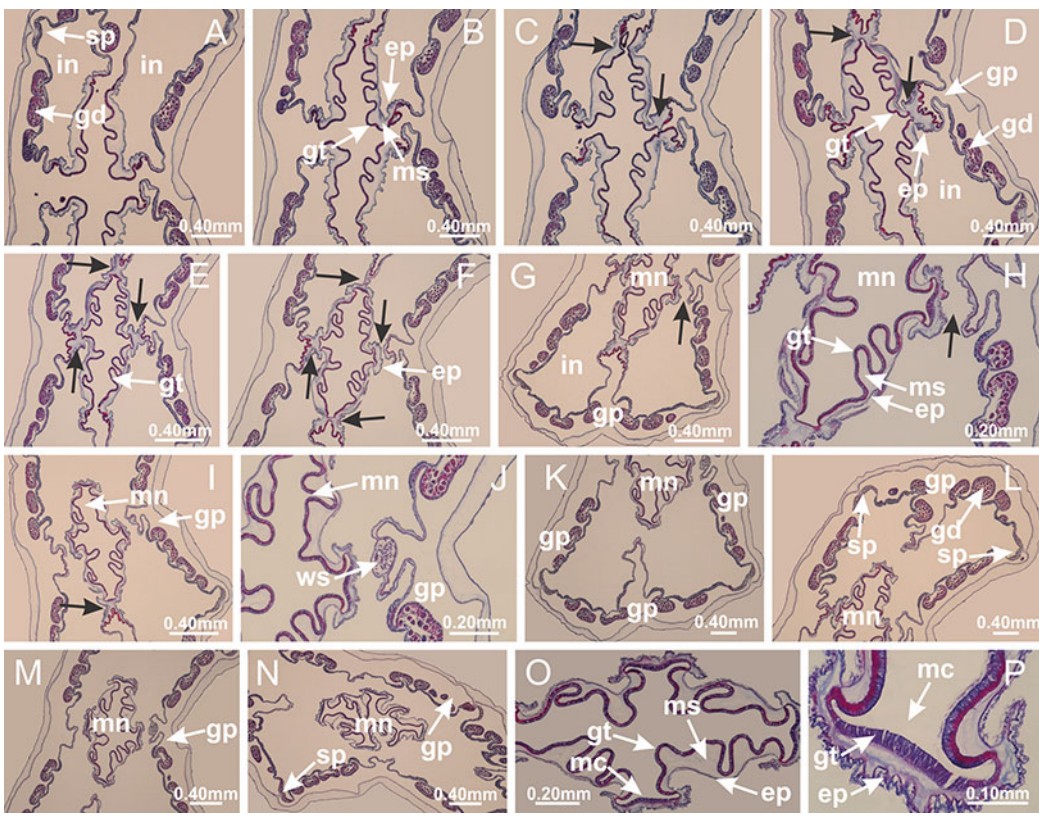

**Figure 6 Manubrium and gastric radial pockets.** *Haliclystus tenuis*: (A) internal organization at calyx base, below manubrium delimitation; (B–D) gastrodermis of adjacent septa gradually merges (indicated by black arrows), delimiting gastrodermis of manubrium and gastrodermis of gastric radial pockets; (E–G) epidermis of adjacent septa (infundibula) gradually merges (indicated by black arrows), delimiting epidermis of manubrium and epidermis of gastric radial pockets; (H) internal organization at base of manubrium; (I–N) manubrium and gastric radial pockets with gonads; gastric radial pockets separated by interradial septa; (O) internal organization of manubrium completely delimited; (P) manubrium corner. (A–P): cross sections. See Table 2 for abbreviations.

gastrodermis of gastric radial pockets through stem (Figs. 9A–9I). At interradial regions, septa prevent direct connection of gastrodermis of interradial anchors with gastrodermis of calyx (Fig. 10A), but at margin of calyx, small ostia connect adjacent gastric radial pockets along margin of calyx, allowing gastrodermis of anchors to be contiguous with gastrodermis of calyx (gastric radial pocket) (Figs. 10B–10F). Anchors without nematocysts. Each gastric radial pocket extending throughout calyx margin, apically continuing into two adradial arms and respective tentacular clusters (Figs. 11A–11H). Subumbrellar epidermis (continuous with epidermis of infundibula) marginally merges with exumbrellar epidermis, dividing gastric radial pockets at origin of arms (Figs. 11A–11H). Eight bands of longitudinal muscles running between calyx base and arms, each band toward each one of eight arms (Figs. 11B–11H), then becoming thinner diffuse muscle bundles toward secondary tentacles (as in other species examined; Figs. 5Q–5V). Eight sections of coronal muscle (Fig. 3E) at calyx margin, each between adjacent arms (as in most of species examined; Figs. 5 and 11). Each arm with two bands (perradial and interradial) of coronal muscle (Figs. 5Q, 11G and 11H). Perradial and

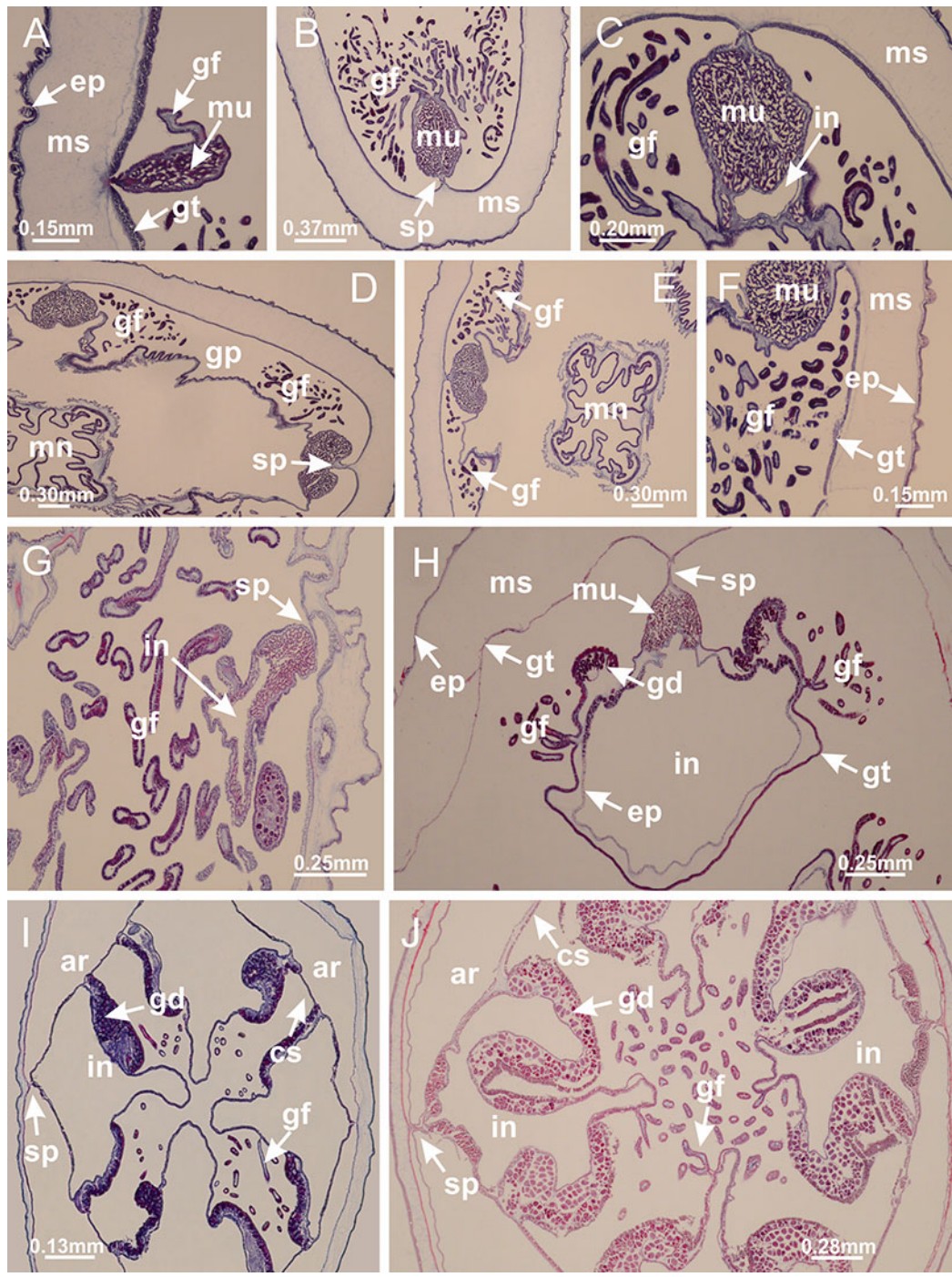

**Figure 7 Gastric filaments.** *Calvadosia corbini*: (A–C) formation of gastric filaments through lateral evagination of septal gastrodermis; (D–F) gastric filaments associated with gastric radial pockets; *Lucernaria quadricornis*: (G) formation of gastric filaments through lateral evagination of septal gastrodermis; *Calvadosia vanhoeffeni*: (H) formation of gastric filaments through lateral evagination of septal gastrodermis; *Manania uchidai*: (I) gastric filaments; *Craterolophus convolvulus*: (J) gastric filaments. (A–J): cross sections. See Table 2 for abbreviations.

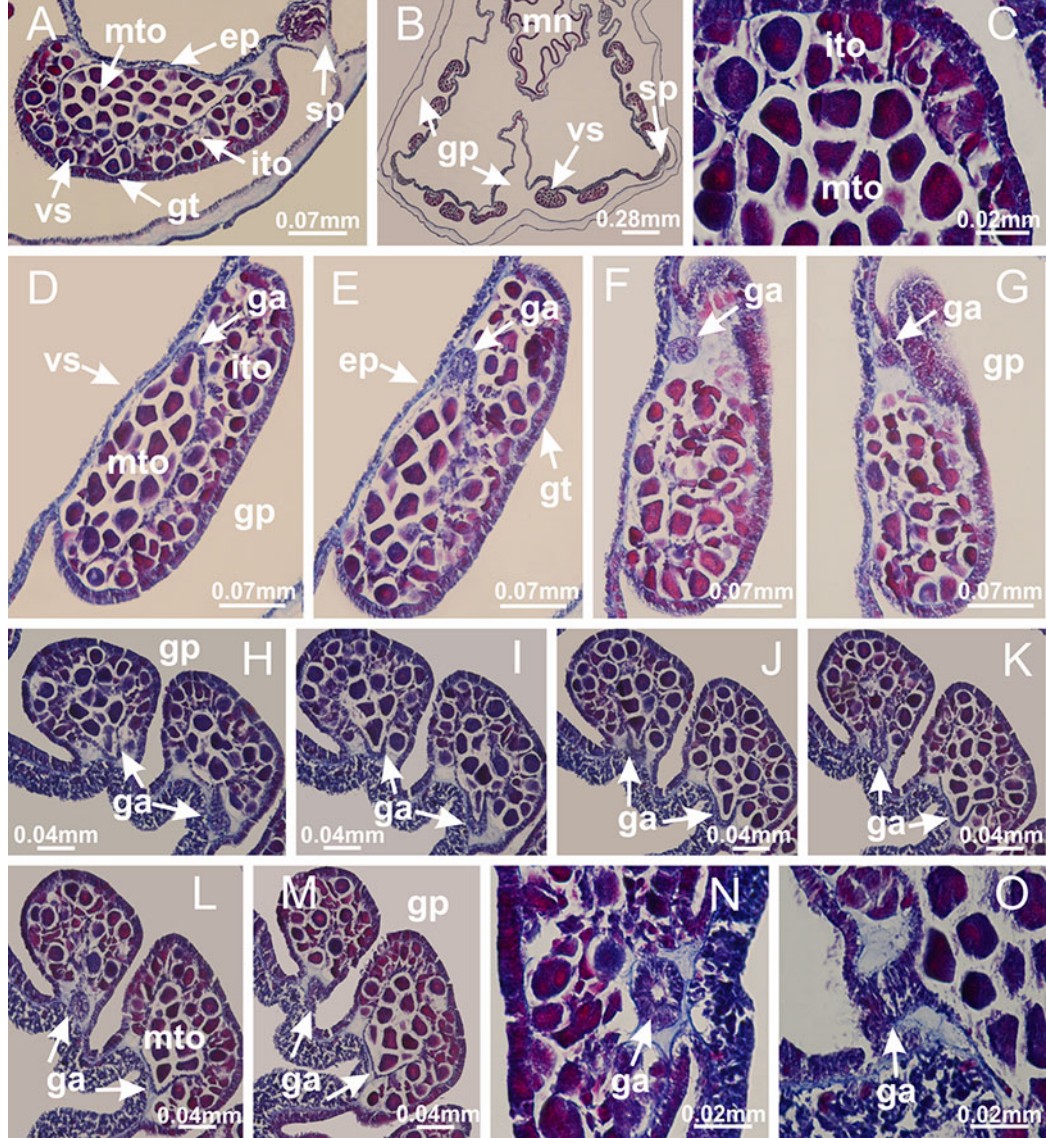

**Figure 8 Gonads and gametoduct of *Haliclystus tenuis*.** (A) Female vesicle, with peripheral layer of immature oocytes, adjacent to gastrodermis, and a central layer of mature oocytes, adjacent to epidermis; (B) general view of vesicles inside gastric radial pockets; (C) detail of immature and mature oocytes; (D–G) sequence of gametoduct connecting the mature oocytes with gastrovascular cavity; (H–M) sequence of gametoduct connecting the mature oocytes with gastrovascular cavity, in two adjacent vesicles; (N, O) detail of gametoduct. (A–N): cross sections; O: longitudinal section. See Table 2 for abbreviations.

interradial white spots of nematocysts on subumbrella (Fig. 3G), between a layer of epidermis and gastrodermis, internally composed of peripheral layer of nematoblasts, and central mature nematocysts (Figs. 12A–12C). Epidermal thickening at central region of white spots of nematocysts (Fig. 12C). Batteries of nematocysts sparsely distributed in exumbrellar epidermis (Fig. 13C). Distal exumbrellar end of arms with "U-shaped" space, a platform connecting arm with secondary tentacles, defined by gastrodermis and a thick layer of mesoglea (Fig. 14). Continuous layer of internal unorganized nematocysts

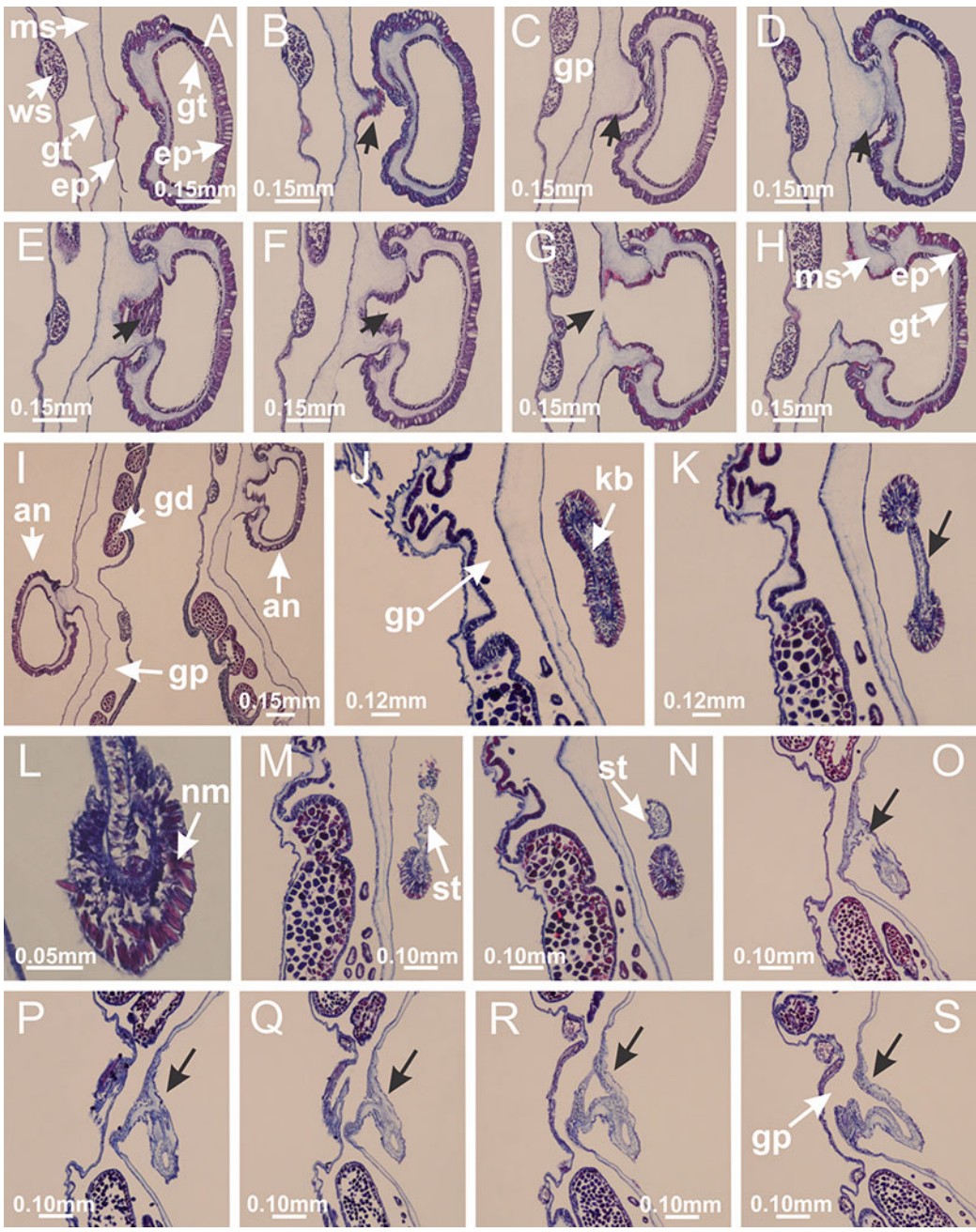

**Figure 9 Perradial anchors and perradial primary tentacles.** *Haliclystus tenuis*: (A–H) gastrodermis of the hollow perradial anchor connecting with gastrodermis of calyx, through the stem of anchor (indicated by black arrows); (I) general view of anchor at calyx margin; *Calvadosia cruciformis*: (J–S) gastrodermis of the hollow perradial primary tentacle connecting with gastrodermis of calyx, through the stem of primary tentacle (indicated by black arrows). (L) detail of the nematocysts present in the knob of primary tentacle. (A–S): longitudinal sections of anchors and primary tentacles (cross sections of animals). See Table 2 for abbreviations.

visible in subumbrellar epidermis, from base of infundibula, passing through gastric radial pockets, arms, to tips of secondary tentacles (as in most of species examined; Figs. 15 and 16). Internal layer of nematocysts continuous with groups of nematocysts at

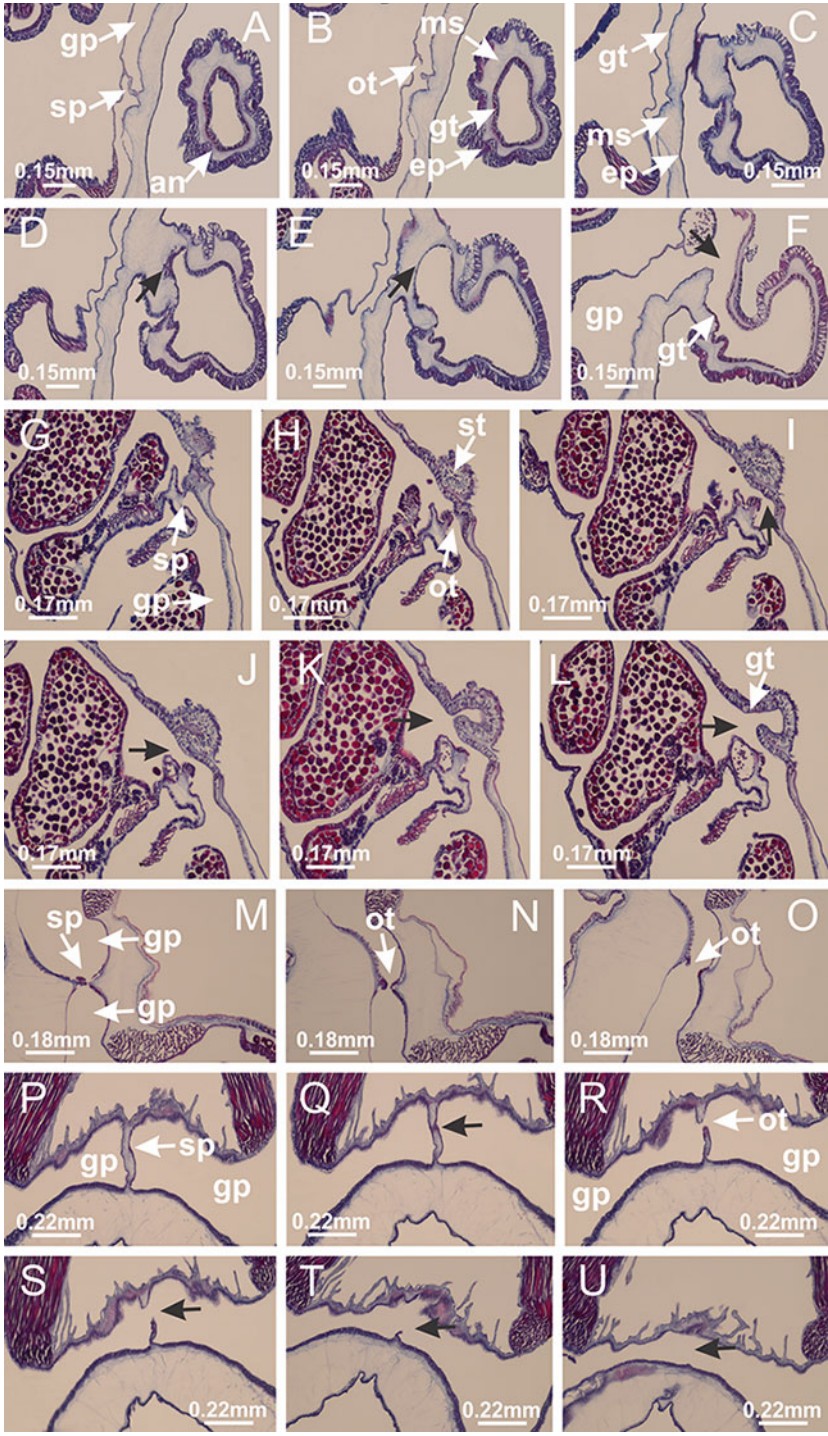

**Figure 10 Interradial anchors, interradial primary tentacles, and ostia.** *Haliclystus tenuis*: (A–F) gastrodermis of hollow interradial anchor connecting with gastrodermis of calyx (indicated by black arrow), through the stem of anchor, by means of small ostium; *Calvadosia cruciformis*: (G–L) gastrodermis of hollow interradial primary tentacle connecting with gastrodermis of calyx (indicated by black arrow), through the stem of primary tentacle, by means of small ostium; *Calvadosia vanhoeffeni*: (M–O) septum detaches from layer of gastrodermis of calyx, forming an ostium, connecting two adjacent gastric radial pockets; *Calvadosia corbini*: (P–U) septum detaches from layer of gastrodermis of calyx, forming an ostium, connecting two adjacent gastric radial pockets (indicated by black arrows). (A–U): cross sections ((A–L): longitudinal sections of anchors and primary tentacles). See Table 2 for abbreviations.

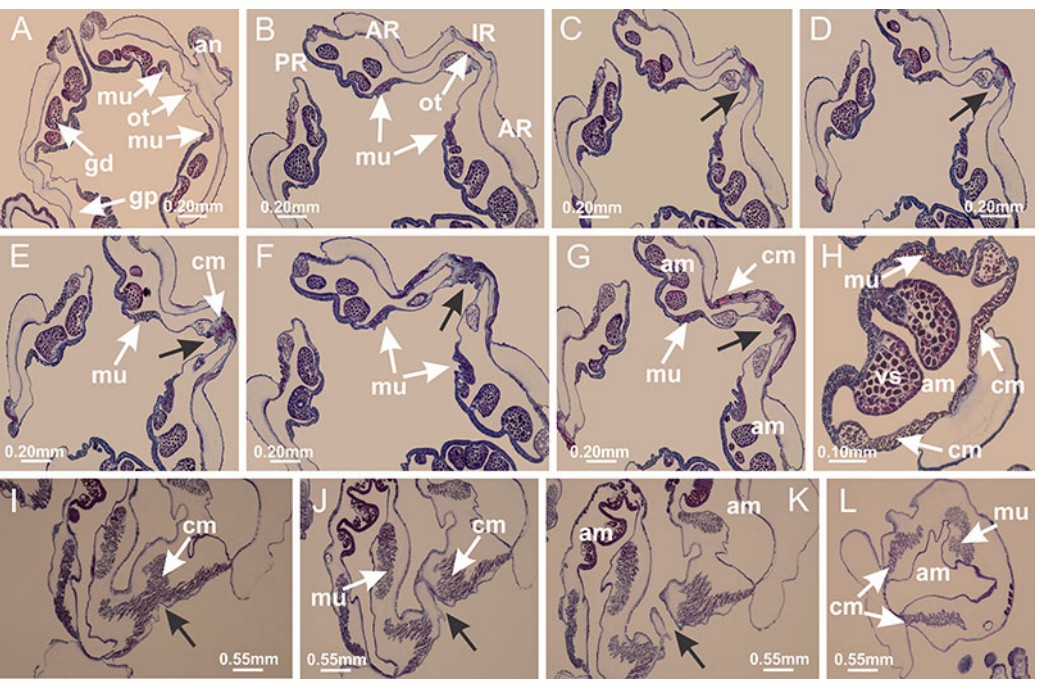

**Figure 11 Arms delimitation.** *Haliclystus tenuis*: (A) general view of the calyx margin, with perradial notches separated; (B–G) progressive separation of interradial notches (indicated by black arrows) through fusion of gastrodermis and epidermis of subumbrella and exumbrella, delimiting the arms; (H) arm, composed of one central band of longitudinal muscle and two lateral bands of coronal muscle; *Calvadosia vanhoeffeni*: (I–K) progressive separation of interradial notches (indicated by black arrows) through fusion of gastrodermis and epidermis of subumbrella and exumbrella, delimiting the arms; (L) arm, composed of one central band of longitudinal muscle and two lateral bands of coronal muscle. (A–L): cross sections. See Table 2 for abbreviations.

tentacular base, in epidermis, with different sizes and types, also unorganized (as in other species examined; Fig. 16). Secondary hollow tentacles (Fig. 3I) composed of two parts, knob and stem (Fig. 17N). Secondary tentacles without pad-like adhesive structures. Each stem with inner layer of gastrodermis, and external layer of epidermis; epidermis with longitudinal muscles extending throughout tentacular stem (Fig. 17O). Nematocysts found at different regions of epidermis of stem of secondary tentacles (as in other species examined; Fig. 16Q). Tentacular knob with a thin layer of gastrodermis and a thick layer of epidermis with an external row of organized nematocysts (Figs. 17P and 17Q). Nematocysts also found at internal region of knob, among supporting cells of epidermis.

Genus *Manania Clark, 1863*

*Manania uchidai* (*Naumov, 1961*) (Figs. 2I, 2J; 5A, 5B, 5N; 7I; 12D, 12E; 13A, 13B; 17K–17M; 18; 19; 20; 21B; 22; 23; 24L–24N; 57; 58B, 58E)

Basal region formed by pedal disk of peduncle with increased surface area due to invaginations (Figs. 18F and 18G). Peduncle with one chamber (delimited by gastrodermis) and four interradial longitudinal muscle bands (epitheliomuscular cells) embedded in mesoglea, near the base (Figs. 19A–19F). Single chamber progressively divided into four chambers, at median region of peduncle, alternating with four

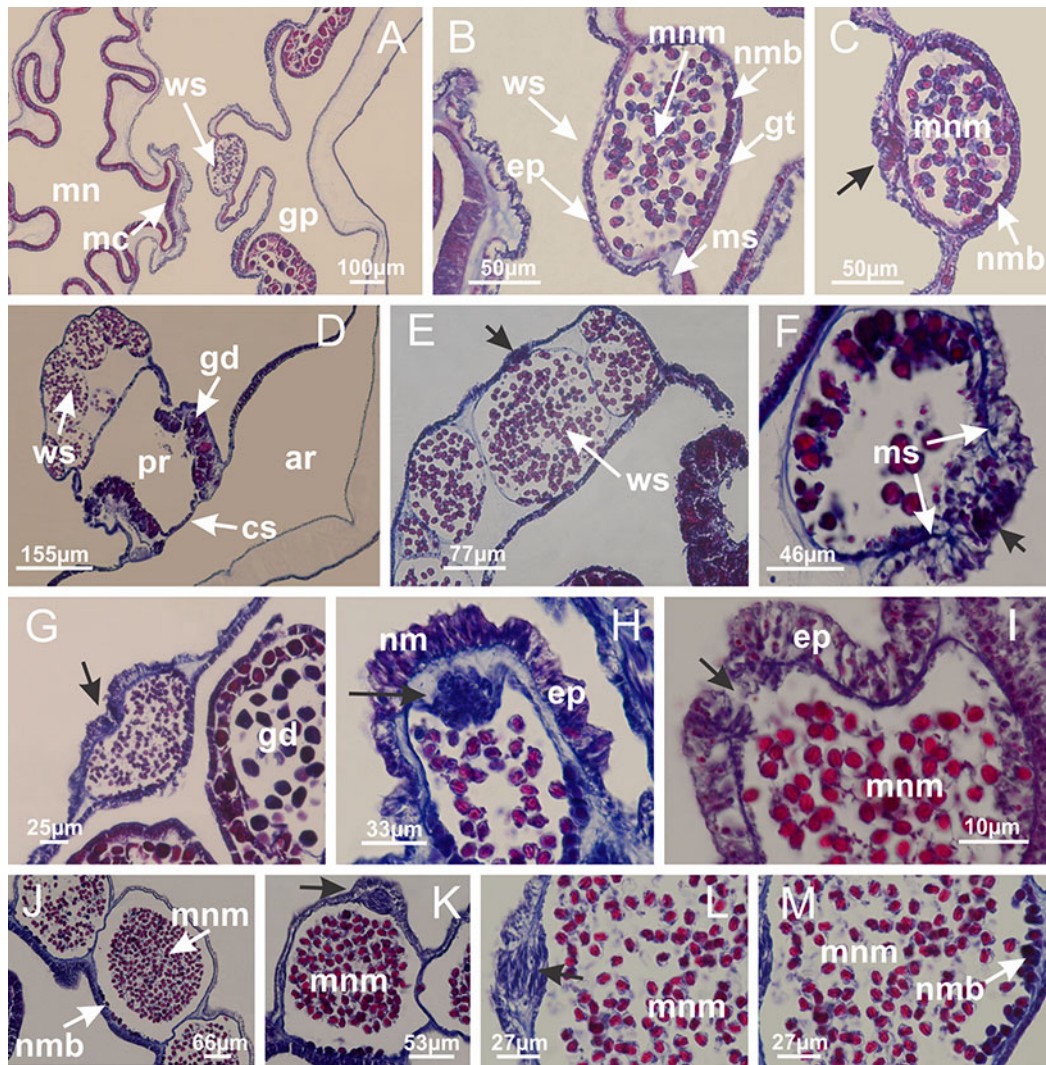

**Figure 12 White spots of nematocysts.** *Haliclystus tenuis*: (A) general view of perradial white spot of nematocysts; (B) internal organization, with central mature nematocysts and a peripheral layer of nematoblasts; (C) central thickening of epidermis of white spot (indicated by black arrow); *Manania uchidai*: (D) white spots, associated with principal radial pocket and gonads; (E) central thickening of epidermis of white spot (indicated by black arrow); *Calvadosia vanhoeffeni*: (F) central thickening of epidermis of white spot (indicated by black arrow); *Calvadosia cruciformis*: (G, H) central thickening of epidermis of white spots (indicated by black arrows); *Lucernaria sainthilairei*: (I) possible communication of mature nematocysts with the outside (indicated by black arrow); *Calvadosia corbini*: (J) internal organization, with central mature nematocysts and a peripheral layer of nematoblasts; (K) central thickening of epidermis of white spot (indicated by black arrow); (L, M) detail of internal organization, with central mature nematocysts and a peripheral layer of nematoblasts, and central thickening of epidermis of white spot (indicated by black arrow). (A–M): longitudinal sections of white spots of nematocysts (cross sections of animals). See Table 2 for abbreviations.

interradial longitudinal muscle bands (Figs. 19G–19R). Perradial chambers do not merge at junction of peduncle and calyx, unlike other stauromedusae (Figs. 19S, 19T, 20A and 20B). Instead, claustra are defined below complete connection of perradial chambers (Figs. 20C–20G). Union of lateral projections of adjacent interradial septa forming claustra (Figs. 20D–20F), tissues composed of central layer of mesoglea surrounded by

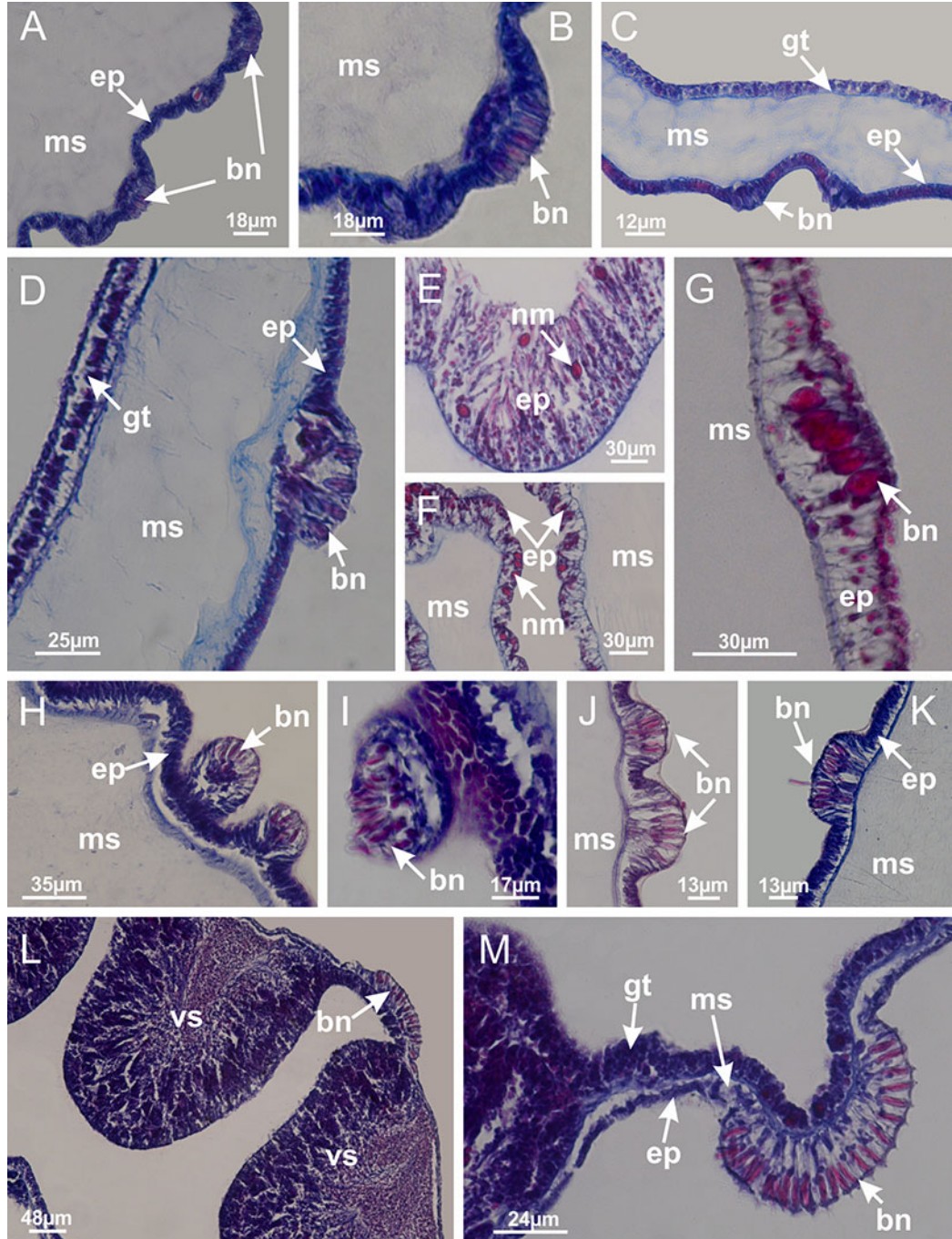

**Figure 13  Batteries of nematocysts.** *Manania uchidai*: (A, B) batteries of nematocysts in the epidermis of exumbrella; *Haliclystus tenuis*: (C) batteries of nematocysts in the epidermis of exumbrella; *Calvadosia cruciformis*: (D) batteries of nematocysts in the epidermis of exumbrella; *Calvadosia vanhoeffeni*: (E, F) nematocysts sparsely distributed in the epidermis of exumbrella of pedal disk; (G) batteries of nematocysts in the epidermis of exumbrella of calyx; *Calvadosia corbini*: (H–K) batteries of nematocysts in the epidermis of exumbrella; (L, M) batteries of nematocysts in the epidermis of subumbrella, associated with gonads. (A–M): cross sections. See Table 2 for abbreviations.

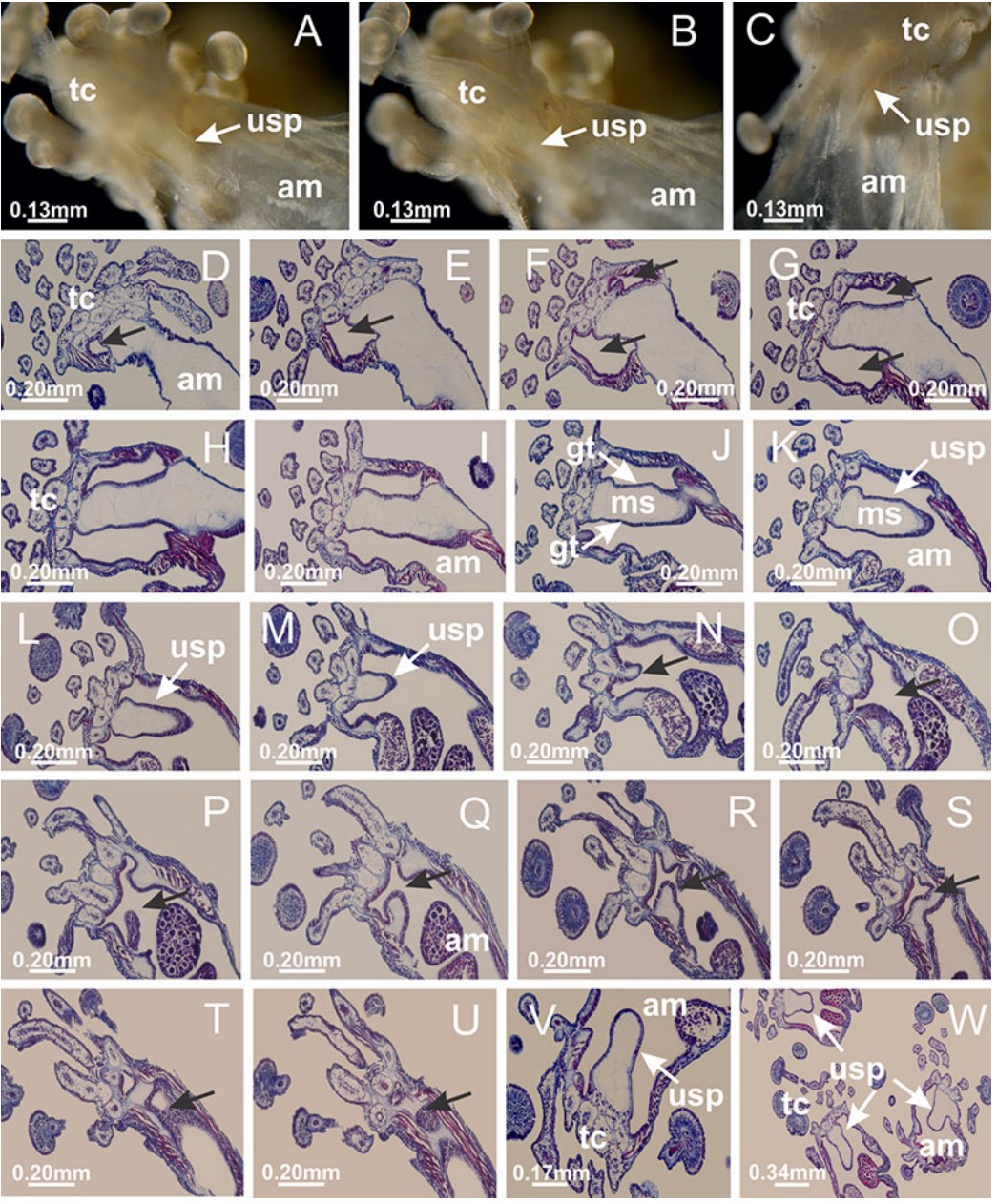

**Figure 14** **"U-shaped" space of *Haliclystus tenuis*.** (A–C) General view of tip of arm, at the internal base of tentacles, with a platform or "U-shaped" space; (D–U) sequence of longitudinal sections of arms, showing the "U-shaped" space (delimitation indicated by black arrows): a double layer of gastrodermis with a central layer of mesoglea; (V, W) general view of "U-shaped" space. (A–W): longitudinal sections. See Table 2 for abbreviations.

gastrodermis, dividing gastrovascular cavity (Figs. 20G and 20H). Four accessory radial pockets delimited (separated from main gastrovascular cavity) by claustra (Figs. 20F–20M). Four infundibula funnel-shaped with blind end, delimited by epidermis, deeply developed down to base of calyx, widening apically, with broad apertures on subumbrella (Figs. 19T, 20C, 20G and 20H). At base of infundibula, interradial longitudinal muscles remain intramesogleal (Fig. 21B), being progressively (toward

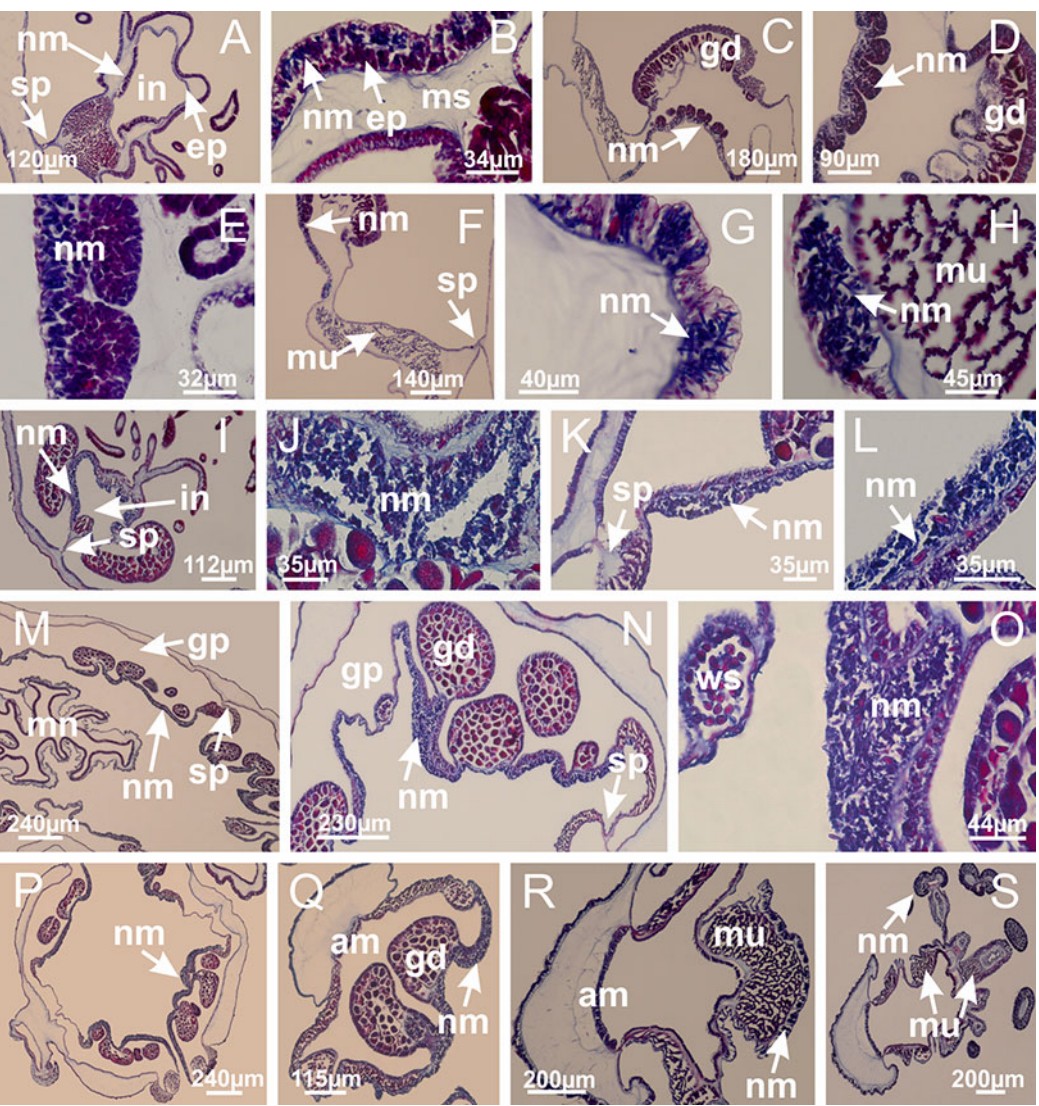

**Figure 15 Internal subumbrellar layer of nematocysts.** *Calvadosia vanhoeffeni*: (A–H) layer of nematocysts in the epidermis of the base of infundibula; *Haliclystus tenuis*: (I–L) layer of nematocysts in the epidermis of infundibula; (M–P) layer of nematocysts in the subumbrellar epidermis of calyx, associated with gastric radial pockets; (Q) layer of nematocysts in the subumbrellar epidermis of arms; *Calvadosia corbini*: (R) layer of nematocysts in the subumbrellar epidermis of arms; (S) layer of nematocysts in the subumbrellar epidermis of arms associated with secondary tentacles. (A–S): cross sections. See Table 2 for abbreviations.

manubrium) compressed, flattened, and divided into two thin bands (Figs. 20I, 20L and 20M). Adjacent septal gastrodermis merge again, dividing gastrovascular cavity once more, forming four principal radial pockets and manubrium (Figs. 20H–20M), similarly (and homologous) to formation of gastric radial pockets in all staurozoan species without claustrum such as *H. tenuis* (Fig. 6). Central part of gastrodermis of each septum joins forming four-sided manubrial gastrodermis, while lateral parts of adjacent septa join forming principal radial pockets (as in the formation of gastric radial pockets of *H. tenuis*, Figs. 6B–6D). Similarly, each infundibular epidermis progressively merges

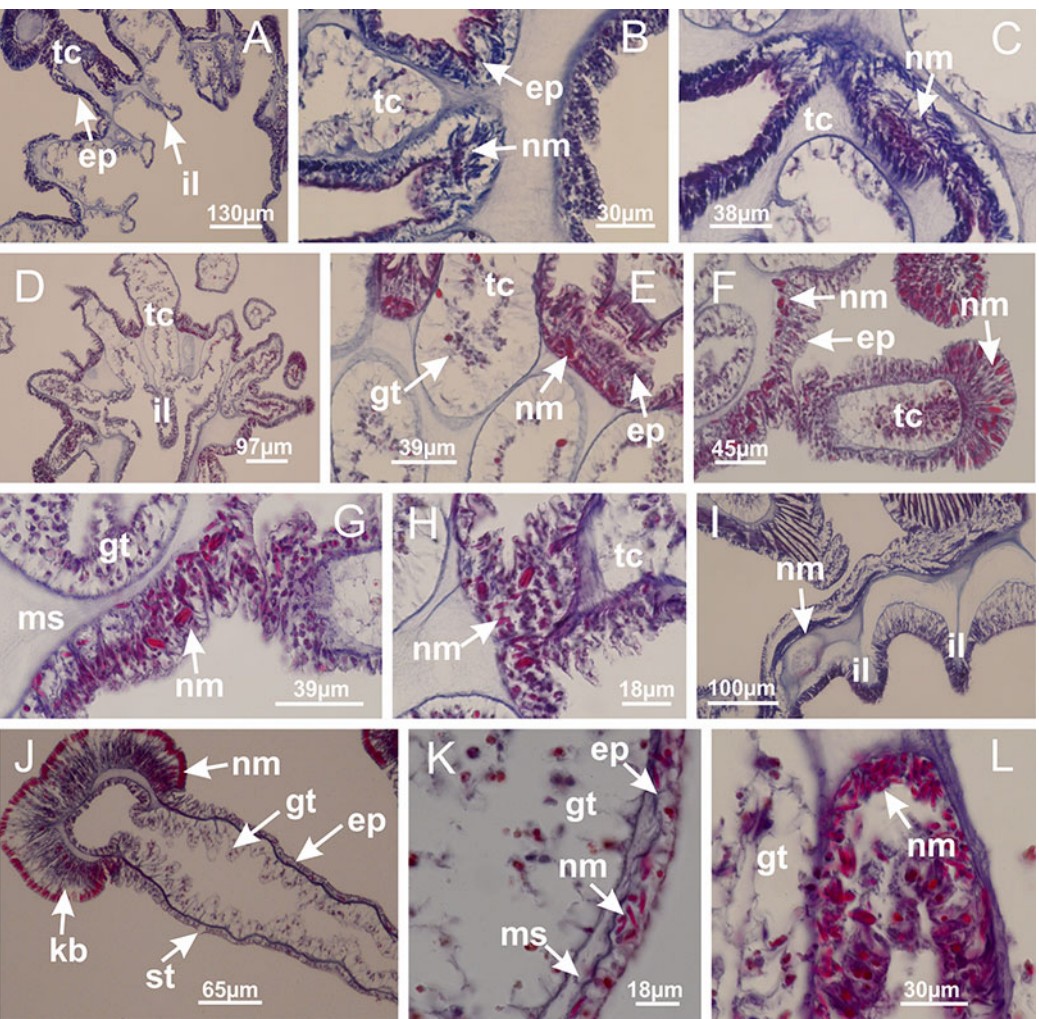

**Figure 16 Internal layer of nematocysts in the secondary tentacles.** *Calvadosia vanhoeffeni*: (A) general view of tips of arms and secondary tentacles; (B, C) accumulation of nematocysts in the epidermis of secondary tentacles (base of stem); *Lucernaria quadricornis*: (D) general view of tips of arms and secondary tentacles; (E–H) accumulation of nematocysts in the epidermis of secondary tentacles; *Calvadosia corbini*: (I) accumulation of nematocysts in the epidermis of secondary tentacles (base of stem); *Lucernaria sainthilairei*: (J) general view of secondary tentacle; (K) detail of nematocysts in the epidermis of tentacular stem; (L) detail of the unorganized internal group of nematocysts, in the epidermis of tentacular stem base. (A–L): longitudinal sections. See Table 2 for abbreviations.

apically: central part of each infundibular epidermis becomes manubrial epidermis, and epidermis of adjacent infundibula forms epidermis of principal radial pockets (as in *H. tenuis* Figs. 6E–6G, 20I and 20J). Principal radial pockets (Figs. 20J–20M) are true gastric radial pockets, composed of same structures as gastric radial pockets in stauromedusae without claustrum (Fig. 6) and associated with gonads (and gametoduct) (Fig. 20M). Therefore, *M. uchidai* has eight radial pockets: four accessory radial pockets, directly associated with chambers in peduncle, anchors and arms; and four principal radial pockets, associated with manubrium and gonads (Figs. 20J and 20L). Four accessory and principal radial pockets separated by claustra (Figs. 20L and 20M). Four accessory radial pockets laterally separated from each other by interradial septa (Fig. 20L);

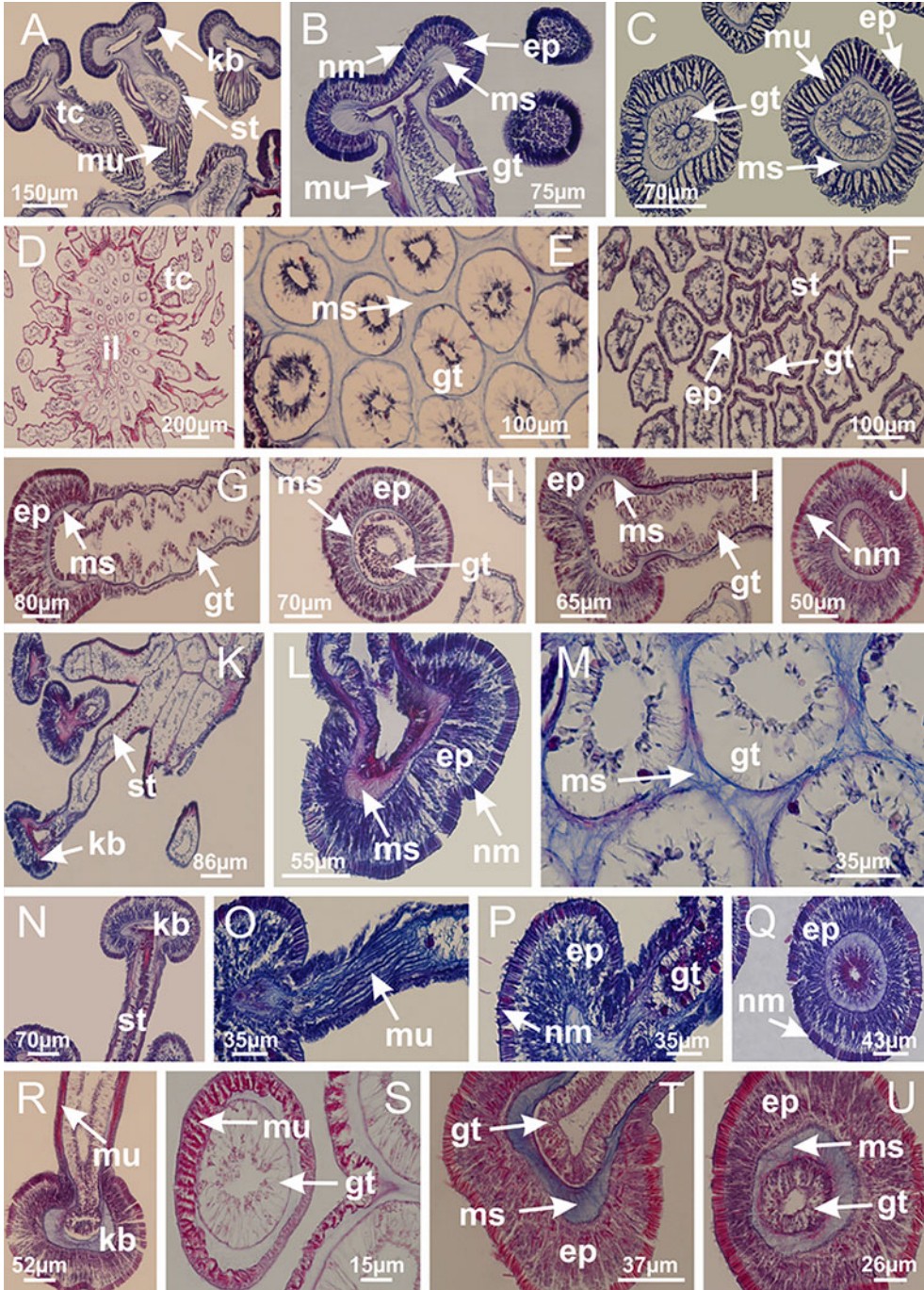

**Figure 17 Secondary tentacles.** *Calvadosia corbini*: (A, B) general organization of secondary tentacles; (C) tentacular stem base highlighting the longitudinal muscle associated with epidermis, and gastrodermis with vacuolated cells; *Lucernaria quadricornis*: (D) intertentacular lobules and tentacular stem base; (E) intertentacular lobules; (F) tentacular stem base; *Lucernaria bathyphila*: (G) general organization of secondary tentacles; (H) tentacular knob with tall epidermis and nematocysts on its apex; *Lucernaria sainthilairei*: (I) general organization of secondary tentacles; (J) tentacular knob with tall epidermis and nematocysts on its apex; *Manania uchidai*: (K) general organization of secondary tentacles; (L) tentacular knob with tall epidermis and nematocysts on its apex; (M) intertentacular lobules; *Haliclystus tenuis*: (N) general organization of secondary tentacles; (O) detail of longitudinal muscle of tentacular stem; (P, Q) tentacular knob with tall epidermis and nematocysts on its apex; *Craterolophus convolvulus*: (R) general organization of secondary tentacles; (S) tentacular stem base highlighting the longitudinal muscle associated with epidermis, and gastrodermis with vacuolated cells; (T, U) tentacular knob with tall epidermis and nematocysts on its apex. (A, B, G, I, K, L, N–P, R, T): longitudinal sections; (C–F, H, J, M, Q, S, U): cross sections. See Table 2 for abbreviations.

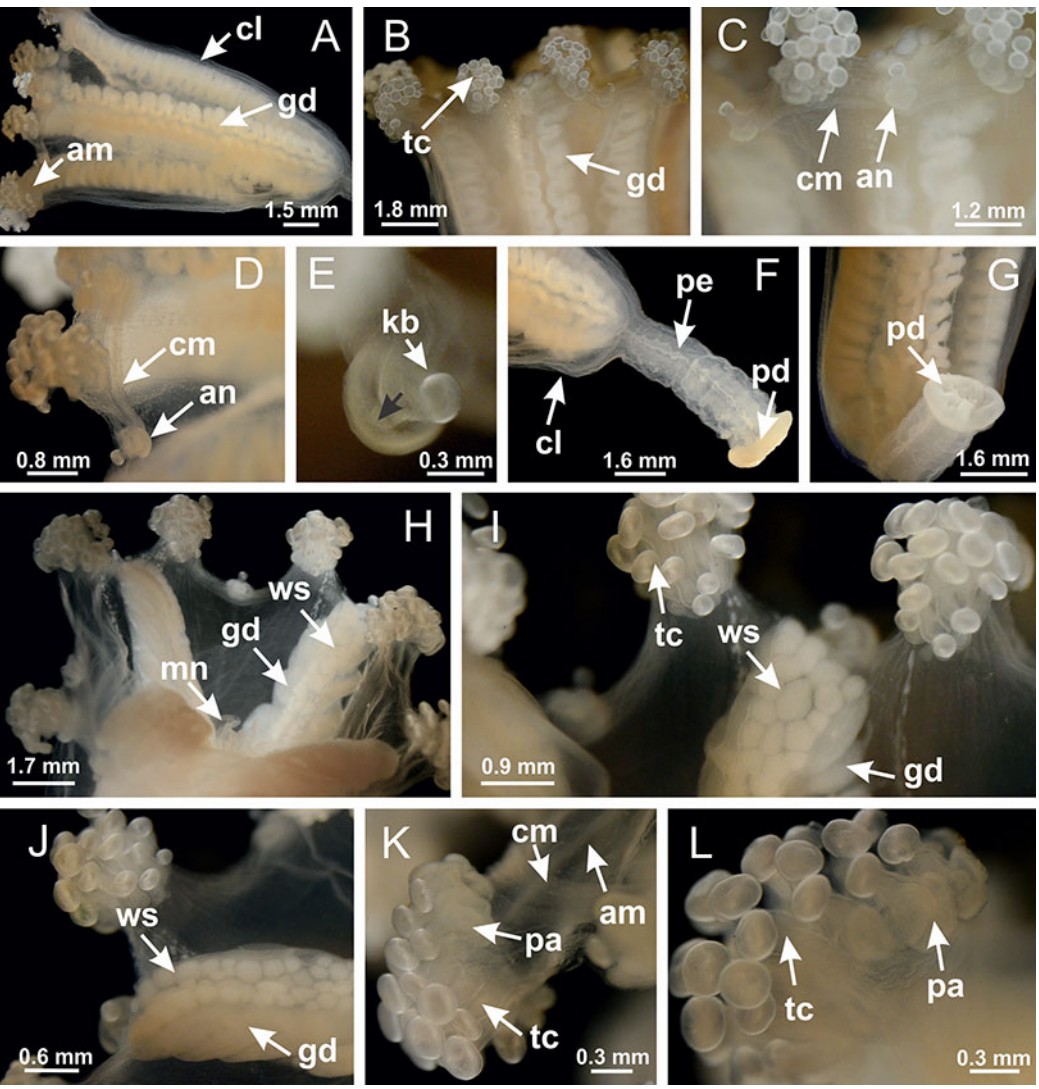

**Figure 18 General view of *Manania uchidai*.** (A) General view of calyx; (B–D) apical region of calyx, with continuous coronal muscle; (E) detail of anchor, with a knobbed remnant of primary tentacle, and a swollen base (indicated by black arrow); (F, G) peduncle and pedal disk; (H) subumbrellar view, with manubrium and gonads; (I, J) white spots, associated with perradial gonads; (K, L) exumbrellar coronal muscle (external to anchors) and tentacular cluster, with pad-like adhesive structures in the outermost secondary tentacles. See Table 2 for abbreviations.

accessory radial pockets directly connected only by means of small interradial ostia at margin of calyx (as in the gastric pocked of *H. tenuis*, Figs. 10A–10F). Manubrium internally defined by gastrodermis, externally by epidermis. Gastric filaments similar to those described for *H. tenuis* (as in other species examined; Fig. 7), and associated with principal radial pockets. Gonads (Figs. 18H–18J) in principal radial pockets, not organized in vesicles (Fig. 22). Gonadal content restricted to one layer between gastrodermis and epidermis of septa (Figs. 22B, 22D and 22G). Gonadal layers of same principal radial pocket formed by lateral tissue of two different interradial septa (two adjacent septa) (Figs. 20H–20K). Male specimen analyzed (Fig. 22), with spermatocytes

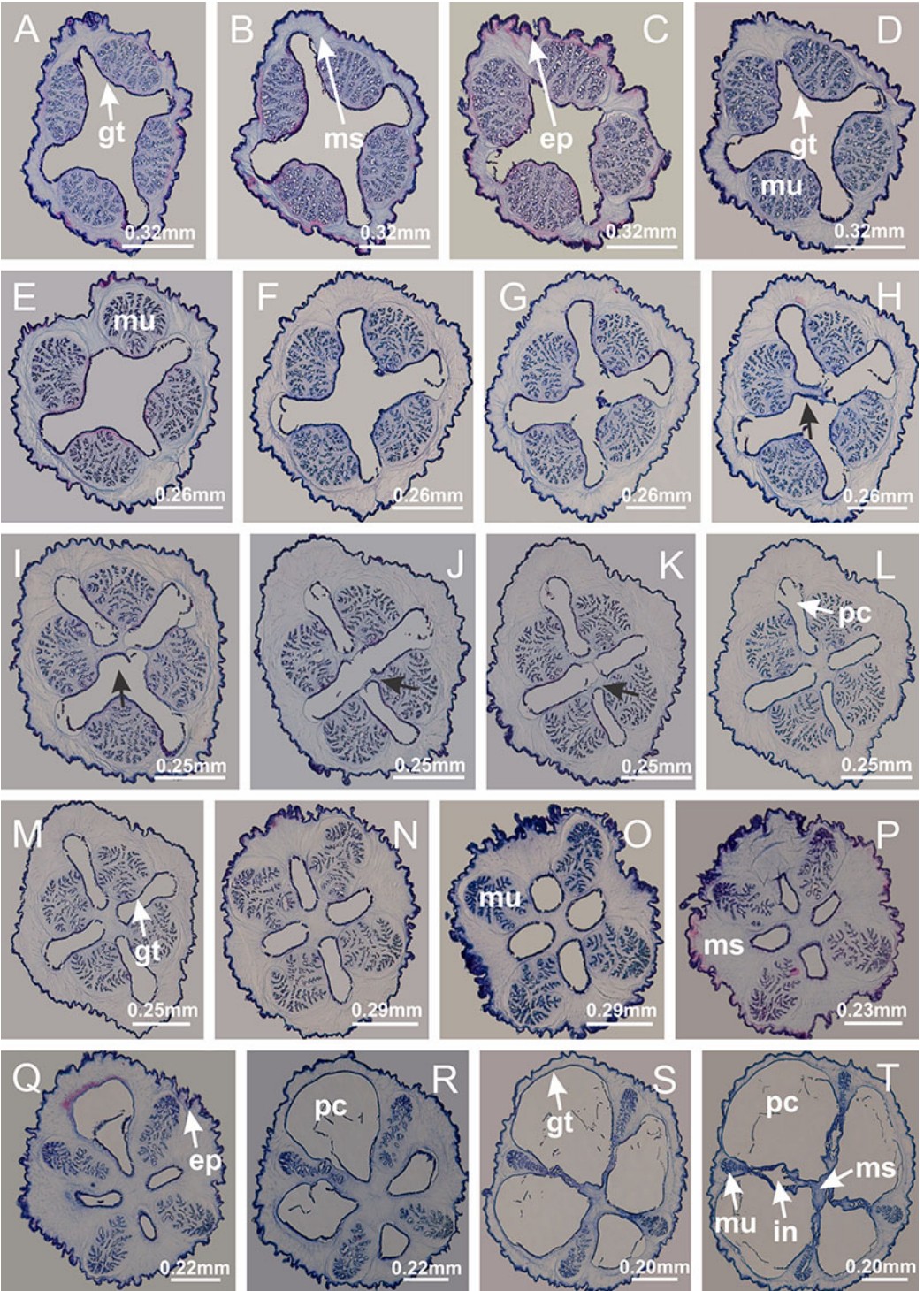

**Figure 19 Peduncle and septa of *Manania uchidai* (from base moving upward in A–T).** (A) Base of peduncle with one chamber delimited by gastrodermis and four interradial longitudinal muscles; (B–G) variation in shape and size of the central chamber; (H–L) gradual division of central chamber in four perradial chambers (indicated by black arrows); (M–S) variation in shape and size of four perradial chambers; (T) four interradial septa, with infundibula delimited by epidermis, connected by a central mesoglea. (A–T): cross sections. See Table 2 for abbreviations.

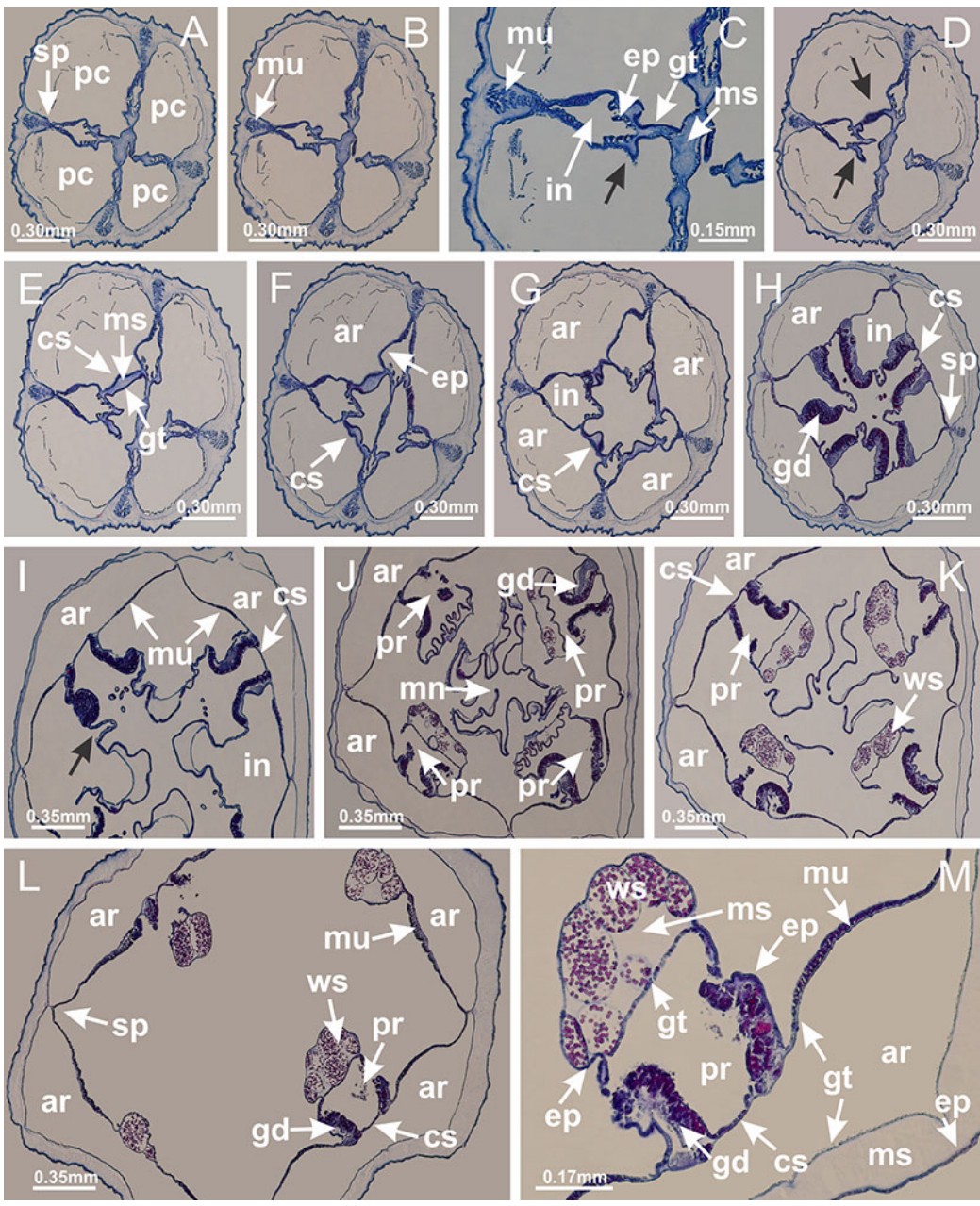

**Figure 20 Claustra in *Manania uchidai*.** (A–F) Claustra delimitation (lateral projections of adjacent septa, indicated by black arrows); (G, H) claustra completely delimited, dividing the gastrovascular cavity, forming accessory radial pockets; (I, J) fusion of gastrodermis and epidermis of adjacent septa (indicated by black arrow), delimiting manubrium and principal radial pocket; (K, L) principal radial pocket, associated with gonads and white spots of nematocysts; (M) detail of principal radial pocket. (A–M): cross sections. See Table 2 for abbreviations.

adjacent to gastrodermis of principal radial pockets (internal), and spermatozoa adjacent to epidermis of principal radial pockets (external) (Figs. 20M and 22G). Spermatozoa divided into different sacs, delimited by cells of gastrodermal origin, which are connected to gastrodermis of principal radial pocket, forming gametoduct (Figs. 22E, 22I–22N). Cilia often associated with gametoduct. Anchors (Figs. 18C–18E) hollow, with hollow

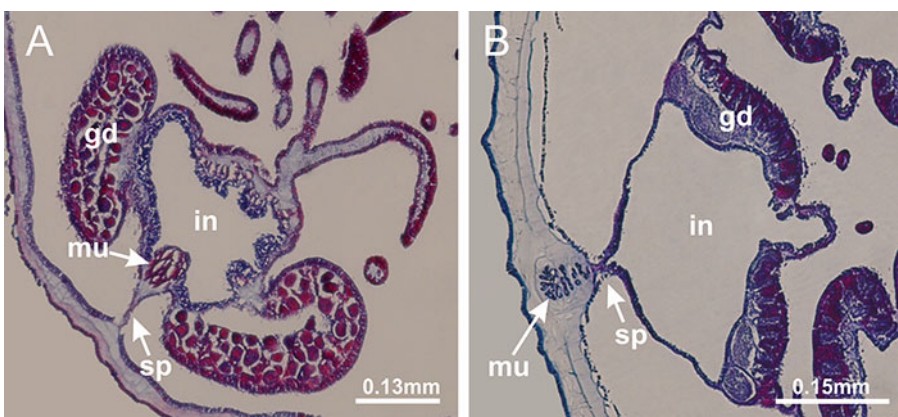

**Figure 21 Position of interradial longitudinal muscle.** *Haliclystus tenuis*: (A) interradial longitudinal muscle internal in relation to septa; *Manania uchidai*: (B) interradial longitudinal muscle external in relation to septa. (A, B): cross sections. See Table 2 for abbreviations.

stem, as evaginations of body surface (Fig. 5N). Eight small anchors, each anchor located between adjacent arms at calyx margin, four perradial and four interradial, (Figs. 18C–18E). Internal organization similar to *H. tenuis*, with interradial ostia (Figs. 9A–9I, 10A–10F). Anchors connected to accessory radial pocket. Nematocysts present in knobbed remnant of primary tentacles at tip of anchors (Figs. 5N and 18E). Coronal muscle entire at calyx margin, and external (exumbrellar) in relation to anchor (Figs. 18C and 18D). Each accessory radial pocket extending throughout calyx margin, apically continuing into two adradial arms and respective tentacular clusters. Internal organization of arms similar to *H. tenuis* (Figs. 11A–11H). Perradial white spots of nematocysts on subumbrella, associated with gonads (Figs. 18H–18J, 20K–20M), between epidermis and gastrodermis of principal radial pockets, with internal organization (Figs. 12D and 12E) similar to *H. tenuis*. Batteries of nematocysts sparsely distributed in exumbrellar epidermis (Figs. 13A and 13B). Distal end of arms with intertentacular lobules, a structure between adjacent secondary tentacles delimited by gastrodermis and one central layer of mesoglea (Figs. 17M and 23). Outermost secondary tentacles with pad-like adhesive structures (epidermal thickening) (Figs. 24L–24N). Continuous layer of internal unorganized nematocysts in subumbrellar epidermis not clearly recognizable. Secondary hollow tentacles composed of two parts, knob and stem, with organization (Figs. 17K and 17L) similar to *H. tenuis*. At stem base, secondary tentacles tightly joined, separated only by thin layer of mesoglea, with beehive appearance in cross section (Fig. 17K).

Family Lucernariidae *Johnston, 1847*

Genus *Lucernaria Müller, 1776*

*Lucernaria quadricornis Müller, 1776* (Figs. 7G; 16D–16H; 17D–17F; 25; 26; 27; 28; 57)

Basal region formed by pedal disk of peduncle with increased surface area due to invaginations (Fig. 25C). Peduncle with one chamber (delimited by gastrodermis), and four interradial gastric septa: each septum consisting of mesoglea surrounded by

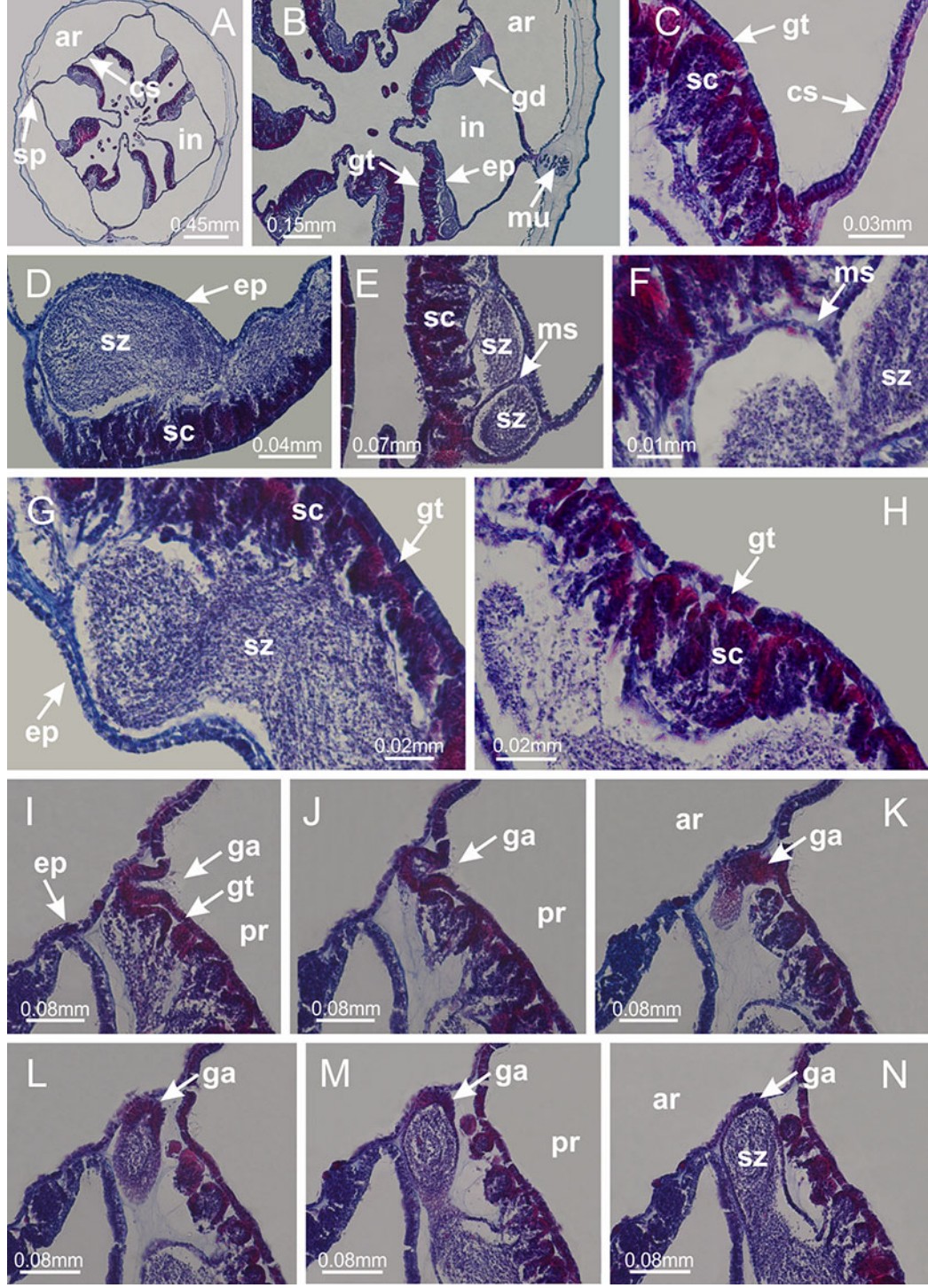

**Figure 22 Gonads and gametoduct of *Manania uchidai*.** (A) General view of septa (calyx base) with gonadal content, below delimitation of principal radial pockets; (B) gonadal content between a layer of gastrodermis (adjacent to spermatocytes) and epidermis (adjacent to spermatozoa); (C) detail of gonad adjacent to claustrum; (D–H) organization of male gonad, with spermatocytes and spermatozoa; (I–N) sequence of gametoduct connecting the spermatozoa with the gastrovascular cavity of principal radial pocket. (A–N): cross sections. See Table 2 for abbreviations.

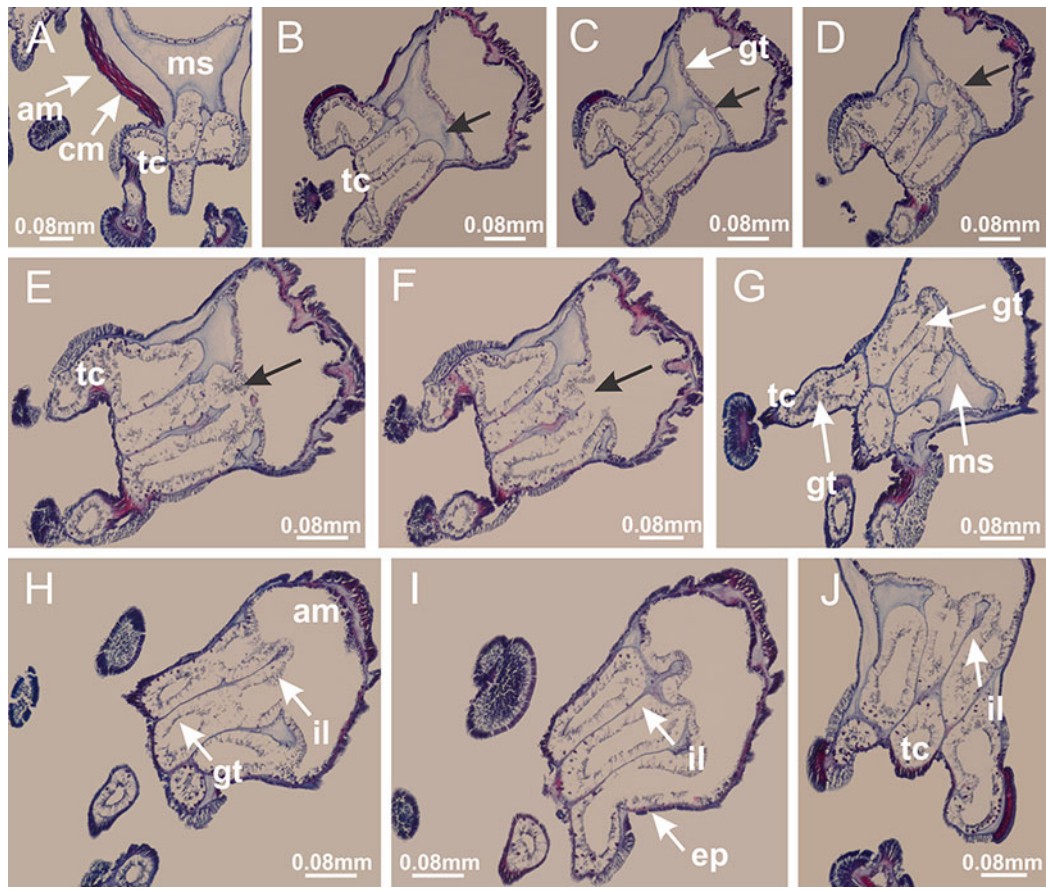

**Figure 23 Intertentacular lobules of *Manania uchidai*.** (A) Tip of arm; (B–F) delimitation of inter-
tentacular lobules (indicated by black arrows; connection of secondary tentacles with gastrovascular
cavity in arm); (G–J) intertentacular lobules, structures composed of a double layer of gastrodermis
(of adjacent tentacles) and a central layer of mesoglea. (A–J): longitudinal sections. See Table 2 for
abbreviations.

gastrodermis and an internal longitudinal muscle band (epitheliomuscular cells)
embedded in mesoglea (Figs. 26E and 26F). Organization of infundibula (Fig. 26H)
similar to *H. tenuis*. Gastrovascular cavity without claustrum. At base of infundibula,
muscle becomes compressed and flattened, progressively divided into two thin bands
apically (as *Lucernaria sainthilairei*; Figs. 5C and 5D). Organization of manubrium and
gastric radial pockets similar to *H. tenuis* (Fig. 6). Four gastric radial pockets laterally
separated from each other by interradial septa; gastric radial pockets directly connected
only by means of small interradial ostia at margin of calyx (as in other species examined;
Figs. 10M–10U); each gastric radial pocket connected to main gastrovascular cavity.
Organization of gastric filaments (Fig. 7G) similar to those of *H. tenuis*. Vesicles of
gonads not clearly defined (Fig. 25H), with irregular shape: gametes located between a
layer of gastrodermis and epidermis of septa, and this layer can be more or less wavy
(Figs. 27A–27C). Vesicles of same gastric radial pocket formed by gastrodermis of two
different interradial septa (two adjacent septa). Female specimen analyzed, with no
evident regionalization of mature and immature oocytes (Fig. 27), although gametoduct

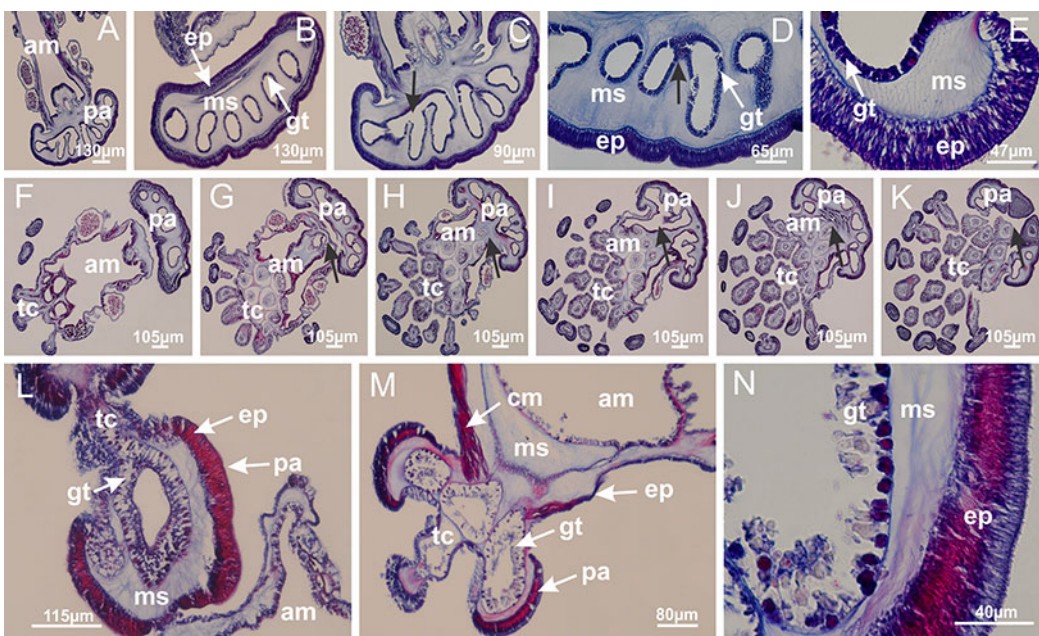

**Figure 24 Pad-like adhesive structures.** *Calvadosia corbini*: (A) general view of pad-like adhesive structure at the tip of arm; (B–E) details of pad-like adhesive structure, with hollow canals (connection of canals indicated by black arrows) delimited by a thin layer of gastrodermis; (F–K) connection of pad-like adhesive structure with the tip of arms (indicated by black arrows); *Manania uchidai*: (L, M) general view of pad-like adhesive structure in outermost secondary tentacle; (N) detail of pad in outermost secondary tentacle, with tall epidermis. (A–K): cross sections; (L–N): longitudinal sections. See Table 2 for abbreviations.

is recognizable (Figs. 27D–27I). Oocytes surrounded by cells of gastrodermal origin (probably follicle cells), leading to gastric radial pockets through fusion of these cells with gastrodermis of the ovarian vesicles (Figs. 27D–27I), forming a gametoduct (Fig. 27I). Cilia often associated with gametoduct (Figs. 27G–27I). Anchors absent (Fig. 25). Arms paired at interradii (Figs. 25A, 25D and 25E), with internal organization similar to *H. tenuis* (Fig. 11). Eight sections of coronal muscle at calyx margin, each between adjacent arms. Organization of longitudinal and coronal muscles in arms similar to *H. tenuis* (as in other species examined; Figs. 5 and 11). Perradial and interradial white spots of nematocysts on subumbrella, with internal organization similar to *H. tenuis* (Figs. 12A–12C). Distal end of arms with intertentacular lobules, a structure between adjacent secondary tentacles delimited by gastrodermis and a central layer of mesoglea (Figs. 17D, 17E and 28). Secondary tentacles of arms without pad-like adhesive structures (Figs. 25D, 25F and 25G). Continuous layer of internal unorganized nematocysts visible in subumbrellar epidermis, from base of infundibula, passing through gastric radial pockets, arms, to tips of secondary tentacles (as in most of species examined; Figs. 15 and 16). Internal layer of nematocysts continuous with groups of nematocysts at tentacular base, in epidermis, with different sizes and types, also unorganized (Figs. 16D–16H). Secondary hollow tentacles composed of two parts, knob and stem, with organization similar to *H. tenuis* (as in other species examined; Fig. 17). At stem base, secondary

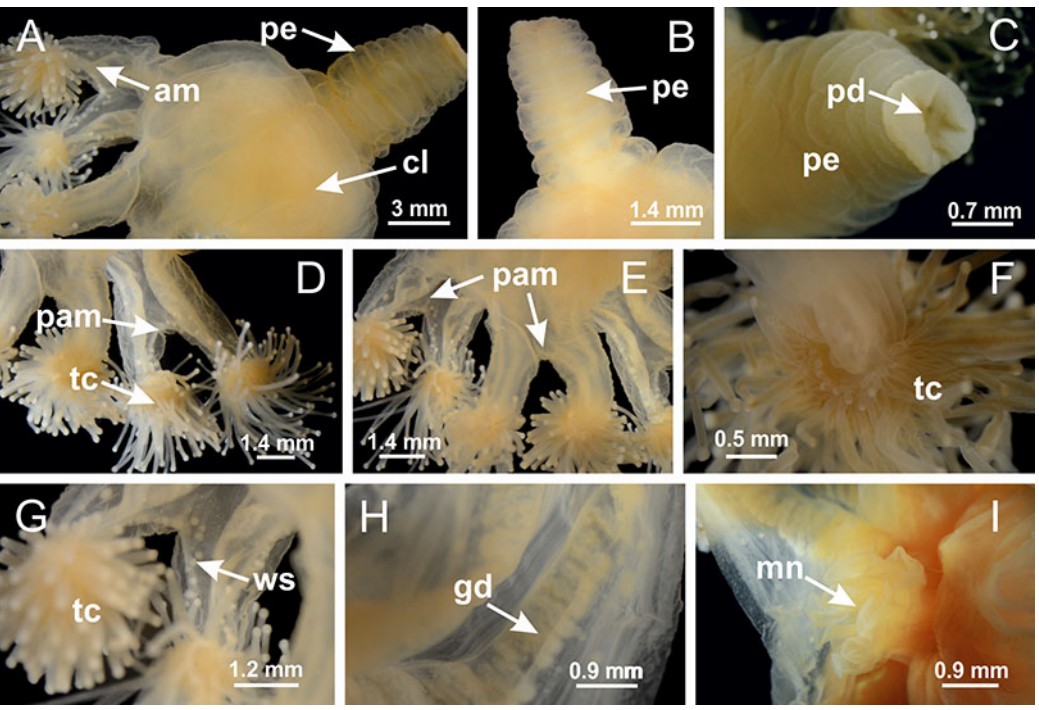

**Figure 25 General view of *Lucernaria quadricornis*.** (A) General view of body, with calyx and peduncle; (B, C) detail of peduncle and pedal disk; (D, E) paired arms and tentacular clusters; (F) detail of tentacular clusters; (G) white spots of nematocysts at margin of calyx (subumbrella); (H) subumbrellar view of gonads; (I) manubrium. See Table 2 for abbreviations.

tentacles tightly joined, separated only by thin layer of mesoglea, with beehive appearance in cross section (Fig. 17F).

*Lucernaria bathyphila Haeckel, 1879* (Figs. 17G, 17H; 29; 30; 31; 32N–32W; 33; 57)

Internal anatomy similar to *Lucernaria quadricornis*. Peduncle with one chamber (delimited by gastrodermis), and four interradial gastric septa: each septum consisting of mesoglea surrounded by gastrodermis and an interradial longitudinal muscle band (epitheliomuscular cells) embedded in mesoglea (Fig. 30). Gastrovascular cavity without claustrum. Four gastric radial pockets laterally separated from each other by interradial septa; gastric radial pockets directly connected only by means of small interradial ostia at margin of calyx (as in other species examined; Figs. 10M–10U); each gastric radial pocket connected to main gastrovascular cavity. Organization of gastric filaments similar to those of *H. tenuis* (as in other species examined, Fig. 7). Vesicles of gonads not clearly defined (Figs. 29G, 29H, 31A–31C), with irregular shape: gametes located between a layer of gastrodermis and epidermis of septa, and this layer can be more or less wavy (Figs. 31A–31C). Vesicles of same gastric radial pocket formed by gastrodermis of two different interradial septa (two adjacent septa). Specimen analyzed probably an immature male and spermatozoa could not be distinguished (Fig. 31). Gametoduct clearly recognizable: cells of gastrodermal origin connected to spermatocytes, leading to gastric radial pockets through fusion of these cells with gastrodermis of vesicles (Figs. 31F–31Q).

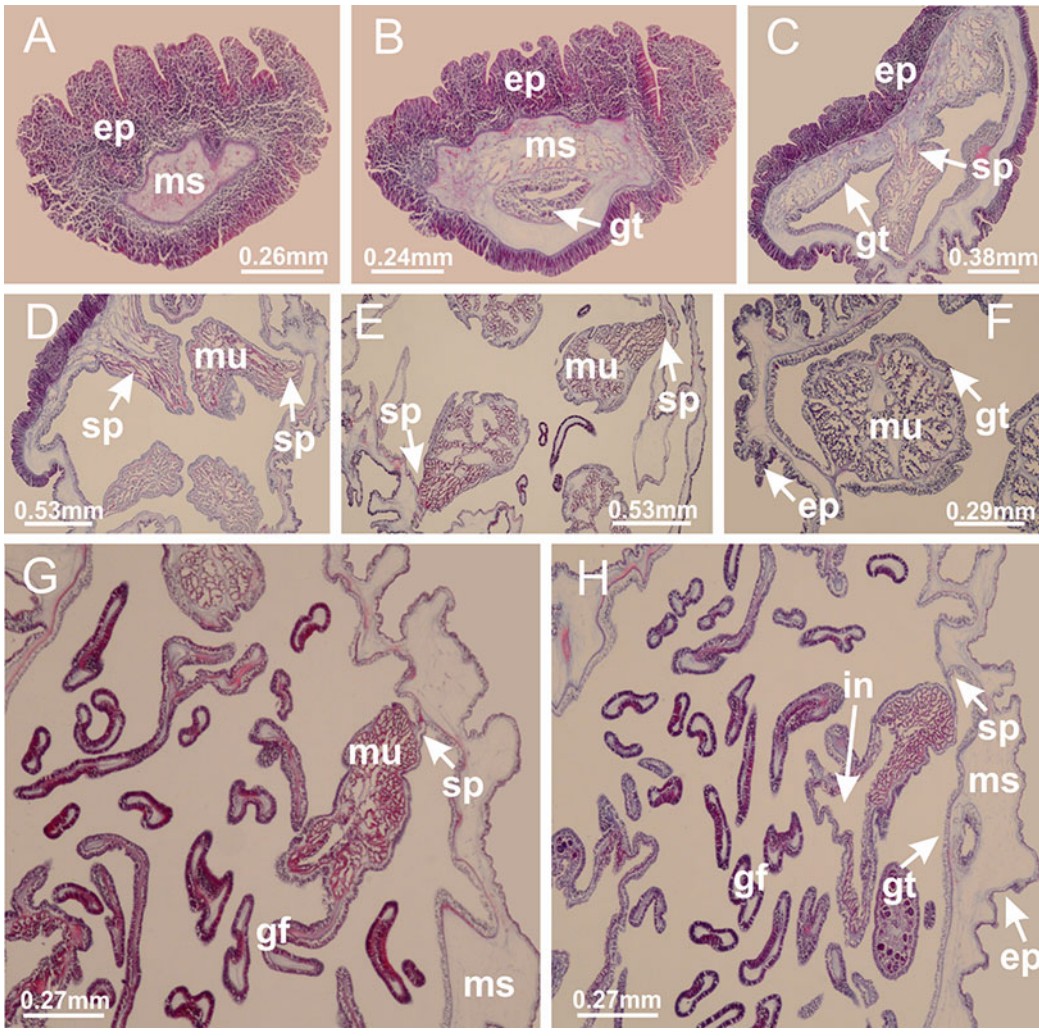

**Figure 26 Peduncle and septa of *Lucernaria quadricornis* (from base moving upward in A–H).** (A, B) Base of peduncle; (C, D) delimitation of septa; (E) interradial septa, with interradial longitudinal muscle and one central chamber in peduncle; (F) detail of septum; (G) formation of gastric filaments through lateral evagination of septal gastrodemis; (H) septum at peduncle/calyx connection, with infundibulum (delimited by epidermis). (A–H): cross sections. See Table 2 for abbreviations.

Cilia often associated with gametoduct (Fig. 31Q). Anchors absent (Fig. 29E). Eight sections of coronal muscle at calyx margin, each between adjacent arms. Organization of longitudinal and coronal muscles in arms similar to *H. tenuis* (as in other species examined; Figs. 5 and 11). Perradial and interradial white spots of nematocysts on subumbrella (Fig. 29H), with internal organization similar to *H. tenuis* (Figs. 12A–12C). Aperture of white spots at subumbrellar epidermis clearly recognizable in its central thicker region (Figs. 32N–32W): a pore divides subumbrellar epidermis and adjacent layer of mesoglea into two regions in a longitudinal section, allowing an outflow to central mature nematocysts (Figs. 32Q–32W). Batteries of nematocysts sparsely distributed in exumbrellar epidermis (Fig. 29C), similar in internal organization to other species examined (Fig. 13). Distal end of arms with intertentacular lobules (Fig. 33). Secondary tentacles of arms (Figs. 29E, 29F and 29H) without pad-like adhesive structures.

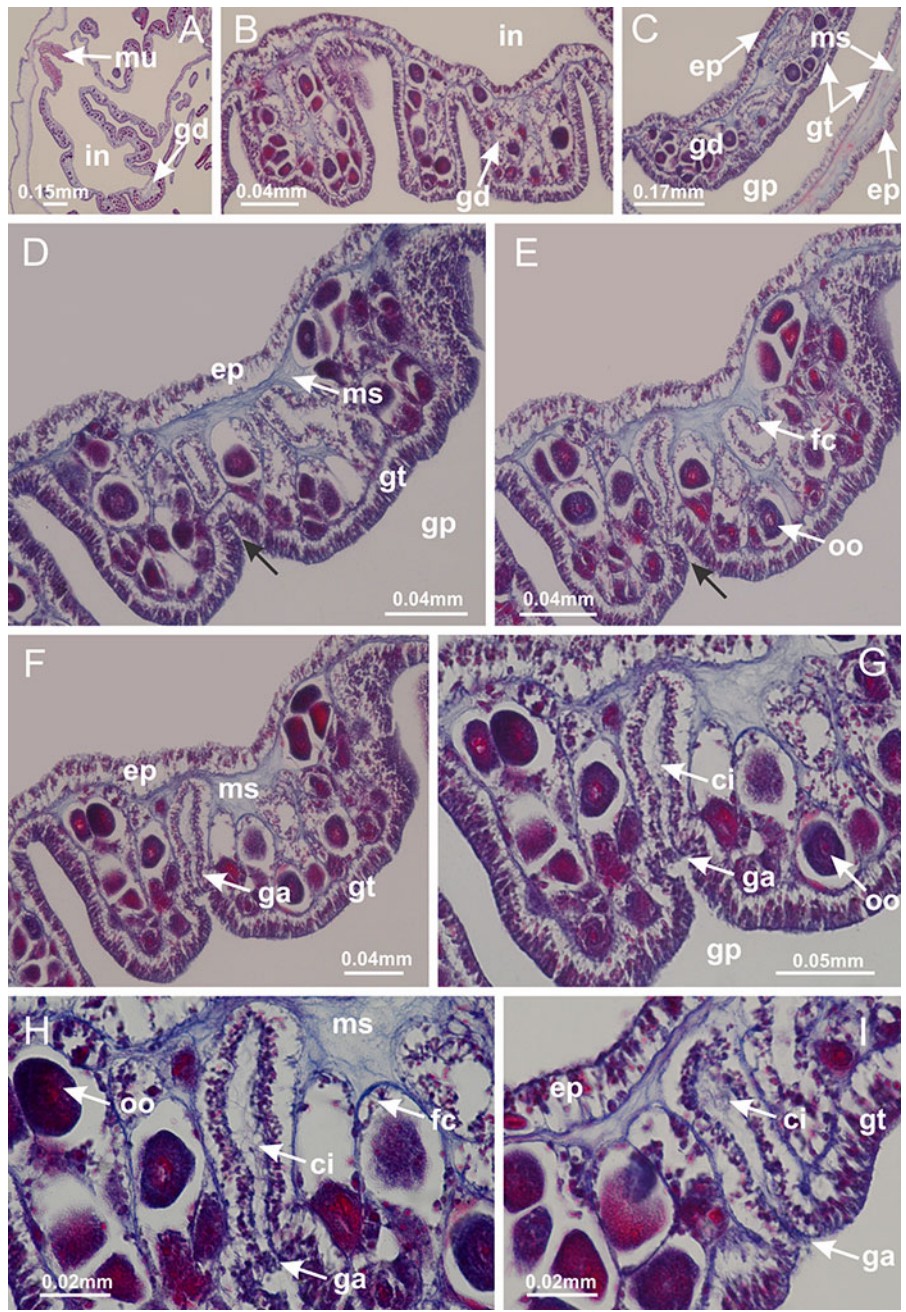

**Figure 27 Gonads and gametoduct of *Lucernaria quadricornis*.** (A) General view of gonad associated with septum; (B, C) female gonadal content, delimited by a layer of gastrodermis and epidermis; (D–G) sequence of gametoduct connecting oocytes with the gastrovascular cavity (indicated by black arrows); (H, I) detail of gametoduct, with cilia. (A–I): cross sections. See Table 2 for abbreviations.

Continuous layer of internal unorganized nematocysts in subumbrellar epidermis, from base of infundibula, passing through gastric radial pockets, arms, to tips of secondary tentacles (as in most of species examined; Figs. 15 and 16). Internal layer of nematocysts continuous with groups of nematocysts at tentacular base, in epidermis, with different sizes and types, also unorganized. Organization of secondary tentacles (Figs. 17G

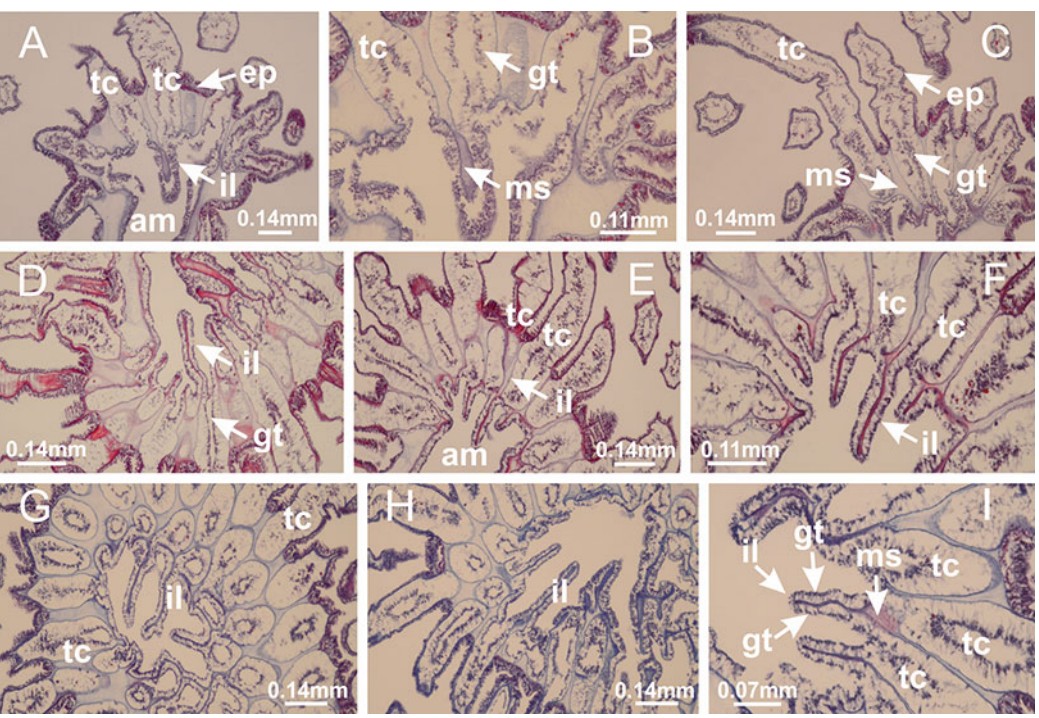

**Figure 28 Intertentacular lobules of *Lucernaria quadricornis*.** (A) General view of tip of arms and base of tentacles, with intertentacular lobules; (B) internal base of tentacles; (C) general view of tip of arms, and base of tentacles; (D–H) tip of arms, in the region between intertentacular lobules and secondary tentacles; (I) detail of intertentacular lobules, structures composed of a double layer of gastrodermis (of adjacent tentacles) and a central layer of mesoglea. (A–I): longitudinal sections. See Table 2 for abbreviations.

and 17H) similar to *H. tenuis*. At stem base, secondary tentacles tightly joined, separated only by thin layer of mesoglea, with beehive appearance in cross section (as in *L. quadricornis*; Fig. 17F).

*Lucernaria sainthilairei* (*Redikorzev, 1925*) (Figs. 5C, 5D; 12I; 16J–16L; 17I, 17J; 34; 35; 36; 37; 57)

Internal anatomy similar to *Lucernaria quadricornis*. Peduncle with one chamber (delimited by gastrodermis), and four interradial gastric septa: each septum consisting of mesoglea surrounded by gastrodermis and an internal longitudinal muscle band (epitheliomuscular cells) embedded in mesoglea (Fig. 35). Gastrovascular cavity without claustrum. Four gastric radial pockets laterally separated from each other by interradial septa; gastric radial pockets directly connected only by means of small interradial ostia at margin of calyx (as in other species examined; Figs. 10M–10U); each gastric radial pocket connected to main gastrovascular cavity. Organization of gastric filaments similar to those of *H. tenuis* (as in other species examined, Fig. 7). Vesicles of gonads not clearly defined, with irregular shape (Fig. 36): gametes located between a layer of gastrodermis and epidermis of septa, and this layer can be more or less wavy (Figs. 36A–36D). Vesicles of same gastric radial pocket formed by gastrodermis of two different interradial septa (two adjacent septa). Specimen analyzed probably an immature male; spermatozoa and

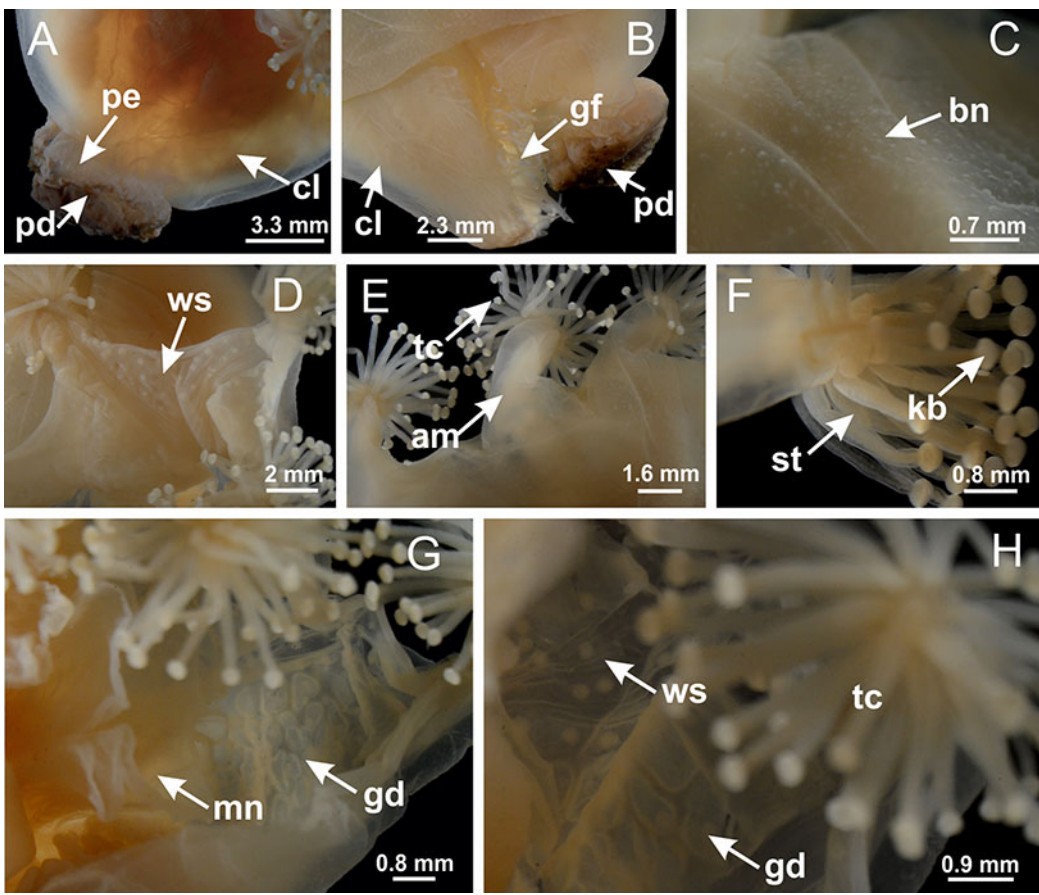

**Figure 29** **General view of *Lucernaria bathyphila*.** (A, B) General view of peduncle and base of calyx; (C) batteries of nematocysts in exumbrella; (D) white spots of nematocysts on subumbrella (calyx); (E) arms and tentacular clusters; (F) detail of tentacles; (G) manubrium and gonads; (H) detail of white spots and gonads. See Table 2 for abbreviations.

gametoduct could not be recognized (Fig. 36). Anchors absent (Figs. 34A, 34B and 34F). Eight sections of coronal muscle at calyx margin, each between adjacent arms. Organization of longitudinal and coronal muscles in arms similar to *H. tenuis* (as in other species examined; Figs. 5 and 11). Perradial and interradial white spots of nematocysts on subumbrella (Figs. 34B and 34F), with internal organization similar to *H. tenuis* (Fig. 12I). Batteries of nematocysts sparsely distributed in exumbrellar epidermis (Fig. 34D), similar in internal organization to other species examined (Fig. 13). Distal end of arms with intertentacular lobules (Fig. 37). Secondary tentacles of arms without pad-like adhesive structures (Fig. 34). Continuous layer of internal unorganized nematocysts in subumbrellar epidermis, from base of infundibula, passing through gastric radial pockets, arms, to tips of secondary tentacles (as in most of species examined; Figs. 15 and 16). Internal layer of nematocysts continuous with groups of nematocysts at tentacular base, in epidermis, with different sizes and types, also unorganized (Figs. 16J–16L). Organization of secondary tentacles (Figs. 17I and 17J) similar to *H. tenuis*. At stem base, secondary tentacles tightly joined, separated only by thin layer of mesoglea, with beehive appearance in cross section (as in *L. quadricornis*; Fig. 17F).

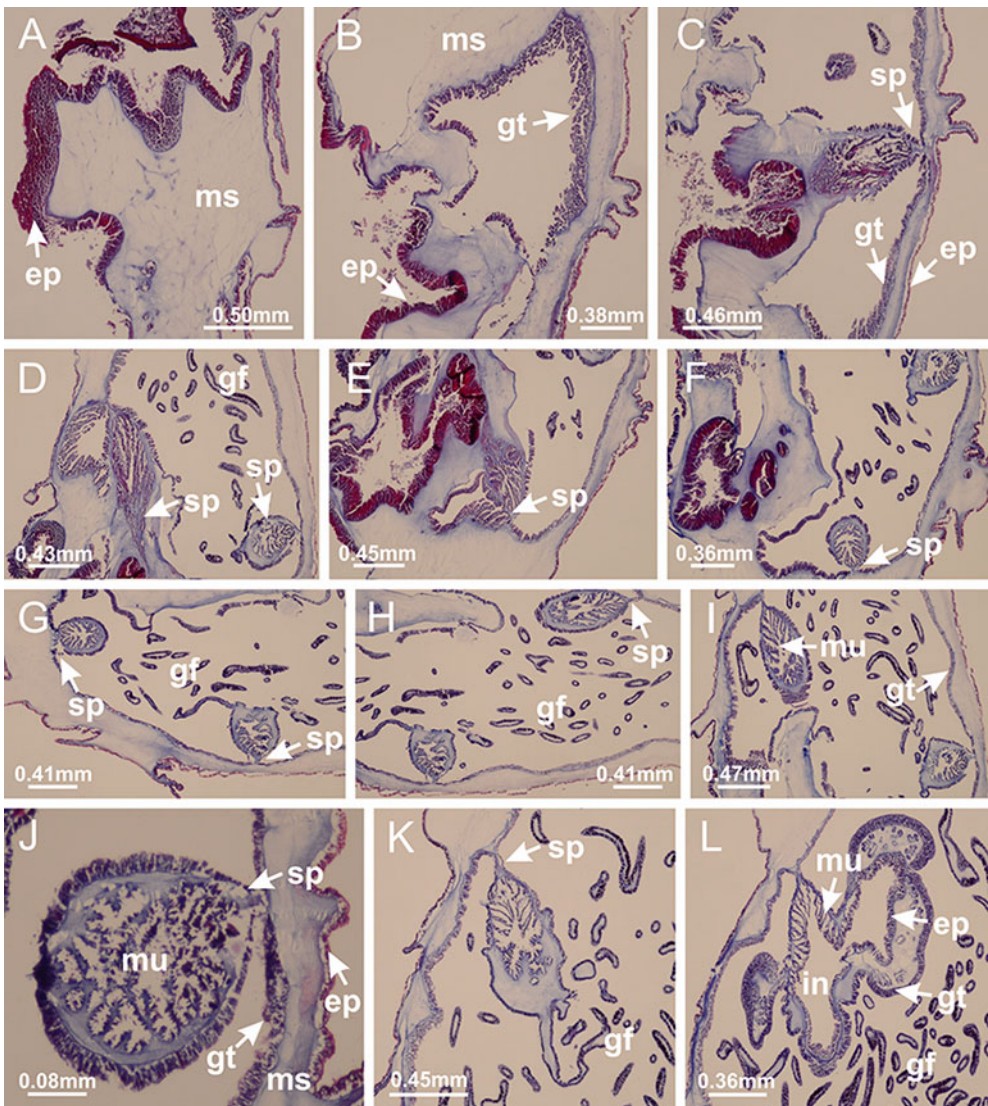

**Figure 30 Peduncle and septa of *Lucernaria bathyphila* (from base moving upward in A–L).** (A, B) Base of peduncle; (C–F) delimitation of septa; (G–I) interradial septa, with interradial longitudinal muscle, and one central chamber in peduncle; (J) detail of septum; (K) formation of gastric filaments through lateral evagination of septal gastrodermis; (L) septum at the connection between calyx and peduncle, with lateral gonads, and central infundibulum (delimited by epidermis). (A–L): cross sections. See Table 2 for abbreviations.

## Suborder Amyostaurida *Miranda, Hirano, Mills, Falconer, Fenwick, Marques & Collins, 2016*

Family Kishinouyeidae *Uchida, 1929*

Genus *Calvadosia Clark, 1863*

*Calvadosia corbini* (*Larson, 1980*) (Figs. 5I–5L, 5O, 5P, 5T, 5U; 7A–7F; 10P–10U; 12J–12M; 13H–13M; 15R, 15S; 16I; 17A–17C; 24A–24K; 38; 39; 40; 41; 57)

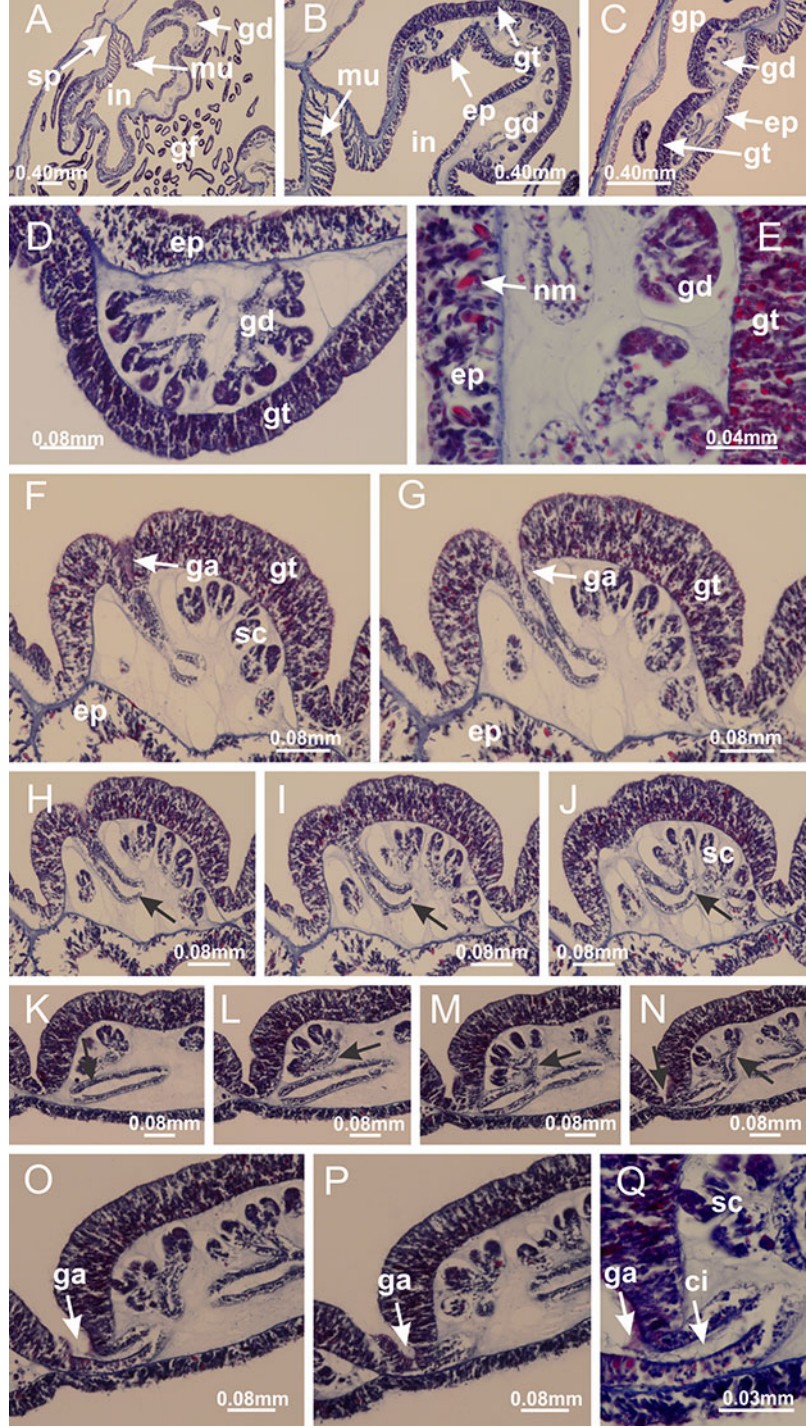

**Figure 31 Gonads and gametoduct of *Lucernaria bathyphila*.** (A) General view of gonad associated with septum; (B) male gonadal content, delimited by a layer of gastrodermis and epidermis; (C) gonads associated with gastric radial pockets; (D, E) internal organization of gonads; (F–J) sequence of gametoduct connecting the spermatocytes with the gastrovascular cavity (indicated by black arrows); (K–P) sequence of gametoduct connecting the spermatocytes with the gastrovascular cavity (indicated by black arrows); (Q) detail of gametoduct, with cilia. (A–E): cross sections; (F–Q): longitudinal sections of gonad (cross sections of animals). See Table 2 for abbreviations.

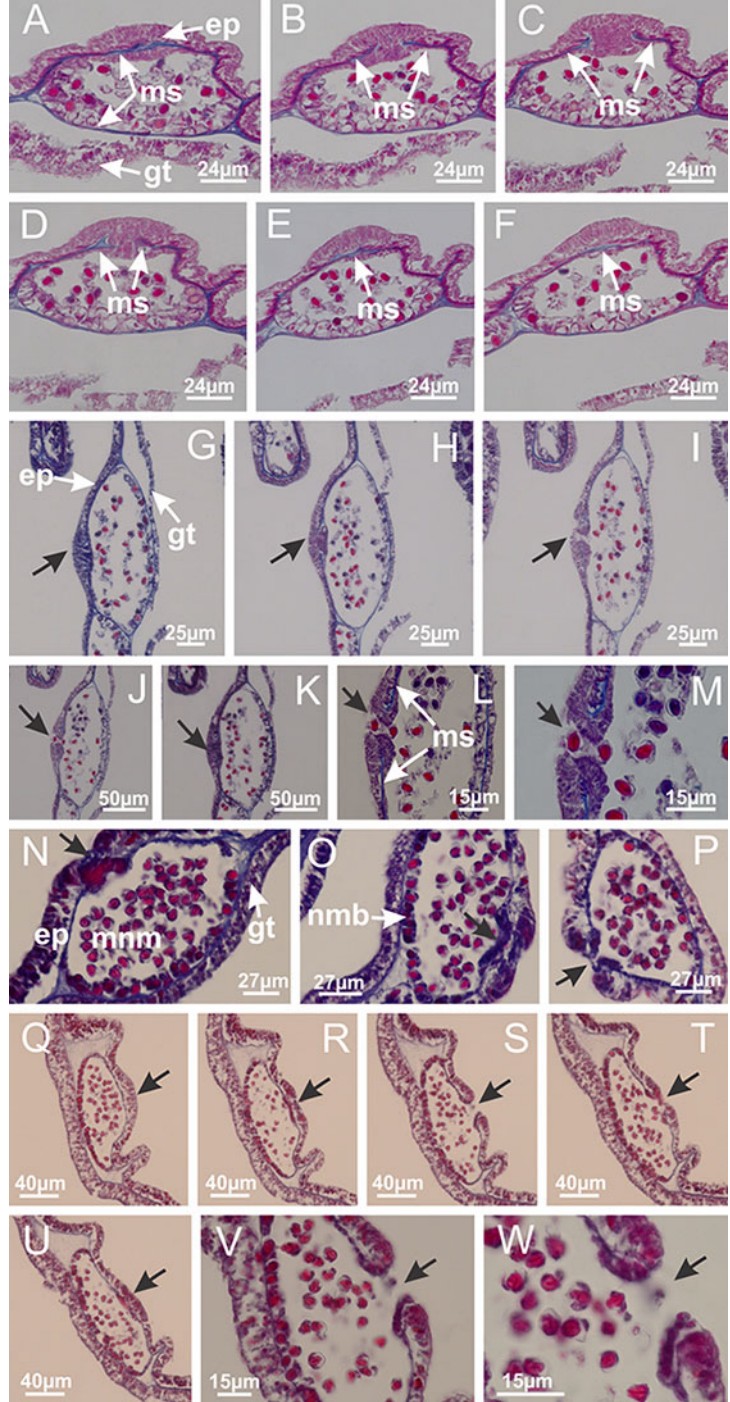

**Figure 32 White spots of nematocysts.** *Craterolophus convolvulus*: (A–F) gradual disconnection and posterior reconnection of a layer of mesoglea associated with the central thickening of epidermis; (G–L) opening of white spots (indicated by black arrows), through a central pore, associated with thickening in epidermis; (M) detail of central opening of white spots (indicated by black arrow); *Lucernaria bathyphila*: (N–P) central thickening of epidermis of white spots (indicated by black arrows); (Q–V) opening of white spots of nematocysts (indicated by black arrows), through a central pore, associated with thickening in epidermis; (W) detail of central opening of white spots (indicated by black arrow). (A–W): longitudinal sections of white spots of nematocysts (cross sections of animals). See Table 2 for abbreviations.

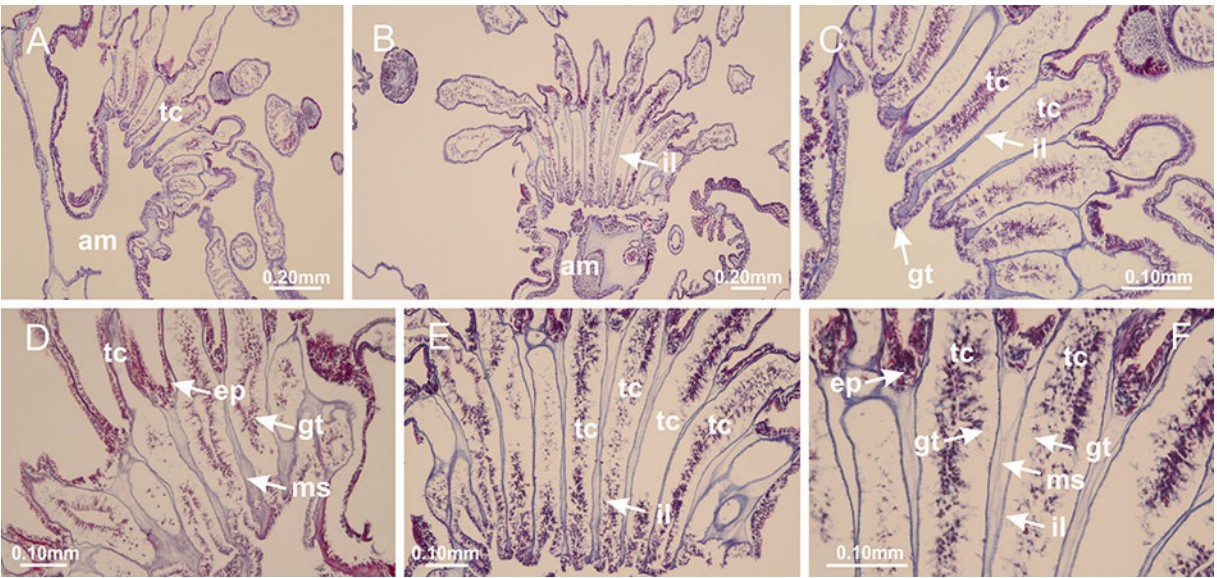

**Figure 33 Intertentacular lobules of *Lucernaria bathyphila*.** (A, B) General view of tip of arms and base of tentacles, with intertentacular lobules; (C–E) internal organization of intertentacular lobules, structures composed of a double layer of gastrodermis (of adjacent tentacles) and a central layer of mesoglea; (F) detail of intertentacular lobules. (A–F): longitudinal sections. See Table 2 for abbreviations.

Peduncle with a broad basal pedal disk (Figs. 38A–38C). Base of peduncle with four perradial chambers (delimited by gastrodermis), with numerous evaginations (Figs. 39C–39H). Gradual connection of four perradial chambers, defining one central gastric chamber, and delimiting four interradial gastric septa toward apical region (Figs. 39E, 39F and 39I). Peduncle without interradial longitudinal muscles (Figs. 39I–39M). Size of septa decreases at peduncle/calyx connection (Figs. 39N–39Q), where interradial longitudinal muscles (epitheliomuscular cells) inside septa become visible (Figs. 39R and 39S). Internal organization of infundibula (Figs. 39X and 39Y) similar to *H. tenuis*. Gastrovascular cavity without claustrum. At base of infundibulum, interradial longitudinal muscle is compressed, and then becomes divided into two bands (Figs. 5I–5L). Internal organization of gastric radial pockets and manubrium similar to *H. tenuis* (Fig. 6). Four gastric radial pockets laterally separated from each other by interradial septa; gastric radial pockets directly connected only by means of small interradial ostia at margin of calyx (Figs. 10P–10U); each gastric radial pocket connected to main gastrovascular cavity. Manubrium internally defined by gastrodermis, externally by epidermis. Gastric filaments (Figs. 7A–7F) similar to those of *H. tenuis*. Gonads with numerous vesicles, irregularly arranged in asymmetrical erected nodules, formed by an external fold of subumbrellar tissue (Figs. 38E–38I, 40A–40E). Male specimen analyzed, with spermatocytes adjacent to gastrodermis, in peripheral position; spermatozoa adjacent to epidermis, in central position (Fig. 40H). Spermatozoa divided into different sacs, delimited by cells of gastrodermal origin, connected to gastrodermis of gastric radial pocket, forming gametoduct (Figs. 40J, 40L–40P). Large batteries of nematocysts in subumbrellar epidermis, associated with gonads, between internal vesicles (Figs. 13L, 13M, 40H and 40I). Anchors absent (Figs. 38A, 38B and 38J).

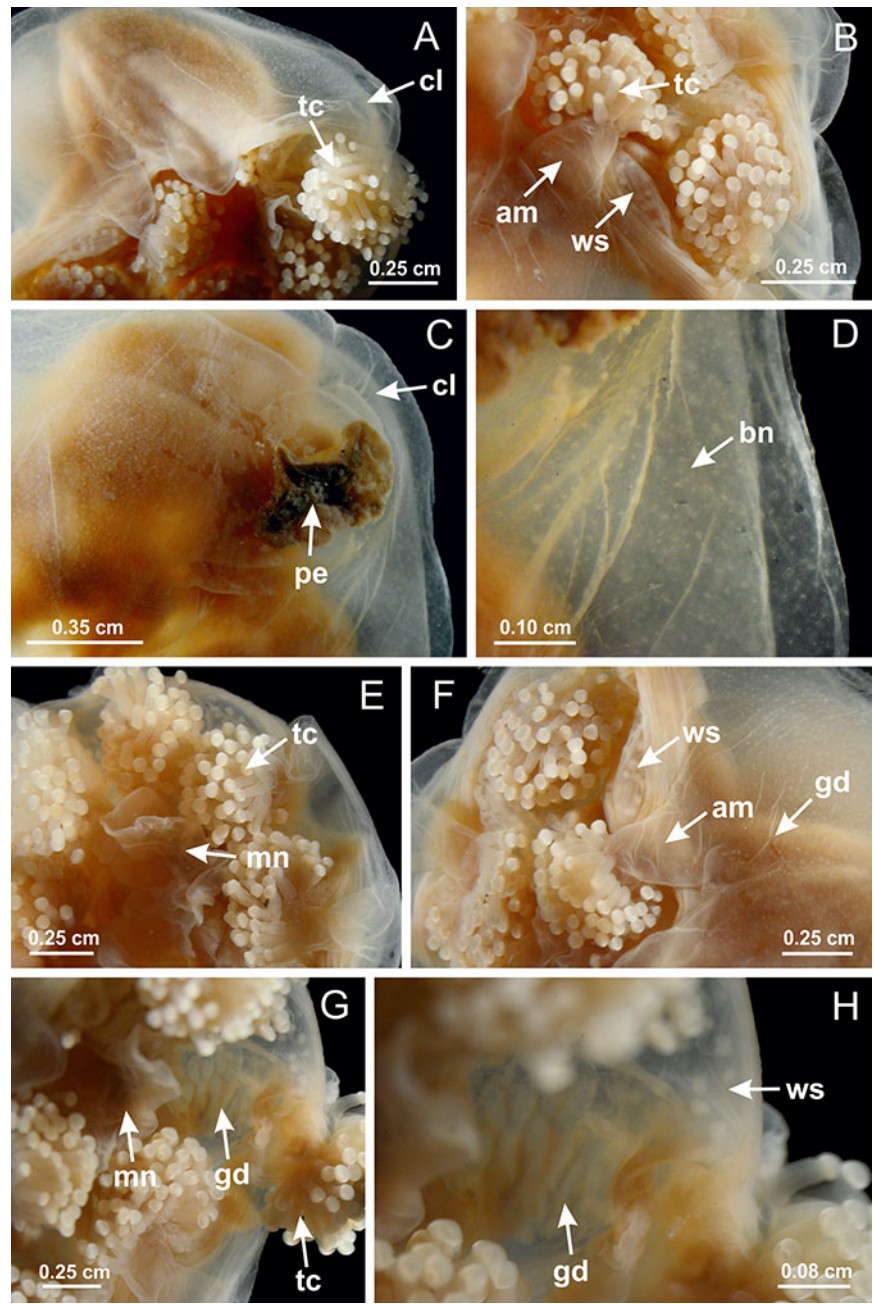

**Figure 34 General view of *Lucernaria sainthilairei*.** (A) General view of calyx; (B) apical region of calyx, with arms and tentacular clusters; (C) short peduncle; (D) batteries of nematocysts in the exumbrella; (E) general view of subumbrella of calyx, with manubrium; (F) exumbrellar view of gonad and arms; (G) manubrium and gonads; (H) detail of gonad and white spots of nematocysts. See Table 2 for abbreviations.

Arms sharply paired at interradii (perradial notches deeper than interradial notches) (Figs. 38A, 38B, 38I and 38J), with internal organization similar to *H. tenuis* (Fig. 11). Eight sections of coronal muscle at calyx margin (Fig. 38D), each between adjacent arms. Organization of longitudinal and coronal muscles in arms similar to *H. tenuis* (as in other species examined; Figs. 5 and 11). Perradial and interradial white spots of nematocysts

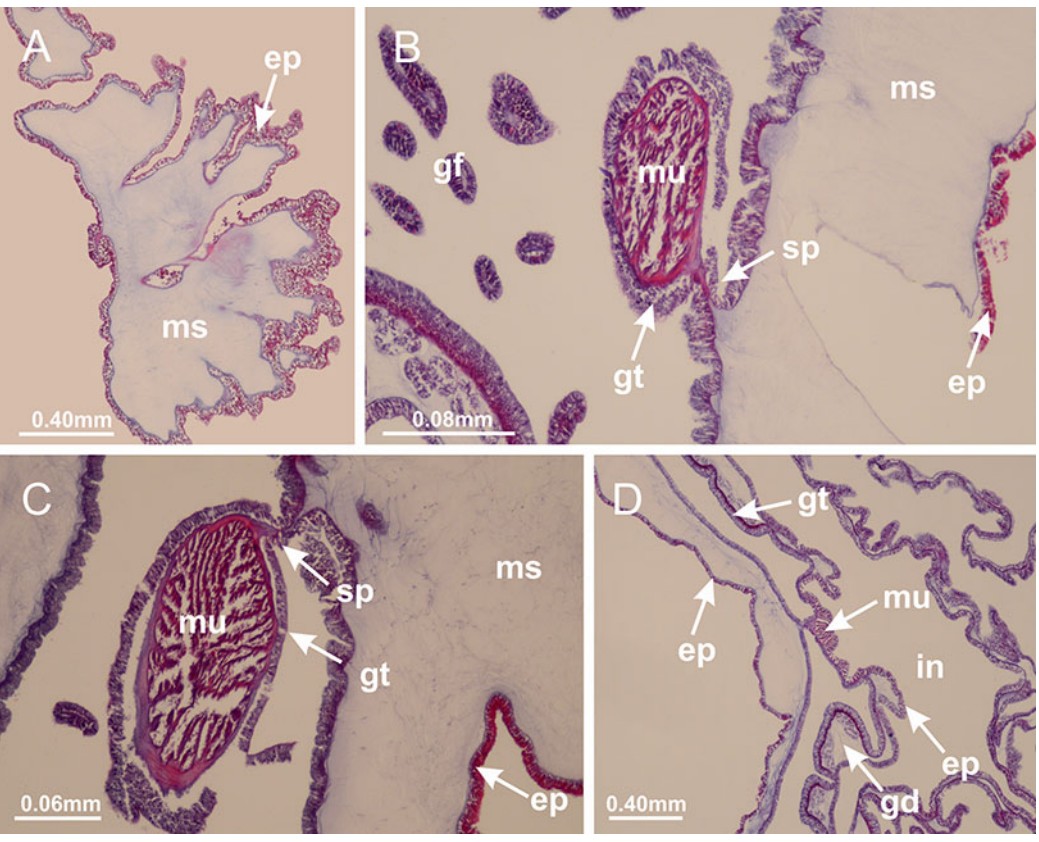

**Figure 35 Peduncle and septa of *Lucernaria sainthilairei* (from base moving upward in A–D).** (A) Base of peduncle; (B, C) interradial septa, with interradial longitudinal muscles, and one central chamber in peduncle; (D) septum at the peduncle/calyx connection, with infundibulum (delimited by epidermis). (A–D): cross sections. See Table 2 for abbreviations.

on subumbrella (Fig. 38G), with internal organization similar to *H. tenuis* (Figs. 12J–12M). Batteries of nematocysts sparsely distributed in exumbrellar epidermis (Figs. 13H–13K). Distal end of arms with intertentacular lobules, a structure between adjacent secondary tentacles delimited by gastrodermis and a central layer of mesoglea (Fig. 41). Tip of each arm with large pad-like adhesive structures (Figs. 38K and 38L), with thick epidermis, mesoglea, and hollow canals delimited by thin layer of gastrodermis; hollow gastrodermal canals gradually connected to gastrovascular cavity at tip of arms (Figs. 24A–24K). Continuous layer of internal unorganized nematocysts in subumbrellar epidermis (as in most of species examined; Figs. 15 and 16). Secondary hollow tentacles composed of two parts, knob and stem, with organization (Figs. 17A–17C) similar to *H. tenuis*. At stem base, secondary tentacles tightly joined, separated only by thin layer of mesoglea, with beehive appearance in cross section as in most of species examined (Fig. 17F).

*Calvadosia cruciformis* (*Okubo, 1917*) (Figs. 2G, 2H; 9J–9S; 10G–10L; 12G, 12H; 13D; 42; 43; 44; 45; 57)

   Basal pedal disk of peduncle (Fig. 42A) with epidermal axial canal, a pronounced and delimited invagination with blind end (Figs. 42I, 43A–43D). Base of peduncle with

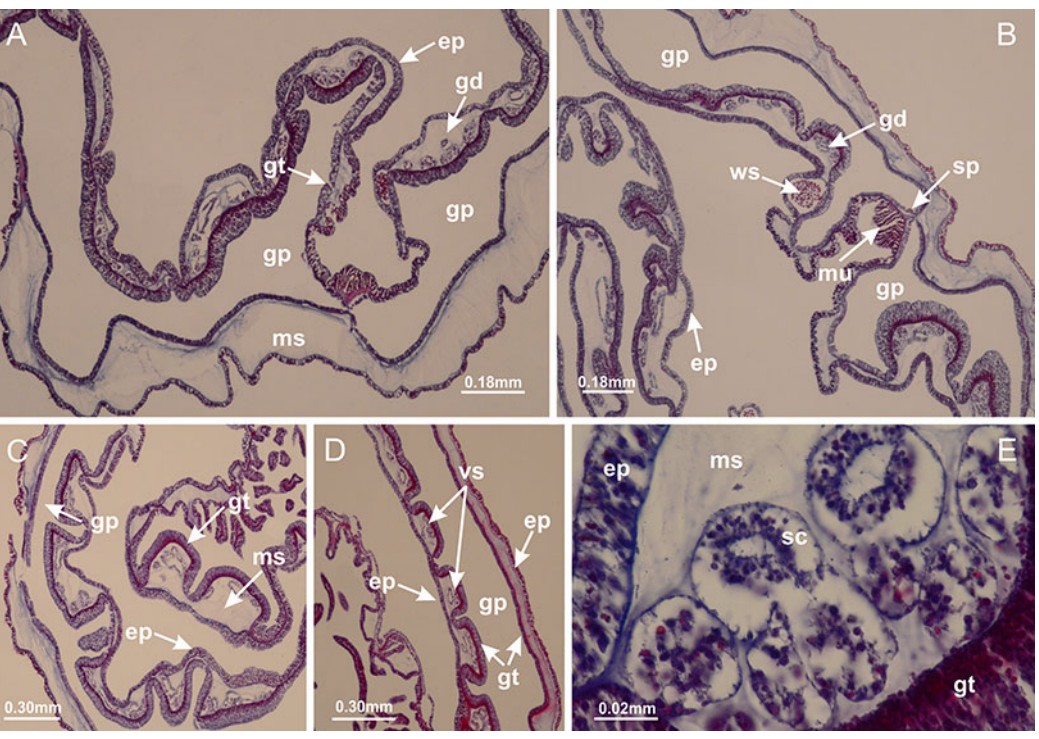

**Figure 36 Gonads of *Lucernaria sainthilairei*.** (A) General view of gonad delimited between a layer of gastrodermis and epidermis of septum; (B–D) gonads associated with gastric radial pockets; (E) detail of immature male gonadal content, with spermatocytes. (A–E): cross sections. See Table 2 for abbreviations.

four gastric perradial chambers (delimited by gastrodermis) (Figs. 43A–43G). Gradual connection of four perradial chambers, defining one central gastric chamber, and delimiting four interradial gastric septa toward apical region (Figs. 43H–43L). Peduncle without interradial longitudinal muscles (Figs. 43A–43M). Size of septa decreases at peduncle/calyx connection, and interradial longitudinal muscles (epitheliomuscular cells) inside septa become visible (Figs. 43N–43P). Internal organization of infundibula (Figs. 43N–43P) similar to *H. tenuis*. Gastrovascular cavity without claustrum. At base of infundibulum, interradial longitudinal muscle is compressed, and then becomes divided into two bands as in *C. corbini* (Figs. 5I–5L). Internal organization of gastric radial pockets and manubrium similar to *H. tenuis* (Fig. 6). Four gastric radial pockets laterally separated from each other by interradial septa; gastric radial pockets directly connected only by means of small interradial ostia at margin of calyx (Figs. 10G–10L); each gastric radial pocket connected to main gastrovascular cavity. Manubrium internally defined by gastrodermis, externally by epidermis. Gastric filaments similar to those of other species examined (Fig. 7). Gonads with vesicles, which are serial gastrodermal evaginations at lateral regions of interradial septa, gastric radial pockets, and arms (Figs. 44A–44C); vesicles of same gastric radial pocket formed by gastrodermis of two different interradial septa (two adjacent septa). Female specimen analyzed (Fig. 44), with ovarian vesicles with two main layers: peripheral layer with immature oocytes in different developmental stages, internal layer with mature oocytes with scattered yolk granules

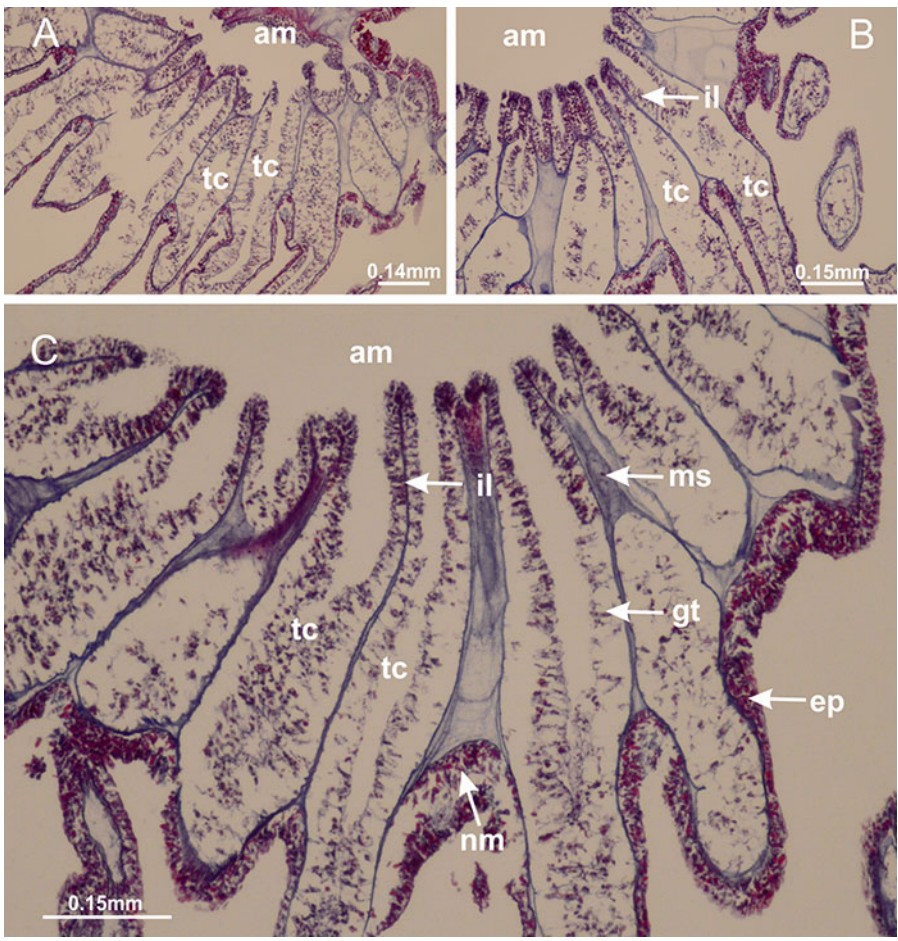

**Figure 37 Intertentacular lobules of *Lucernaria sainthilairei*.** (A, B) Tip of arms, in the region between intertentacular lobules and secondary tentacles; (C) detail of intertentacular lobules, structures composed of a double layer of gastrodermis (of adjacent tentacles) and a central layer of mesoglea. (A–C): longitudinal sections. See Table 2 for abbreviations.

(Figs. 44B, 44F–44L). Mature oocytes surrounded by cells of gastrodermal origin (probably follicle cells), which are connected to gastrodermis of gastric radial pocket, forming gametoduct (Figs. 44D–44L). Perradial and interradial primary tentacles present, presenting curved knob with nematocysts, and a small white disk at stem (Figs. 9J–9S, 10G–10L, 42D and 42E). Interradial ostia connecting interradial primary tentacles with gastrovascular cavity (Figs. 10G–10L). Arms sharply paired at interradii (Figs. 42A and 42B), with internal organization similar to *H. tenuis* (Fig. 11). Eight sections of coronal muscle at calyx margin, each between adjacent arms. Organization of longitudinal and coronal muscles in arms similar to *H. tenuis* (as in other species examined; Figs. 5 and 11). Perradial and interradial white spots of nematocysts on subumbrella (Figs. 12G, 12H, 42B and 42C), with internal organization similar to *H. tenuis*. Batteries of nematocysts sparsely distributed in exumbrellar epidermis (Fig. 13D). Distal end of arms with intertentacular lobules, a structure between adjacent secondary tentacles delimited by gastrodermis and a central layer of mesoglea (Fig. 45). Outermost secondary

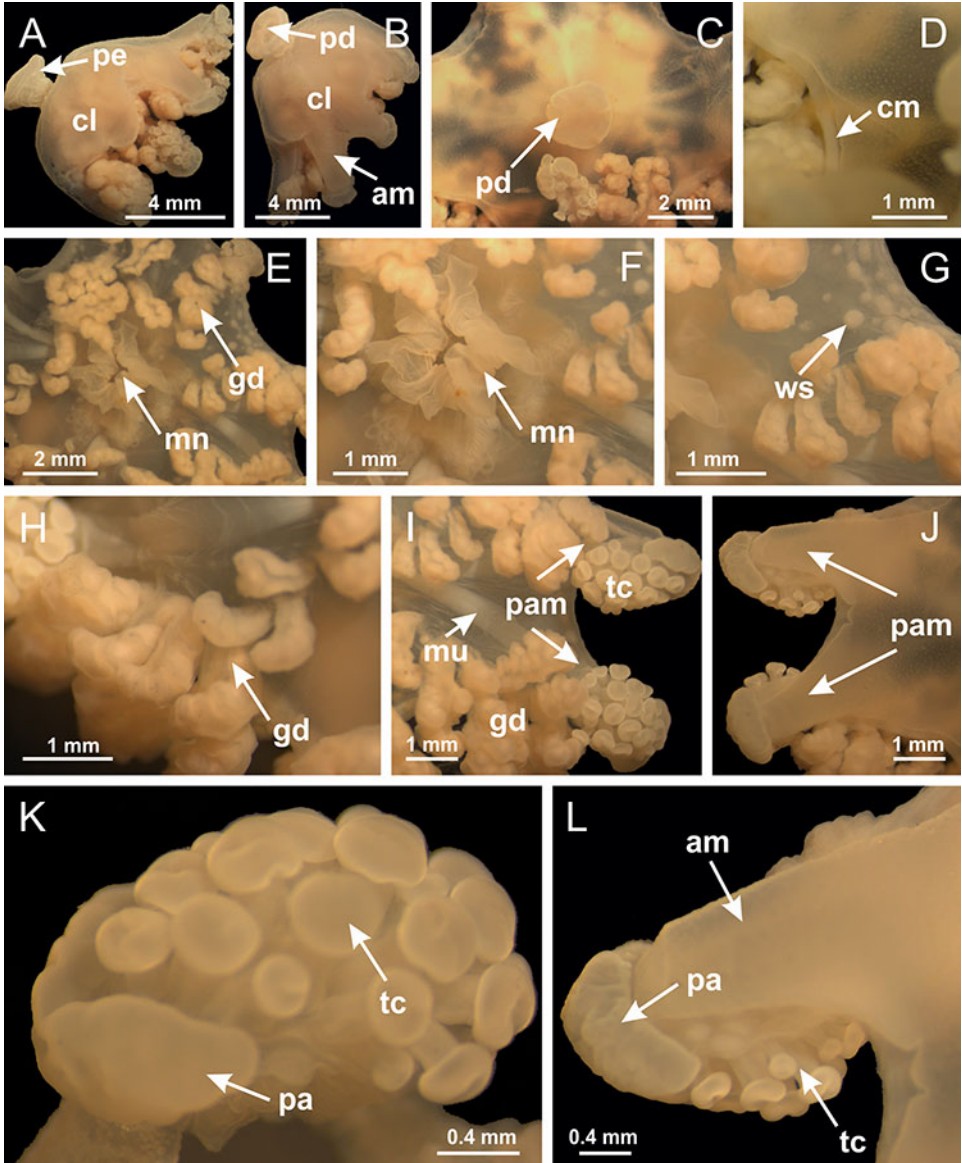

**Figure 38 General view of *Calvadosia corbini*.** (A, B) General view of body, with calyx and peduncle; (C) pedal disk of peduncle; (D) marginal coronal muscle; (E) subumbrellar view of calyx, with central manubrium and gonads; (F) detail of manubrium; (G) white spots of nematocysts at calyx margin; (H) detail of nodular gonads; (I) subumbrellar view of paired arms; (J) exumbrellar view of paired arms; (K, L) pad-like adhesive structures at the tip of arms, and tentacular clusters. See Table 2 for abbreviations.

tentacles with pad-like adhesive structures (epidermal thickening) (as in *M. uchidai*; Figs. 24L–24N and 42H). Clearly recognizable continuous layer of internal unorganized nematocysts in subumbrellar epidermis, from base of infundibula to tips of secondary tentacles, as in most of species examined (Figs. 15 and 16). Secondary hollow tentacles composed of two parts, knob and stem, with organization similar to *H. tenuis* (as in other species examined; Figs. 17 and 45).

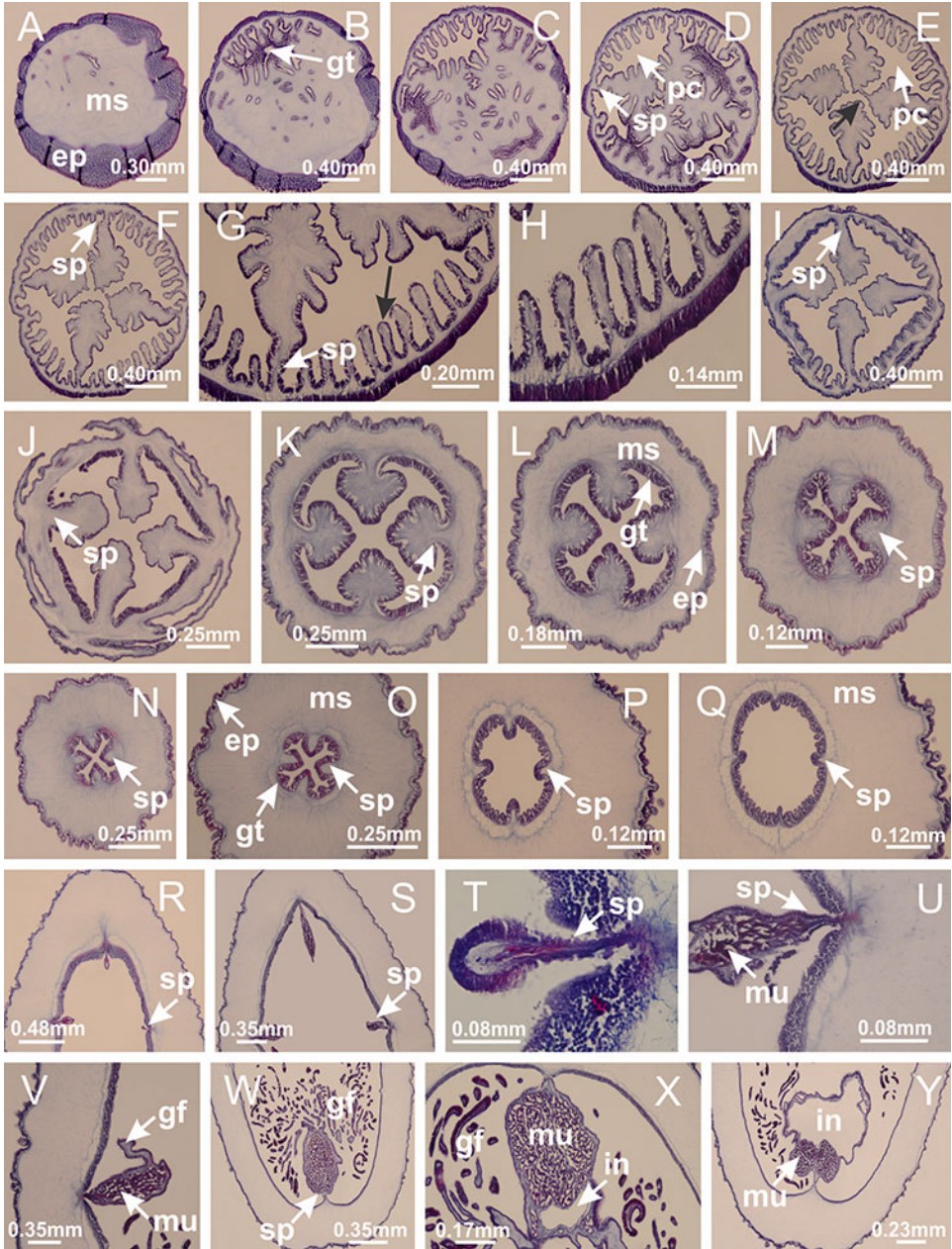

**Figure 39 Peduncle and septa of *Calvadosia corbini* (from base moving upward in A–Y).** (A) Base of peduncle; (B–D) delimitation of four perradial chambers; (E, F) fusion of four perradial chamber (indicated by black arrow), and delimitation of the four interradial septa; (G) detail of interradial septum, without interradial longitudinal muscle; and evaginations in the gastrodermis of chamber; (H) detail of evaginations in the gastrodermis; (I) central cruciform chamber, delimited by gastrodermis, and four interradial septa, without interradial longitudinal muscle; (J–Q) modification in shape and size of interradial septa and chamber; (R–T) septa and chamber at the peduncle/calyx connection; (U) interradial septa at calyx base, with interradial longitudinal muscle; (V, W) gastric filaments as lateral evaginations of septal gastrodermis; (X, Y) septa with infundibula delimited by epidermis, at calyx base. (A–Y): cross sections. See Table 2 for abbreviations.

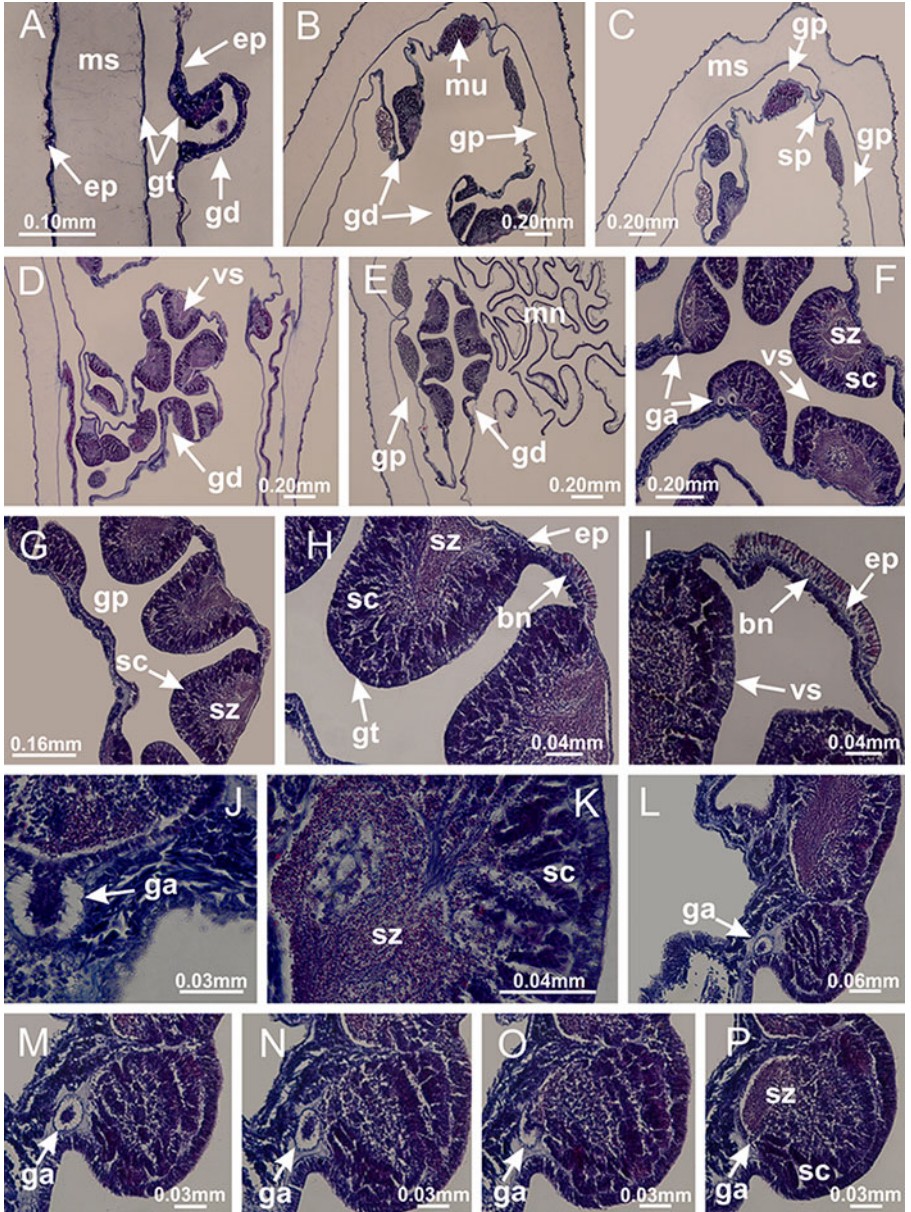

**Figure 40 Gonads and gametoduct of *Calvadosia corbini*.** (A–E) Internal organization of nodular gonads, composed of many vesicles, as an evagination of gastric radial pocket; (F–H) internal organization of male vesicles, with spermatocytes adjacent to gastrodermis, and spermatozoa adjacent to epidermis; (I) battery of nematocysts in the epidermis (subumbrella) between adjacent vesicles; (J) detail of gametoduct; (K) detail of internal organization of male vesicle; (L–P) sequence of gametoduct connecting the spermatocytes with the gastrovascular cavity. (A–P): longitudinal sections of gonad (cross sections of animals). See Table 2 for abbreviations.

*Calvadosia vanhoeffeni* (*Browne, 1910*) (Figs. 5E–5H; 5R, 5S; 7H; 10M–10O; 11I–11L; 12F; 13E–13G; 15A–15H; 16A–16C; 46; 47; 48; 49; 50; 57)

Basal pedal disk of peduncle with increased surface area due to invaginations (Figs. 46E and 46F). Base of peduncle with four gastric radial chambers (delimited by gastrodermis) (Figs. 47D and 47E). Gradual connection of four perradial chambers, defining one central

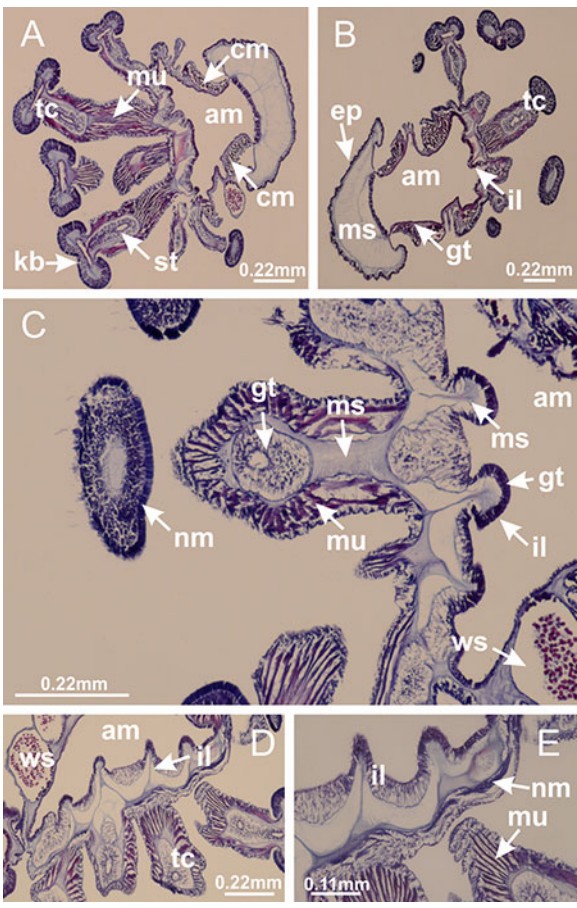

**Figure 41 Intertentacular lobules of *Calvadosia corbini.*** (A, B) Tip of arms; (C) region between secondary tentacles and intertentacular lobules, structures composed of a double layer of gastrodermis (of adjacent tentacles) and a central layer of mesoglea; (D, E) region between secondary tentacles and intertentacular lobules. (A–E): longitudinal sections. See Table 2 for abbreviations.

gastric chamber, and delimiting four interradial gastric septa toward apical region (Figs. 47F–47K). Peduncle without interradial longitudinal muscles, which are visible inside septa only at peduncle/calyx connection (Figs. 47L and 47M). Internal organization of infundibula (Figs. 47O and 47P) similar to *H. tenuis.* Gastrovascular cavity without claustrum. At base of infundibulum, interradial longitudinal muscle is compressed, and then becomes divided into two bands (Figs. 5E–5H). Internal organization of gastric radial pockets and manubrium similar to *H. tenuis* (Fig. 6). Four gastric radial pockets laterally separated from each other by interradial septa; gastric radial pockets directly connected only by means of small interradial ostia at margin of calyx (Figs. 10M–10O); each gastric radial pocket connected to main gastrovascular cavity. Manubrium internally defined by gastrodermis, externally by epidermis. Gastric filaments (Fig. 7H) similar to those of *H. tenuis.* Vesicles of gonads with irregular shape, but clearly defined, with prominent gastrodermis (Figs. 48 and 49). Two sets of vesicles in each gastric radial pocket; each vesicle formed by gastrodermis of two different interradial septa (two adjacent septa). Male specimen analyzed (Figs. 48 and 49), with spermatocytes adjacent to gastrodermis, in peripheral position; spermatozoa adjacent to epidermis, in central

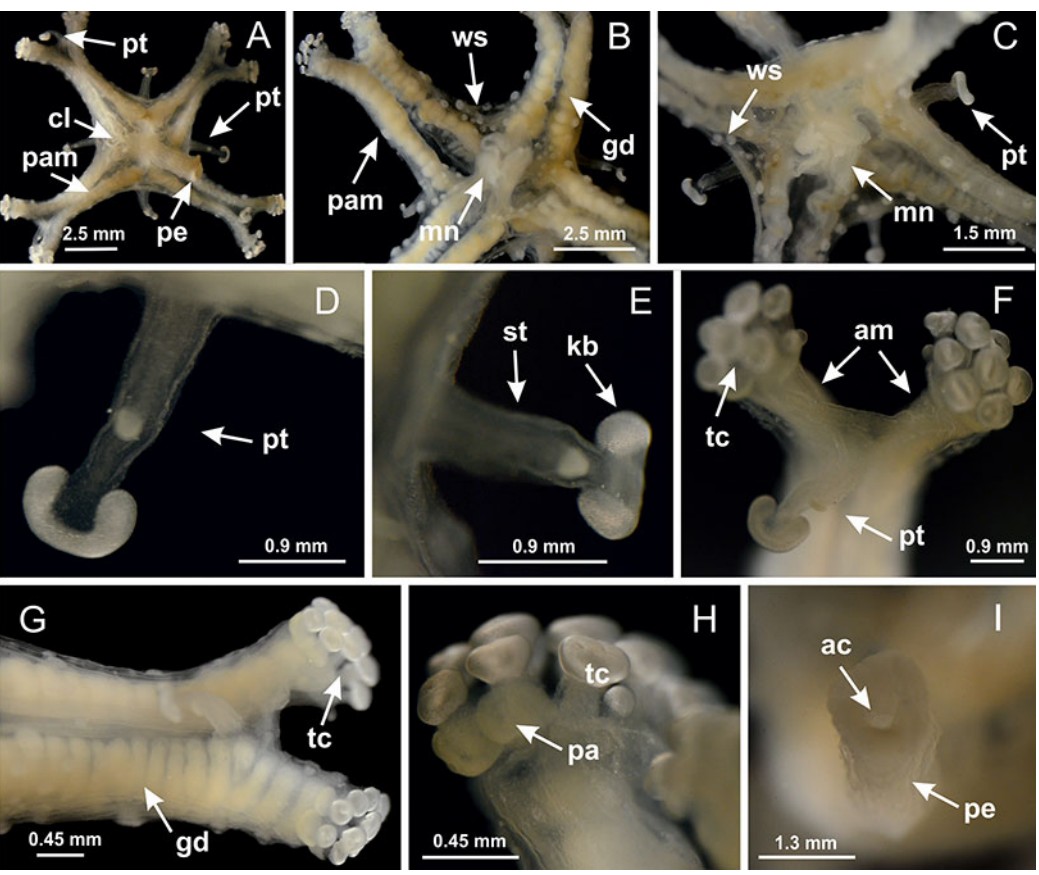

**Figure 42 General view of *Calvadosia cruciformis*.** (A) General view (exumbrella) of paired arms, peduncle, and primary tentacles; (B, C) general view (subumbrella) of paired arms, manubrium, gonads, and white spots of nematocysts; (D, E) primary tentacles, with horseshoe shaped knob; (F, G) paired arms with gonads, tentacular cluster and interradial primary tentacle, (H) pad-like adhesive structures in the outermost secondary tentacles; (I) detail of pedal disk with axial canal. See Table 2 for abbreviations.

position (Figs. 48B and 49L). Spermatozoa divided into different sacs, delimited by cells of gastrodermal origin, connected to gastrodermis of gastric radial pockets, forming gametoduct (Figs. 48F–48X and 49). Anchors absent (Figs. 46A and 46B). Arms with internal organization similar to *H. tenuis* (Figs. 11I–11L). Eight sections of coronal muscle at calyx margin, each between adjacent arms. Organization of longitudinal and coronal muscles in arms (Figs. 5R, 5S and 11L) similar to *H. tenuis*. Perradial and interradial white spots of nematocysts on subumbrella (Fig. 46H), with internal organization (Fig. 12F) similar to *H. tenuis*. Batteries of nematocysts sparsely distributed in exumbrellar epidermis (Figs. 13E–13G). Distal end of arms with intertentacular lobules (Fig. 50), a structure between adjacent secondary tentacles delimited by gastrodermis and a central layer of mesoglea. Outermost secondary tentacles with pad-like adhesive structures (epidermal thickening) as in *M. uchidai* (Figs. 24L–24N, 46J–46L). Clearly recognizable continuous layer of internal unorganized nematocysts in subumbrellar epidermis, from base of infundibula to tips of secondary tentacles (as in most of species examined; Figs. 15A–15H, 16A–16C). Secondary hollow tentacles (Figs. 46B, 46J and 46L) composed

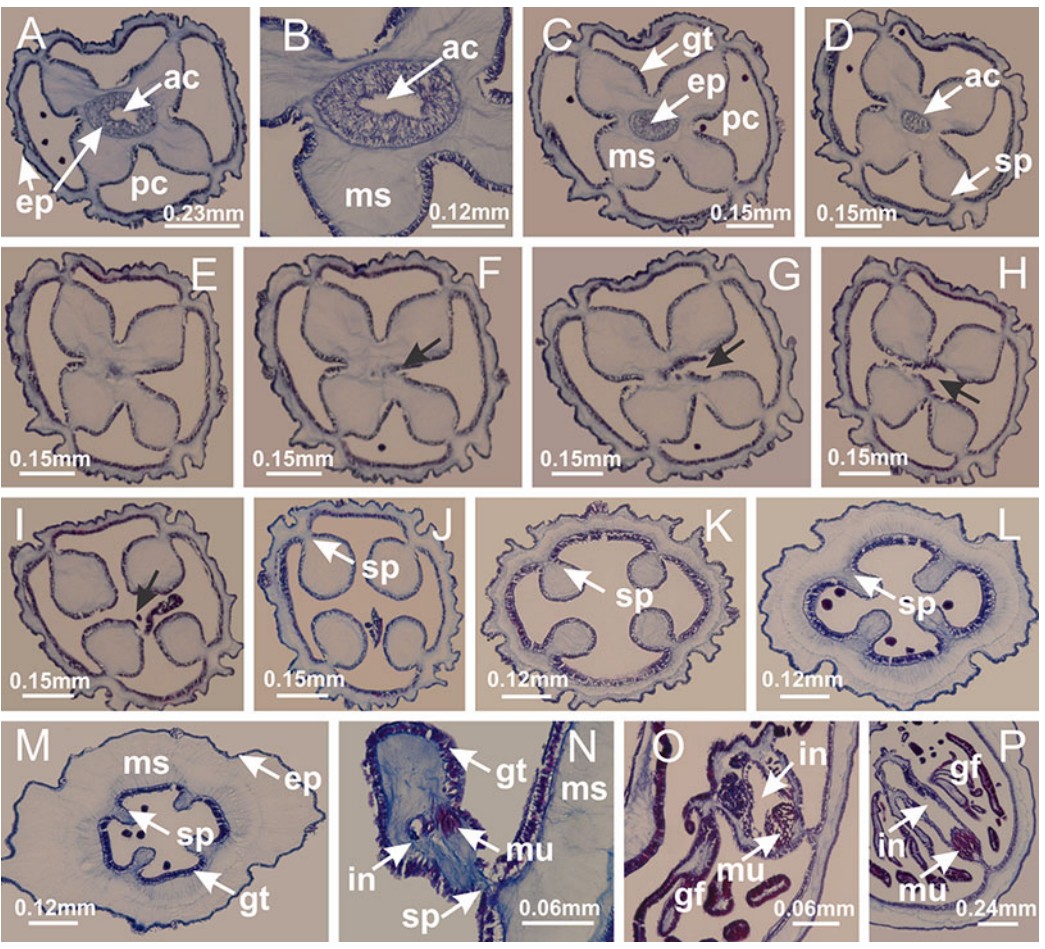

**Figure 43 Peduncle and septa of *Calvadosia cruciformis* (from base moving upward in A–P).**
(A) Base of peduncle, with four perradial chambers (delimited by gastrodermis), and a central axial canal (delimited by exumbrellar epidermis); (B) detail of axial canal; (C, D) four perradial chambers (delimited by gastrodermis), and a central axial canal (delimited by exumbrellar epidermis); (E–J) gradual fusion of perradial chambers (indicated by black arrows) and delimitation of four interradial septa without interradial longitudinal muscles; (K–M) modification in shape and size of interradial septa and central cruciform chamber; (N) septum at the peduncle/calyx connection, with infundibulum delimited by epidermis, and interradial longitudinal muscle; (O, P) septa at calyx base, with gastric filaments as lateral evaginations of septal gastrodermis, and infundibula delimited by epidermis. (A–P): cross sections. See Table 2 for abbreviations.

of two parts, knob and stem, with organization similar to *H. tenuis* (Figs. 17N–17Q and 50). At stem base, secondary tentacles tightly joined, separated only by thin layer of mesoglea, with beehive appearance in cross section as in most of species examined (Fig. 17).

**Family Craterolophidae** *Uchida, 1929*

**Genus** *Craterolophus* *Clark, 1863*

*Craterolophus convolvulus* (*Johnston, 1835*) (Figs. 7J; 17R–17U; 32A–32M; 51; 52; 53; 54; 55; 56; 57)

   Basal pedal disk of peduncle with increased surface area due to invaginations and a central pit (Figs. 51B–51C). Peduncle with four perradial chambers (delimited by

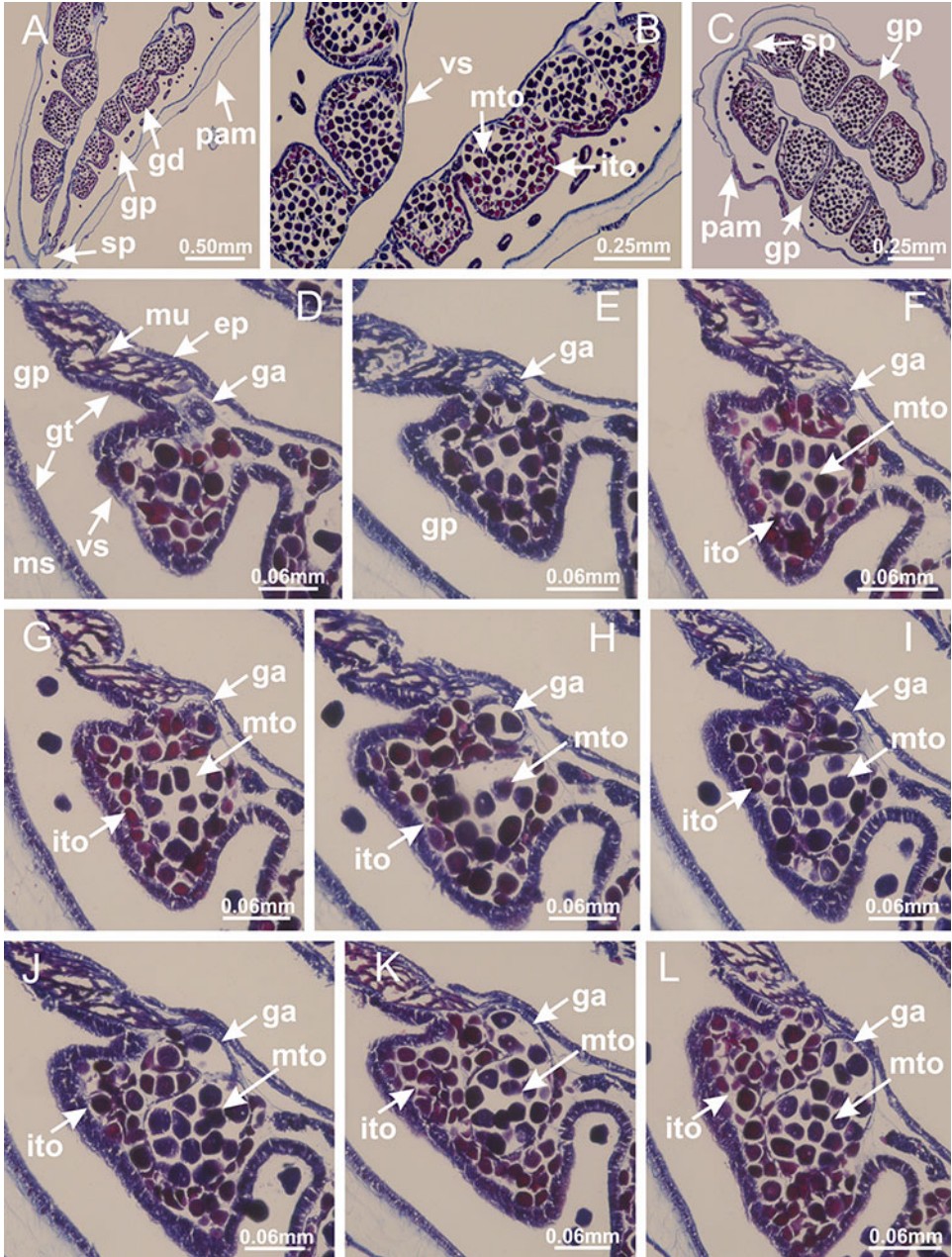

**Figure 44 Gonads and gametoduct of *Calvadosia cruciformis*.** (A) General view of female vesicles, inside gastric radial pockets in paired arms; (B) detail of vesicle, with central mature oocytes and peripheral immature oocytes; (C) organization of vesicles in a paired arm; (D–L) sequence of gametoduct connecting the mature oocytes with the gastrovascular cavity (gastric radial pockets). (A–L): cross sections of animal (longitudinal sections of vesicles). See Table 2 for abbreviations.

gastrodermis), and without interradial longitudinal muscles (Figs. 52A–52K). Size of four chambers gradually increases to merge as a single chamber at peduncle/calyx connection (Figs. 52M, 52N and 52P), delimiting four interradial septa (Figs. 52O and 52P). At peduncle/calyx connection, lateral projections of adjacent interradial septa, composed of a central layer of mesoglea surrounded by gastrodermis, progressively merges, defining the

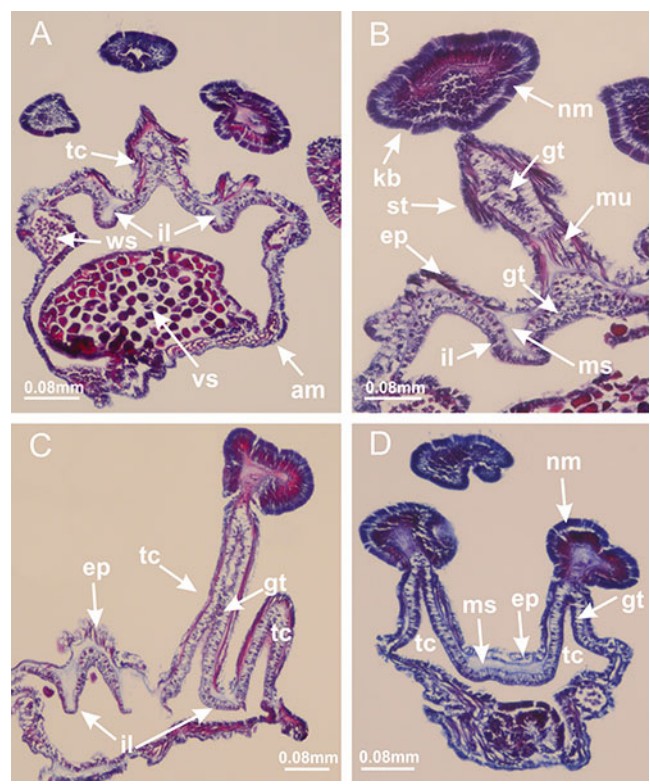

**Figure 45 Intertentacular lobules of *Calvadosia cruciformis*.** (A) General organization of tip of arms; (B–D) secondary tentacles and intertentacular lobules, structures composed of a double layer of gastrodermis (of adjacent tentacles) and a central layer of mesoglea. (A–D): longitudinal sections. See Table 2 for abbreviations.

claustra, tissues that divide gastrovascular cavity (Figs. 52Q–52S, 53A–53F). Four accessory radial pockets delimited (separated from main gastrovascular cavity) by claustra (Figs. 53B–53D and 53F). Four infundibula funnel-shaped with blind end, delimited by epidermis, deeply developed down to base of calyx, widening apically, with broad apertures on subumbrella (Figs. 53C and 53F). Interradial longitudinal muscle (epitheliomuscular cells) divided into two bands, visible only after complete formation of claustra (Figs. 53G and 53H). Septa with complex lateral folds, delimiting auxiliary radial pockets (Figs. 53G, 53H, 53J and 54). Below manubrium delimitation, lateral folds of septa produce auxiliary radial pockets inside infundibula (Figs. 53G and 53H), composed of external epidermis, a layer of gonadal content, and internal gastrodermis (Fig. 54B). As adjacent septal gastrodermis and epidermis merge, dividing once more the gastrovascular cavity and delimiting four principal radial pockets and manubrium (similarly to *M. uchidai*, Fig. 20), these auxiliary radial pockets are externalized, still connected to subumbrella (Figs. 53J, 54A and 54B). Above manubrium and principal radial pocket delimitation, new auxiliary radial pockets formed inside principal radial pocket (Fig. 54C), also as a result of irregular folds of lateral tissue of principal radial pocket, always containing gonads (Figs. 54E–54H). Auxiliary radial pocket inside principal radial pocket composed of external gastrodermis, a layer of gonadal content, and internal epidermis (Fig. 54D). Gastrodermis and epidermis of auxiliary radial pockets

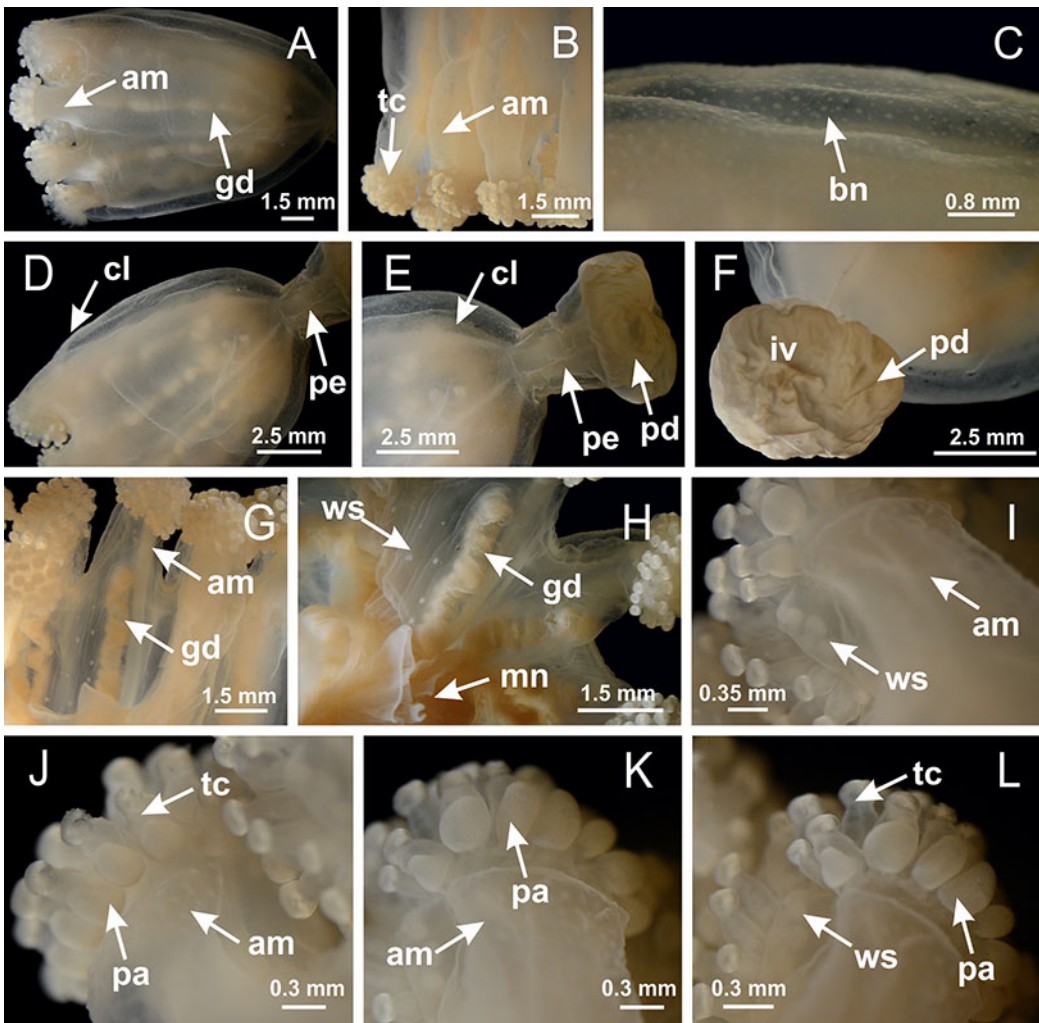

**Figure 46 General view of *Calvadosia vanhoeffeni*.** (A, B) General view of calyx (exumbrella), with gonads, arms, and tentacular cluster; (C) batteries of nematocysts in the exumbrella; (D) general view of calyx and peduncle; (E) detail of peduncle; (F) detail of pedal disk, with invaginations; (G, H) sub-umbrellar view of calyx, with arms, gonads, white spots of nematocysts, and manubrium; (I–L) tip of arms, with tentacular cluster and outermost secondary tentacles with pad-like adhesive structures. See Table 2 for abbreviations.

progressively merge with lateral gastrodermis and epidermis of principal radial pocket, externalizing auxiliary radial pockets (Figs. 54I–54L). Therefore, gastrovascular system in *C. convolvulus* divided into numerous radial pockets (Figs. 53J, 54A and 54L): four accessory radial pockets, directly associated with perradial chambers in peduncle, arms and secondary tentacles; four principal radial pockets, associated with manubrium and gonads; and numerous auxiliary radial pockets (external and internal), associated with gonads (Figs. 53 and 54). Accessory and principal radial pockets separated by claustra (Figs. 53J, 54A and 54L). Accessory radial pockets laterally separated from each other by interradial septa (Fig. 53F); directly connected only by means of small interradial ostia at margin of calyx as in *M. uchidai*, and in gastric radial pockets of species without claustrum, such as *C. vanhoeffeni* and *C. corbini* (Figs. 10M–10U). Manubrium

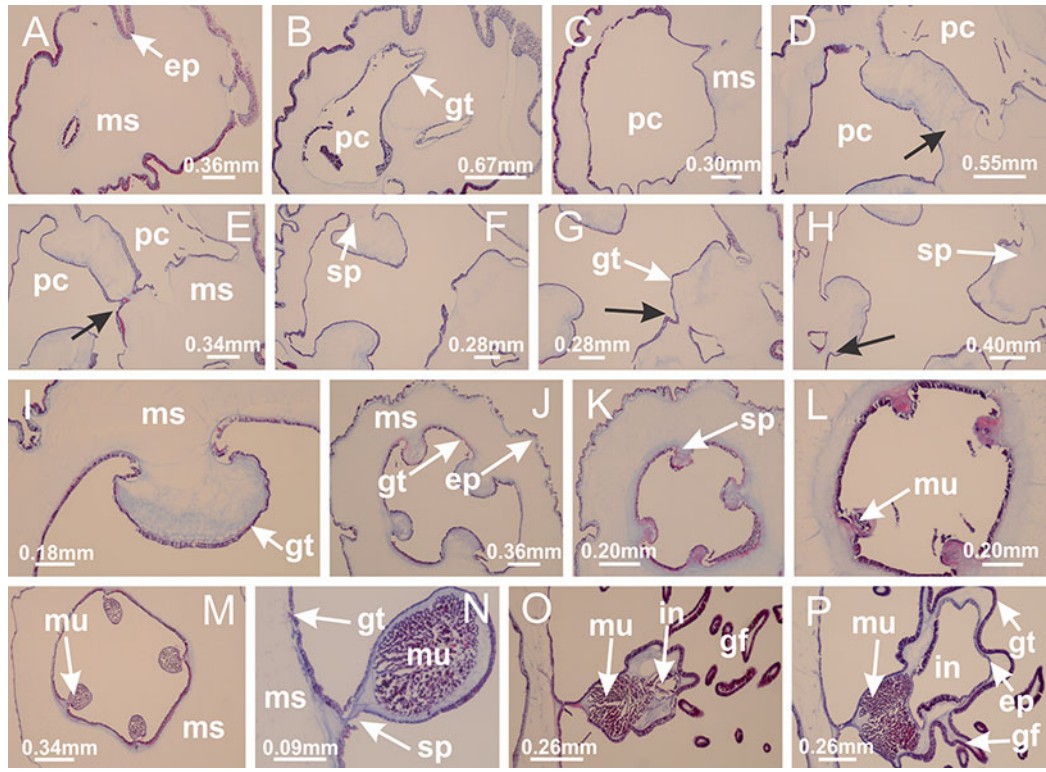

**Figure 47 Peduncle and septa of *Calvadosia vanhoeffeni* (from base moving upward in A–P).**
(A) Base of peduncle; (B, C) perradial chamber (one in evidence, but four in total), delimited by gastrodermis, and separated by interradial and central layer of mesoglea; (D–H) fusion of four perradial chambers, and delimitation of the four interradial septa (indicated by black arrows); (I) detail of interradial septum, without interradial longitudinal muscle; (J, K) four interradial septa, with a central chamber delimited by gastrodermis; (L, M) septa at the peduncle/calyx connection, with interradial longitudinal muscles; (N) detail of septum with interradial longitudinal muscle; (O, P) gastric filaments as lateral evaginations of septal gastrodermis, and septa with infundibula delimited by epidermis, at calyx base. (A–P): cross sections. See Table 2 for abbreviations.

(Figs. 51I and 54C) internally defined by gastrodermis, externally by epidermis. Gastric filaments similar to those of *H. tenuis* in internal organization, associated with principal radial pocket (Fig. 7J). Gonadal content restricted to one layer between gastrodermis and epidermis of septa, principal radial pockets, and auxiliary radial pockets, not organized in vesicles (Figs. 53–55). Gonadal layers of same principal radial pocket formed by two different adjacent interradial septa. Female specimen analyzed (Figs. 53–55), with immature oocytes adjacent to gastrodermis and peripheral; mature oocytes adjacent to epidermis and central (Figs. 55A–55C). Mature oocytes surrounded by a layer of cells with gastrodermal origin (probably follicle cells), which merges with gastrodermis of gastrovascular cavity (including gastrodermis of principal radial pocket and gastrodermis of auxiliary radial pocket), forming gametoduct (Figs. 55D–55Q). Cilia often associated with gametoduct (Fig. 55D). Anchors absent (Fig. 51A). Each accessory radial pocket extending throughout calyx, apically continuing into two adradial arms and respective tentacular clusters. Internal organization of arms similar to *H. tenuis* (Fig. 11). Eight sections of coronal muscle at calyx margin, each between adjacent arms (Fig. 51A).

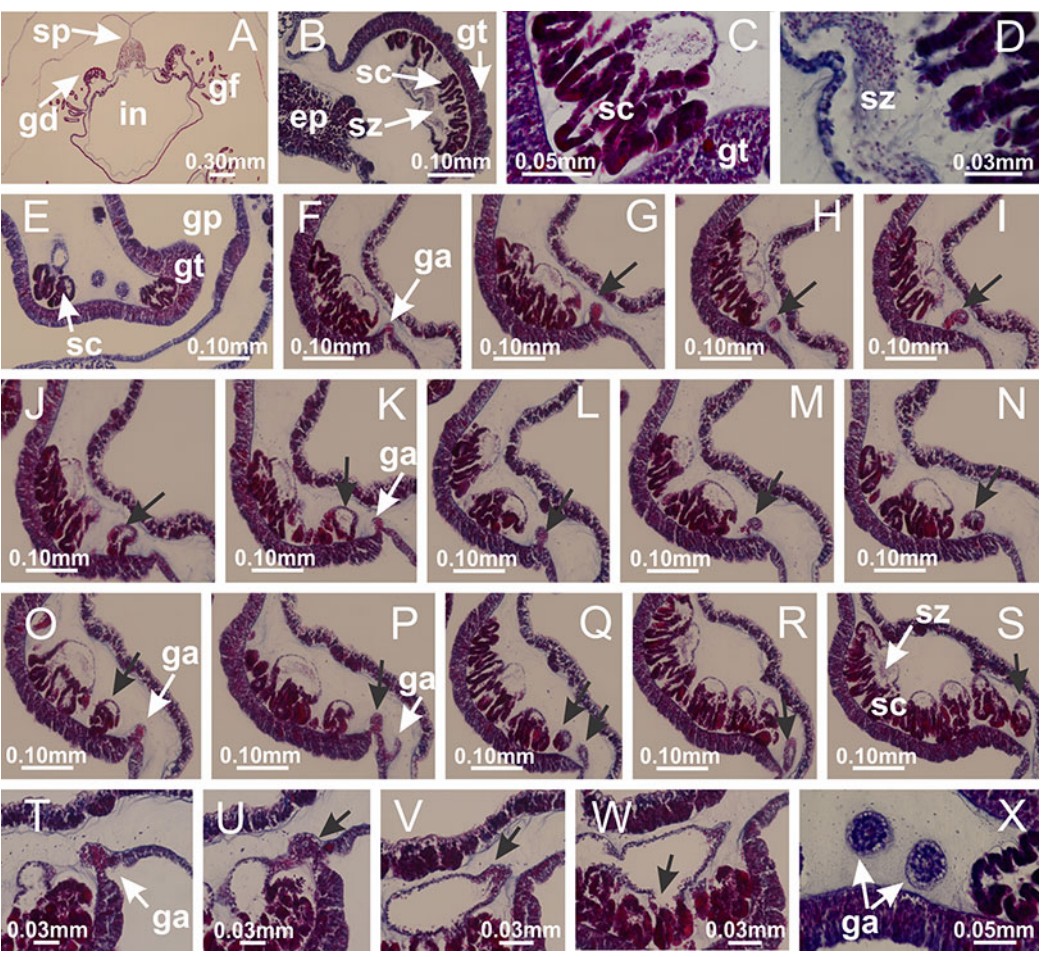

**Figure 48 Gonads and gametoduct of *Calvadosia vanhoeffeni*.** (A) General view of gonad, as lateral evaginations of septum; (B–E) gonadal content between a layer of gastrodermis (adjacent to spermatocytes) and epidermis (adjacent to spermatozoa); (F–S) sequence of gametoduct connecting the spermatozoa and spermatocytes with the gastrovascular cavity of gastric radial pocket (indicated by black arrows); (T–W) sequence of gametoduct connecting the spermatozoa and spermatocytes with the gastrovascular cavity of gastric radial pocket (indicated by black arrows); (X) detail of gametoduct. A–X: cross sections of body, and longitudinal sections of gonads. See Table 2 for abbreviations.

Organization of longitudinal and coronal muscles in arms similar to *H. tenuis* (as in other species examined; Figs. 5 and 11). Perradial and interradial white spots of nematocysts on subumbrella, with internal organization (Figs. 32A–32M) similar to *H. tenuis*. Aperture of white spots at subumbrellar epidermis clearly recognizable in its central thicker region (Figs. 32G–32M): a pore divides subumbrellar epidermis and adjacent layer of mesoglea into two regions in a longitudinal section, allowing an outflow to central mature nematocysts (Fig. 32M). Batteries of nematocysts sparsely distributed in exumbrellar epidermis (as in other species examined; Fig. 13). Distal end of arms with intertentacular lobules, a structure between adjacent secondary tentacles delimited by gastrodermis and a central layer of mesoglea (Fig. 56). Outermost secondary tentacles with pad-like adhesive structures (epidermal thickening) as in *M. uchidai* (Figs. 24L–24N, 51E–51G). Continuous layer of internal unorganized nematocysts in subumbrellar

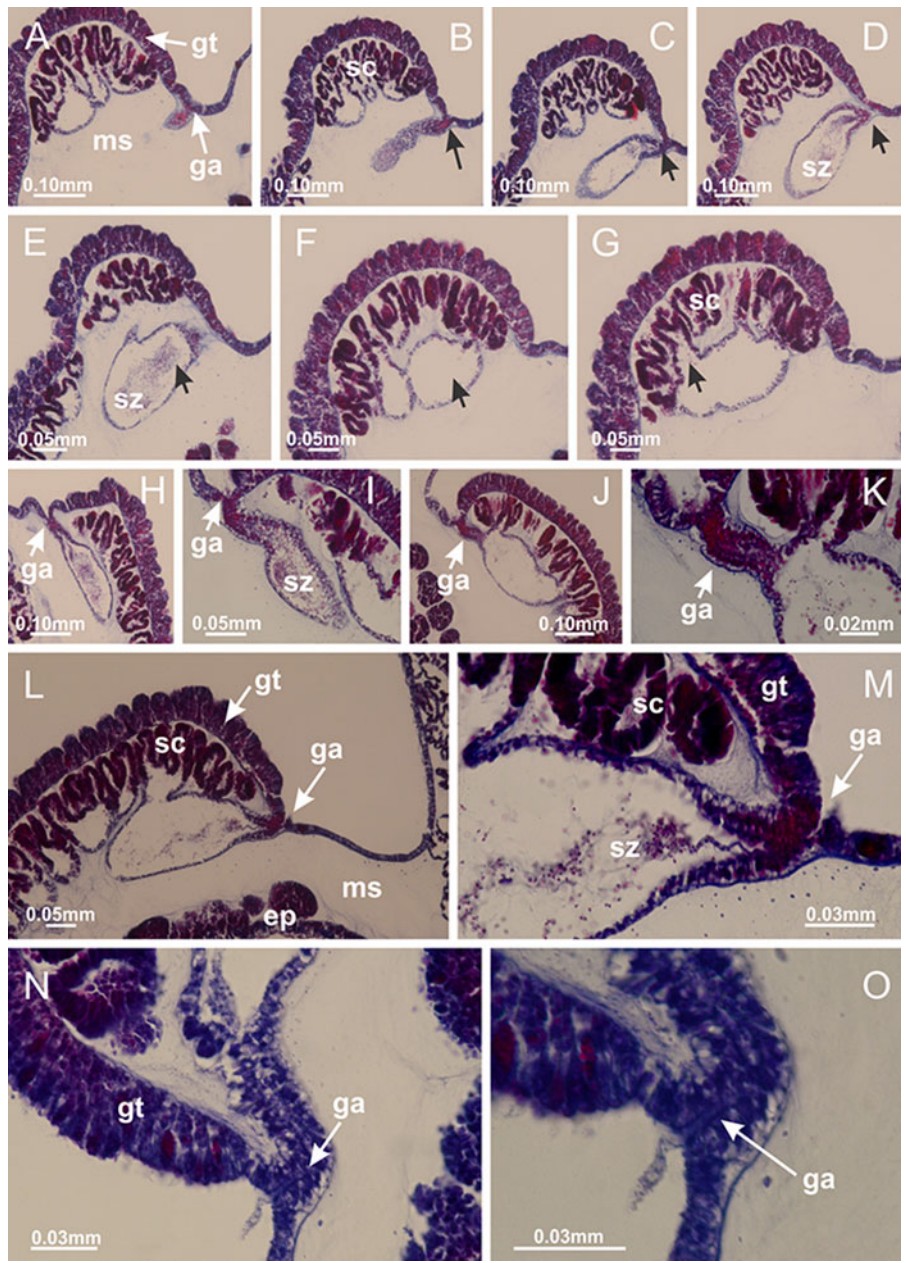

**Figure 49 Gonads and gametoduct of *Calvadosia vanhoeffeni*.** (A–G) Sequence of gametoduct connecting the spermatozoa and spermatocytes with the gastrovascular cavity of gastric radial pocket (indicated by black arrows); (H–J) sequence of gametoduct connecting the spermatozoa and spermatocytes with the gastrovascular cavity of gastric radial pocket; (K) detail of gametoduct; (L) gametoduct connecting the spermatozoa and spermatocytes with the gastrovascular cavity of gastric radial pocket; (M–O) detail of gametoduct. A–O: longitudinal sections of gonads. See Table 2 for abbreviations.

epidermis as in most of other species examined (Figs. 15 and 16). Secondary hollow tentacles composed of two parts, knob and stem (Figs. 17R–17U and 56), with organization similar to *H. tenuis*. At stem base, secondary tentacles tightly joined, separated only by thin layer of mesoglea, with beehive appearance in cross section as in most of species examined (Fig. 17).

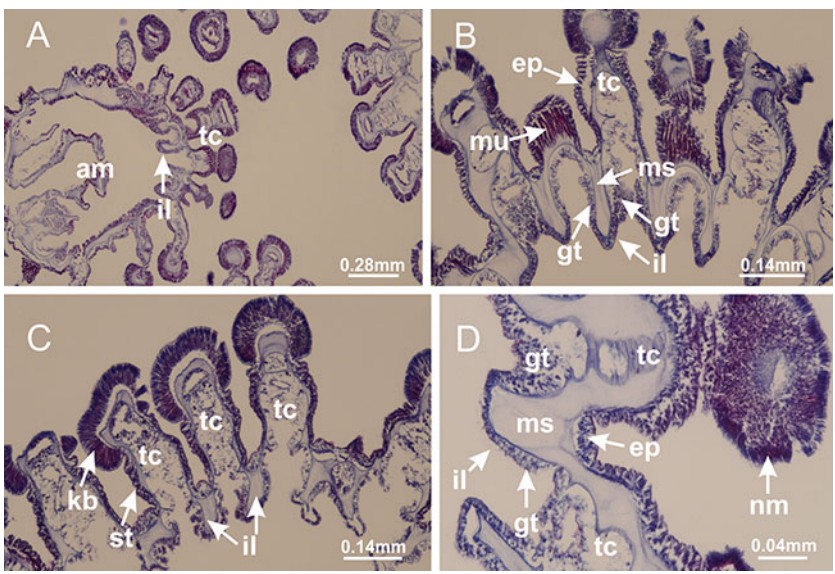

**Figure 50 Intertentacular lobules of *Calvadosia vanhoeffeni*.** (A) General organization of tip of arms, in the region between intertentacular lobules and secondary tentacles; (B–D) detail of intertentacular lobules, structures composed of a double layer of gastrodermis (of adjacent tentacles) and a central layer of mesoglea. (A–D): longitudinal sections. See Table 2 for abbreviations.

# DISCUSSION

Taxonomic characters of Staurozoa, such as the number of secondary tentacles, presence/absence of anchors/primary tentacles, the shape of anchors, presence/absence of claustrum, the number of chambers in the peduncle, and the presence/absence of four interradial longitudinal muscles in the peduncle have traditionally been employed in the taxonomy of the group (*Clark, 1863*; *Uchida, 1929*; *Kramp, 1961*). However, some of these characters vary ontogenetically, and must be cautiously employed to differentiate species (*Uchida, 1929*; *Hirano, 1986*; *Miranda, Morandini & Marques, 2009*). Additionally, the recent molecular phylogenetic hypothesis indicates the need to reassess and reinterpret many of the internal characters, such as the claustrum and the longitudinal muscles (see *Miranda et al., 2016*). We will focus our discussion on the internal anatomy of staurozoans from functional and evolutionary perspectives, in the historical context provided by the phylogenetic hypothesis and classification proposed for the group (Figs. 1, 57 and 59; *Miranda et al., 2016*).

## Longitudinal and coronal muscles

The musculature of stalked jellyfishes is organized into two muscular arrangements: circular (coronal) and longitudinal (radial) muscles (*Uchida, 1929*; *Berrill, 1963*; *Miranda, Collins & Marques, 2013*). The main muscular arrangement in stauromedusae is longitudinal, and not circular as generally observed in active and planktonic scyphomedusae and cubomedusae (*Gwilliam, 1960*; *Arai, 1997*; *Satterlie, Thomas & Gray, 2005*), consistent with the benthic and sessile habit of staurozoans (*Gwilliam, 1960*; *Miranda, Collins & Marques, 2015*).

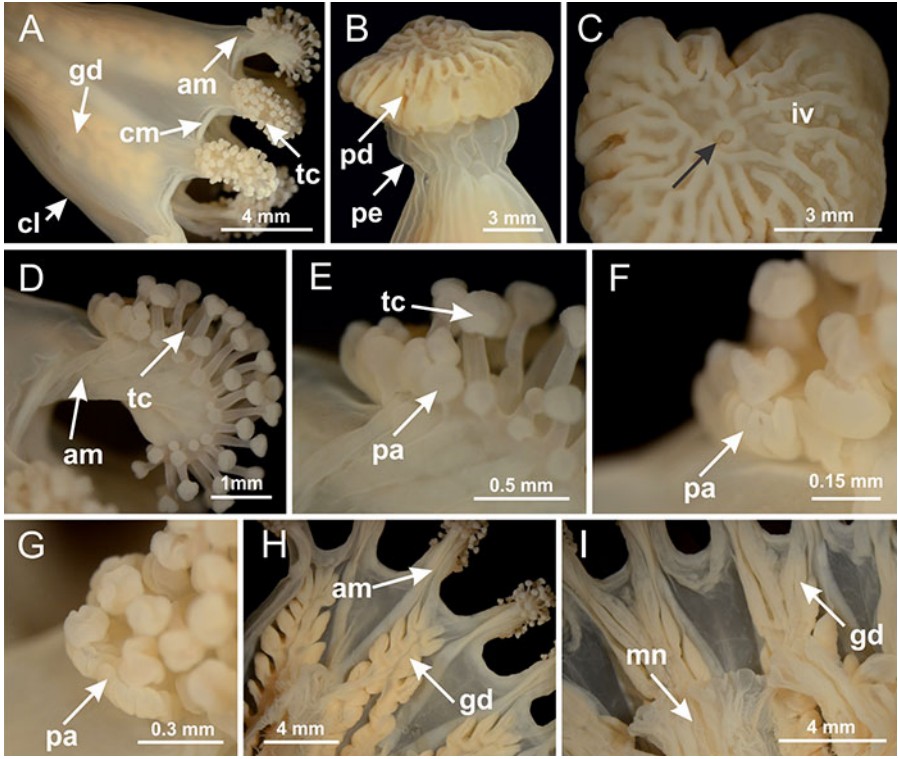

**Figure 51 General view of *Craterolophus convolvulus*.** (A) General view of calyx; (B) peduncle; (C) detail of pedal disk, with a central pit (indicated by black arrow); (D) tentacular cluster; (E–G) secondary tentacles and pad-like adhesive structures in the outermost secondary tentacles; (H, I) sub-umbrellar view, with manubrium and gonads. See Table 2 for abbreviations.

The recently proposed phylogenetic hypothesis for Staurozoa (Fig. 1) suggests that interradial longitudinal muscles in the peduncle played a fundamental role in the evolution of the class (*Miranda et al., 2016*). The new phylogeny-based classification scheme separates the group into two suborders, Myostaurida and Amyostaurida, animals with and without interradial longitudinal muscle in the peduncle at the stauromedusa stage, respectively (Figs. 1, 5, 57 and 59; Table 3; *Miranda et al., 2016*). Because the last common ancestor of Staurozoa is inferred to have had peduncular muscles (Table 3), a loss of these muscles can be inferred to have happened in the lineage leading to Amyostaurida (Fig. 59). As expected, our new observations of internal anatomy corroborate the absence of interradial longitudinal muscles in the peduncle in representatives of Amyostaurida, *Craterolophus* and *Calvadosia*, differing from the condition in Myostaurida, as represented by species of *Haliclystus*, *Lucernaria*, and *Manania* (Fig. 57).

In *Lucernaria* species, the interradial longitudinal muscles are associated with the interradial septa (occupying an internal position in the septa) from the base of the peduncle (Figs. 26, 30, 35 and 57; *Collins & Daly, 2005*), whereas in *H. tenuis* and *M. uchidai* the interradial longitudinal muscles in the peduncle are intramesogleal because the septa are only present in the region where the calyx connects to the peduncle (Figs. 4, 19 and 57; *Uchida, 1929*; *Ling, 1939*; *Miranda, Collins & Marques, 2013*). In species of

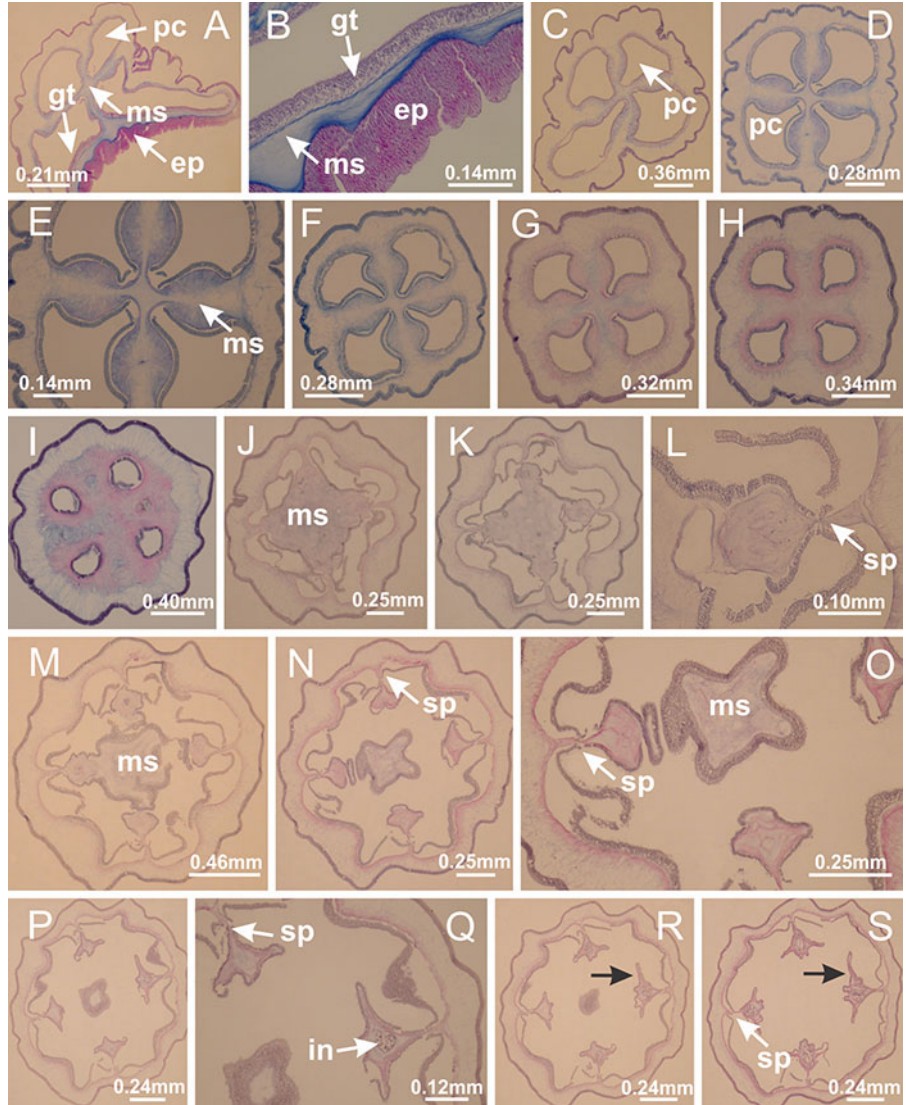

**Figure 52 Peduncle and septa of *Craterolophus convolvulus* (from base moving upward in A–S).**
(A) Four perradial chambers in peduncle; (B) detail of epidermis, mesoglea, and gastrodermis of peduncle; (C-I) variation of shape and size of perradial chambers in peduncle; (J, K) delimitation of four interradial septa, (L) detail of delimitation of septum; (M–O) delimitation of four interradial septa (connection of four perradial chambers and reduction of central mesoglea), (P–S) projections of lateral tissue (double layer of gastrodermis and central layer of mesoglea; indicated by black arrows), and infundibula at central region of septa. (A–S): cross sections. See Table 2 for abbreviations.

*Haliclystus*, the gastrodermis of the central chamber envelops the interradial longitudinal muscles in the region of the peduncle/calyx connection (Fig. 4H; *Uchida, 1929*; *Uchida & Hanaoka, 1934*; *Ling, 1939*; *Miranda, Collins & Marques, 2013*), delimiting the four interradial septa with internal interradial longitudinal muscles, similar to *Lucernaria* (Fig. 57). On the other hand, *M. uchidai* has a slightly different organization (Figs. 19 and 20): the interradial septa are gradually formed as the size of the four perradial chambers increases in the region of the peduncle/calyx connection. The interradial longitudinal muscles do not occupy an internal position in the septa,

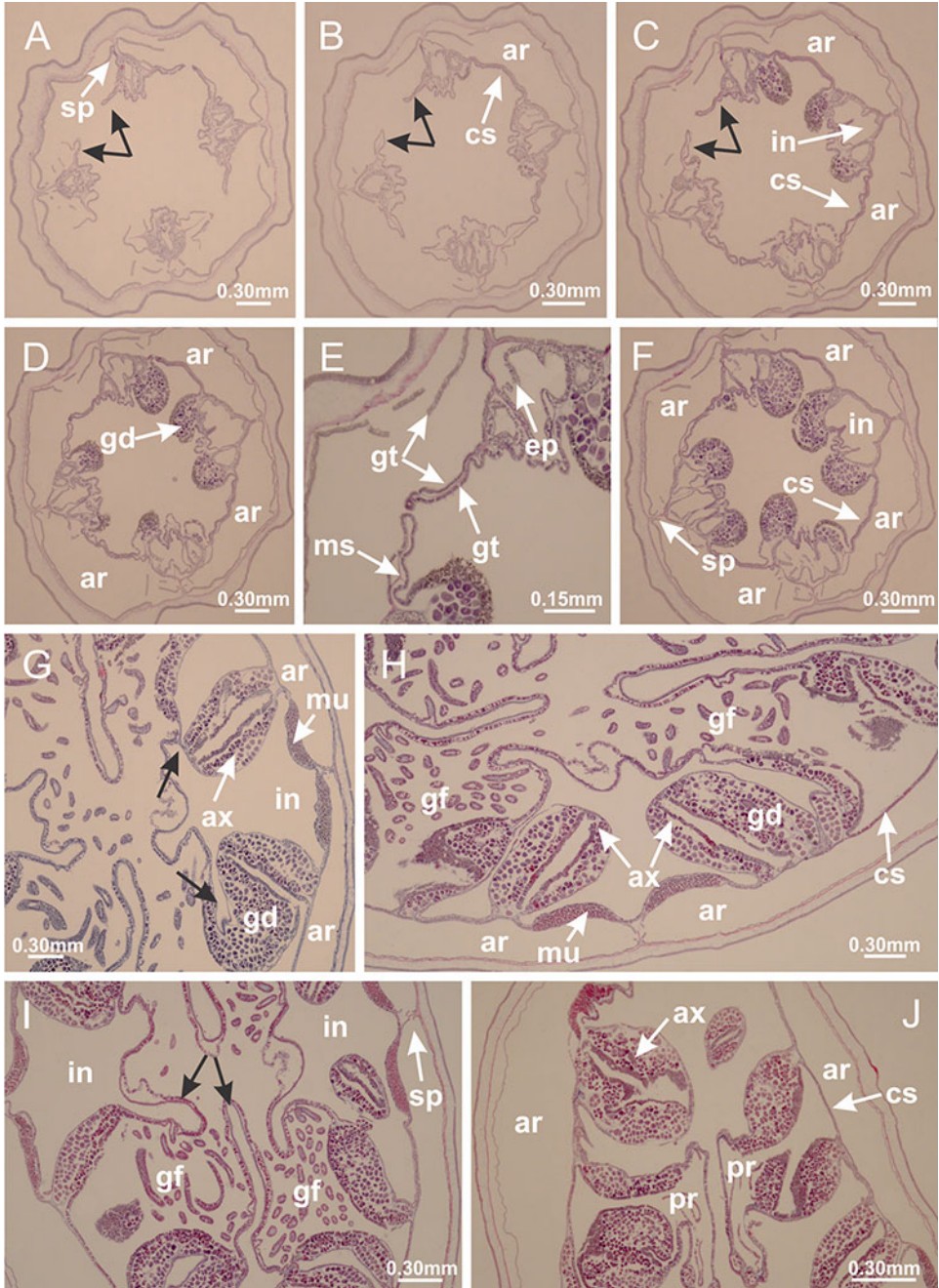

**Figure 53 Claustra in *Craterolophus convolvulus*.** (A–D) Claustra delimitation (lateral projections of adjacent septa, indicated by black arrows); (E) detail of claustra delimitation (fusion of lateral projections of adjacent septa); (F) claustra dividing the gastrovascular cavity, delimiting accessory radial pockets; (G) formation of auxiliary radial pockets, due to folds in the septum containing gonadal content (indicated by black arrows); (H) organization of septum with auxiliary radial pockets at the base of calyx (below manubrium delimitation); (I) fusion of gastrodermis and epidermis of adjacent septa (indicated by black arrows), during delimitation of manubrium and principal radial pocket (gastric filaments associated with principal radial pocket); (J) complete delimitation of principal radial pocket and auxiliary radial pocket. (A–J): cross sections. See Table 2 for abbreviations.

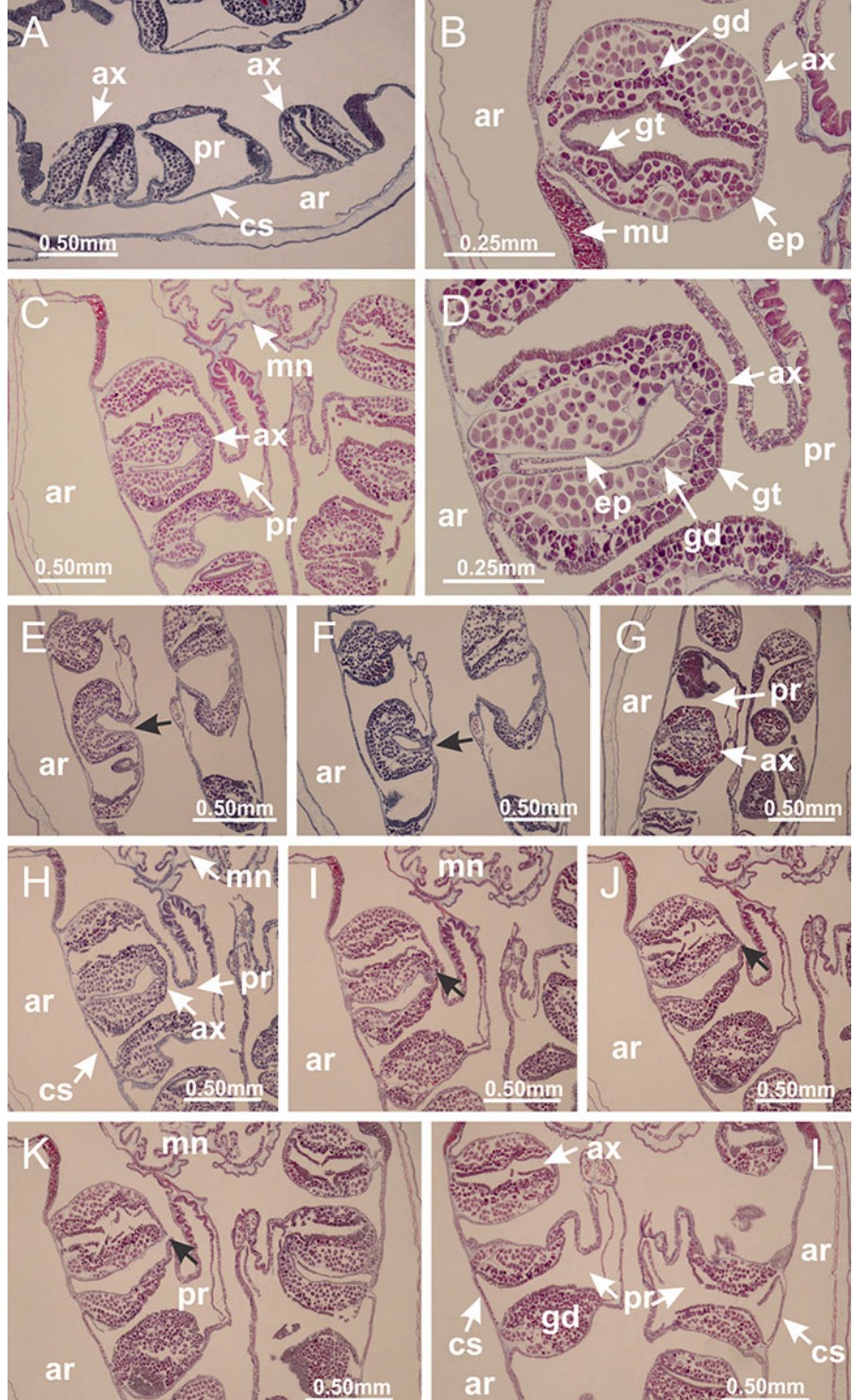

**Figure 54 Organization of gonads in *Craterolophus convolvulus*.** (A) Gonads associated with principal radial pockets and auxiliary radial pockets; (B) organization of auxiliary radial pocket inside infundibulum or outside principal radial pocket (central gastrodermis associated with immature oocytes; and mature oocytes associated with external epidermis); (C) auxiliary radial pocket inside principal radial pocket; (D) organization of auxiliary radial pocket inside principal radial pocket (central epidermis associated with mature oocytes, and external gastrodermis associated with immature oocytes); (E–G) delimitation of internal auxiliary radial pocket (indicated by black arrows); (H–L) delimitation of external auxiliary radial pocket (indicated by black arrows). (A–L): cross sections. See Table 2 for abbreviations.

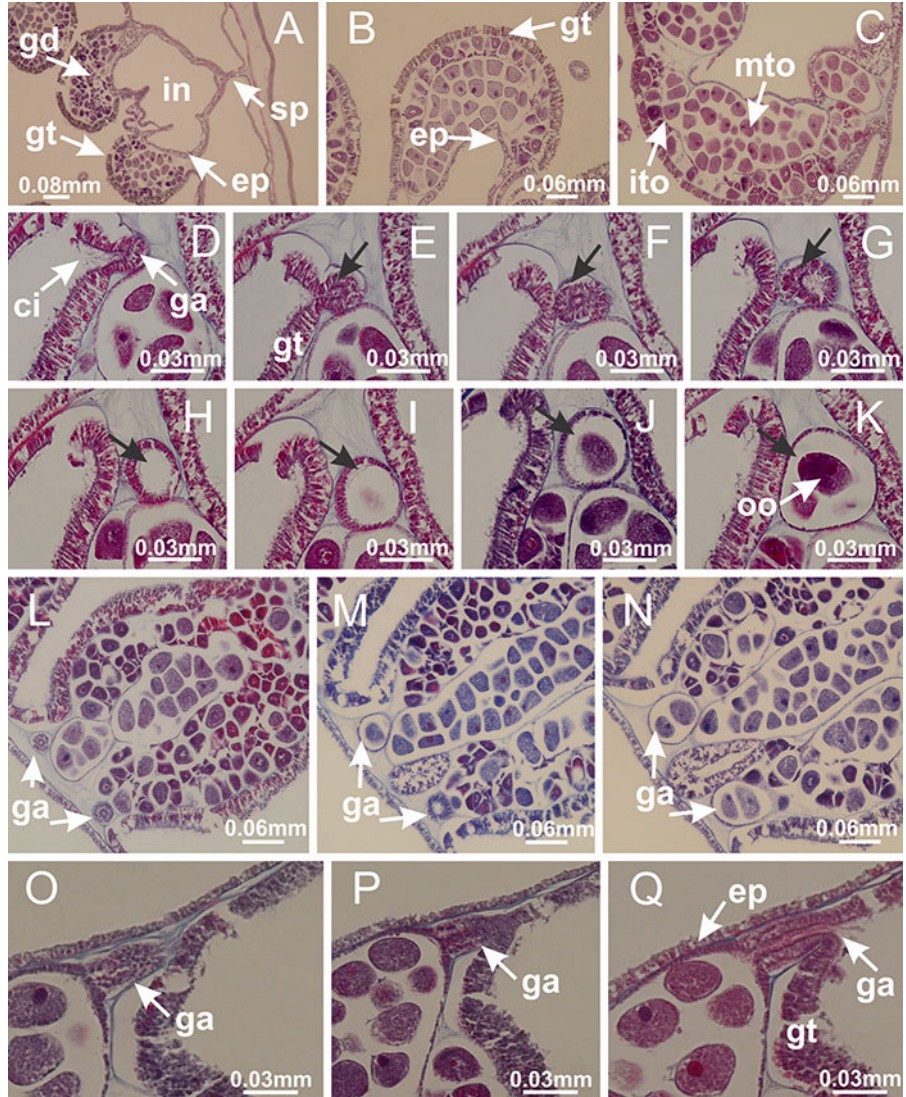

**Figure 55 Gonads and gametoduct of *Craterolophus convolvulus*.** (A, B) Gonadal content between a layer of gastrodermis and epidermis of septum; (C) immature oocytes adjacent to gastrodermis, and mature oocytes adjacent to epidermis; (D–K) gametoduct connecting the mature oocytes with the gastrovascular cavity of principal radial pocket (indicated by black arrows); (L–N) gametoduct; (O–Q) detail of gametoduct, showing direct connection of mature oocytes with gastrovascular cavity. (A–N): cross sections; (O–Q): longitudinal sections. See Table 2 for abbreviations.

remaining external in relation to the septal constriction (Fig. 21), a condition that changes only at the base of the manubrium (Fig. 20I). *Manania uchidai* has a central tissue (mesoglea and gastrodermis) (Figs. 20A and 20B) connecting the four interradial septa that disappears only after the formation of claustra (Figs. 20D–20G). However, our observations contrast with those of *Uchida & Hanaoka (1933)*, who reported that the septa are completely separated below the formation of the claustra in *M. uchidai* (identified therein as *Manania distincta*). Either the central connection of septa varies intraspecifically or one report is an artefact.

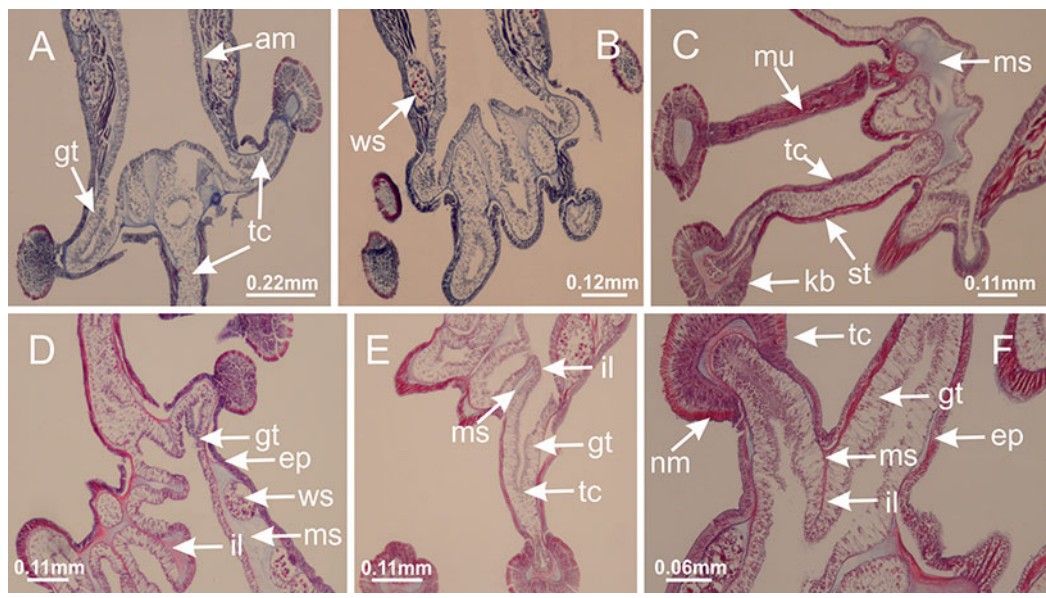

**Figure 56 Intertentacular lobules of _Craterolophus convolvulus_.** (A, B) Tip of the arms; (C) tentacles, with longitudinal muscle; (D, E) organization of tip of arms and intertentacular lobules; (F) detail of intertentacular lobules, structures composed of a double layer of gastrodermis (of adjacent tentacles) and a central layer of mesoglea. (A–F): longitudinal sections. See Table 2 for abbreviations.

Species of Amyostaurida do not have interradial longitudinal muscle in the peduncle (Figs. 5, 57 and 59; Table 3). The examined species of _Calvadosia_ have four perradial chambers at the base of the peduncle, which progressively merge, producing four interradial septa and one central gastric chamber (Figs. 39, 43, 47 and 57). The interradial longitudinal muscles only appear at the peduncle/calyx connection, internal to the septa (Figs. 39, 43, 47 and 57). _Craterolophus convolvulus_ has four perradial chambers separated by mesoglea throughout the peduncle (Fig. 52). At the peduncle/calyx connection, the perradial chambers of _C. convolvulus_ increase in size and merge centripetally, delimiting the interradial septa (Figs. 52J–52N). A central mesoglea delimited by gastrodermis is still visible (Figs. 52N–52P), but progressively disappears apically (Figs. 52R and 52S). Interradial muscles associated with septa are clearly visible only after claustra formation (Fig. 53G), differing from other species of Amyostaurida, in which the interradial longitudinal muscles become visible at the base of the infundibula (Figs. 39, 43 and 47).

Interradial longitudinal muscles are divided into two bands in the calyx of all species (Figs. 5D, 5H and 5L). These bands gradually assume an adradial position toward the calyx margin (Figs. 5M and 11B). In the amyostaurid _C. convolvulus_, when the interradial longitudinal muscles are visible in the cross sections of the calyx, they are already divided into two bands (Figs. 53G and 53H) that progressively assume a more adradial position. Each longitudinal muscle band is associated apically with one arm and the correspondent stems of secondary tentacles (Figs. 5Q–5V and 11). Accordingly, there is a continuum of the longitudinal muscles found in the peduncle (in Myostaurida) or at the base of the calyx (Amyostaurida), with the muscles in the stem of secondary tentacles.

In addition, other regions of the body, such as the manubrium, are provided with thin muscle fibers (*Gwilliam, 1960*; *Miranda, Collins & Marques, 2015*).

The coronal muscle can be entire, as in *M. uchidai* (Figs. 18C and 18D), running as an external (exumbrellar) and non-interrupted band in the arms; or discontinuous, divided into perradial and interradial sections by the arms, as in *H. tenuis* (Fig. 59; Table 3; *Clark, 1863*; *Carlgren, 1935*; *Gwilliam, 1956*; *Kramp, 1961*). Cross sections in the arms clearly show two lateral bands of coronal muscle, one from interradii and the other from perradii (Figs. 5Q, 5R, 11H and 11L). The presence of coronal muscle (vestigial in *Stylocoronella*; *Kikinger & Salvini-Plawen, 1995*) and its relative position with respect to the anchors/primary tentacles have also been used to differentiate genera (*Clark, 1863*; *Mayer, 1910*; *Uchida, 1929*; *Carlgren, 1935*; *Gwilliam, 1956*; *Kramp, 1961*; *Miranda et al., 2016*).

The contraction of the coronal musculature, along with contraction of the longitudinal muscles, considerably reduces the total volume of the animal, probably making its adherence to substratum more efficient in wave-exposed habitats (*Hyman, 1940*; *Miranda, Collins & Marques, 2013*; see also *Kier, 2012*, for discussion about the arrangement of muscular fibers and their role in support and movement in hydrostatic skeletons). In addition, longitudinal muscles in arms and secondary tentacles are likely related to feeding behaviors (*Hyman, 1940*; *Larson, 1980*; *Miranda, Collins & Marques, 2013*). Spontaneous neuromuscular activity in *Haliclystus* is arrhythmic, and stauromedusae lack the ability to perform pulsating swimming motions, unlike free-swimming jellyfishes (*Gwilliam, 1960*).

## Chambers in the peduncle

Most of the staurozoan genera have four chambers in the peduncle and this state was suggested as a potential synapomorphy of Staurozoa (Fig. 59; Table 3; *Collins & Daly, 2005*; *Miranda et al., 2016*). However, the number of chambers in the peduncle has been considered to be a useful character to distinguish staurozoan genera (*Clark, 1863*; *Mayer, 1910*; *Uchida, 1929*; *Kramp, 1961*; *Larson & Fautin, 1989*). The stalked jellyfishes can have either (a) one chamber in the peduncle (e.g., *Lucernaria*, *Lipkea*; *Kramp, 1961*); (b) four perradial chambers (e.g., *Haliclystus*, *Craterolophus*; *Kramp, 1961*); (c) four perradial chambers at the base, merging into one chamber medially (e.g., *Calvadosia*; *Mayer, 1910*; *Miranda et al., 2016*); (d) one chamber at the base splitting into four chambers medially (e.g., some *Manania*, *Larson & Fautin, 1989*).

In addition, the numbers of chambers in the peduncle were also proposed as the "most useful" character for differentiating species of *Manania* (*Larson & Fautin, 1989*: 1547). The genus *Manania* includes species with one chamber throughout the peduncle, four chambers throughout the peduncle, or one chamber basally with four chambers at the median region (*Larson & Fautin, 1989*). However, the number of chambers varies during staurozoan development (*Mayer, 1910*; *Uchida, 1929*; *Hirano, 1986*), as reported in *Haliclystus* and *Manania* (*Wietrzykowski, 1911*; *Wietrzykowski, 1912*; *Uchida, 1929*; *Hirano, 1986*), and this character must be cautiously employed in taxonomic studies (*Miranda et al., 2016*).

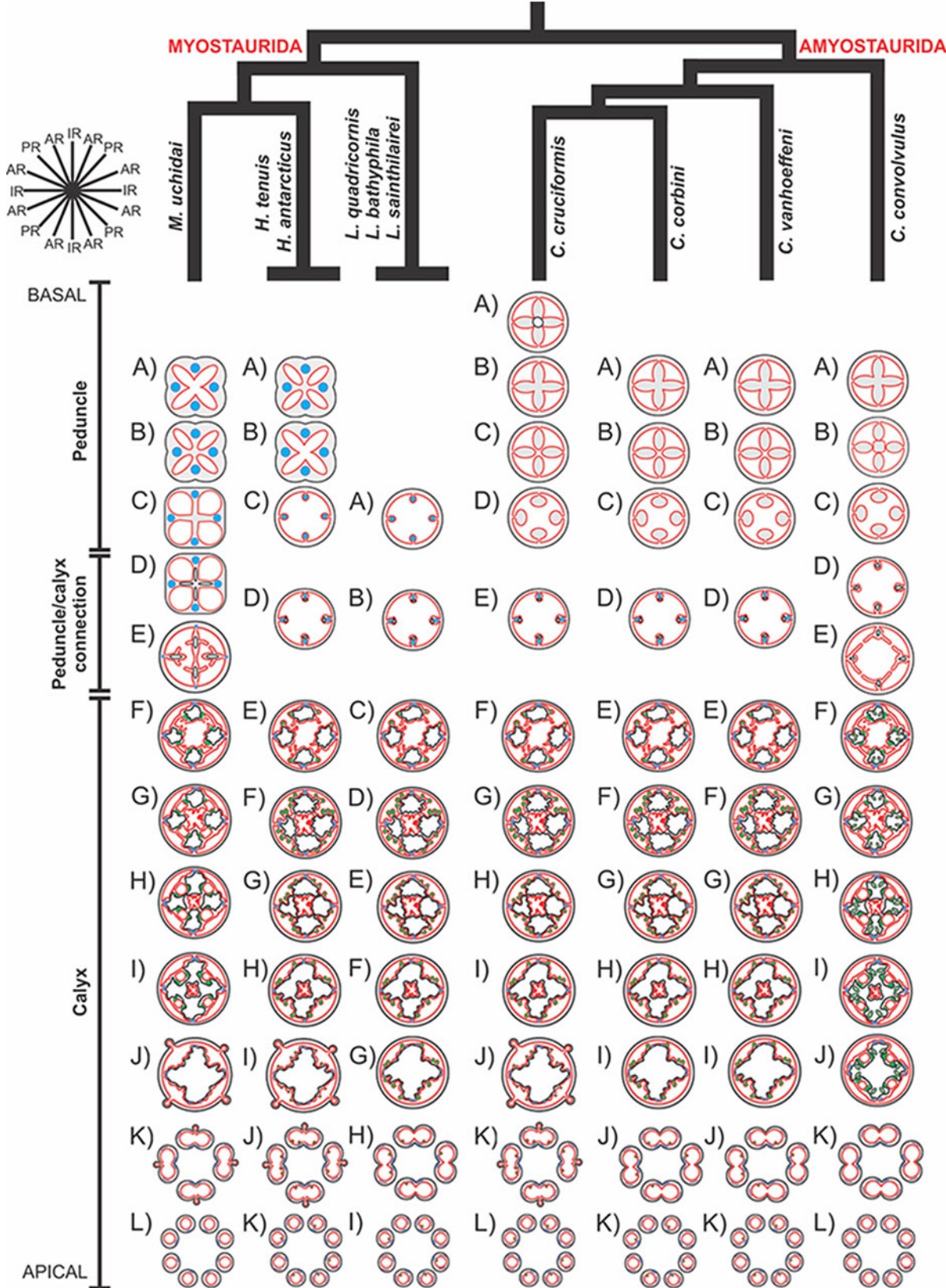

**Figure 57 General body plan of different species and their phylogenetic relationship (modified from *Miranda et al., 2016*).** *Manania uchidai*: (A) peduncle with one chamber and four interradial longitudinal muscles; (B, C) peduncle with four chambers; (D) four interradial infundibula; (E) delimitation of claustra; (F) claustra and four accessory radial pockets; (G, H) delimitation of principal radial pockets and manubrium; (I) accessory and principal radial pockets separated by claustra; (J) interradial ostia, and perradial anchors connected to the gastrovascular cavity; (K) interradial anchors connected to the gastrovascular cavity; L) total separation of arms; *Haliclystus tenuis* and *H. antarcticus* (modified from *Miranda, Collins & Marques, 2013*): (A) four perradial chambers and four interradial longitudinal muscles; (B) fusion of the four chambers into one chamber; (C) delimitation of interradial septa; (D) four interradial infundibula; (E) formation of gonads and gastric filaments; (F, G) delimitation of gastric radial pockets and manubrium; (H) four gastric radial pockets and a central manubrium; (I) interradial ostia, and perradial anchors connected to the gastrovascular cavity; (J) interradial anchors connected to the gastrovascular cavity; (K) total separation of arms; *Lucernaria quadricornis*, L. bathyphila, L. sainthilairei: (A) four gastric septa with internal interradial longitudinal muscles; (B) four interradial infundibula; (C) formation of gonads and gastric filaments; (D, E) delimitation of gastric radial pockets and manubrium; (F) four gastric radial pockets and a central manubrium; (G) interradial ostia; (H, I) separation of arms; *Calvadosia cruciformis*: (A) four perradial chambers and one central axial canal; (B) four perradial chambers; (C, D) delimitation of four interradial septa; (E) four interradial septa with four interradial longitudinal muscles and infundibula; (F) formation of gonads and gastric filaments; (G, H) delimitation of gastric radial pockets and manubrium; (I) four gastric radial pockets and a central manubrium; (J) interradial ostia, and perradial primary tentacles connected to the gastrovascular cavity; (K) interradial primary tentacles connected to the gastrovascular cavity; (L) total separation of arms; *Calvadosia corbini* and *Calvadosia vanhoeffeni*: (A) four perradial chambers; (B, C) delimitation of four interradial septa; (D) four interradial septa with four interradial longitudinal muscles and infundibula; (E) formation of gonads and gastric filaments; (F, G) delimitation of gastric radial pockets and manubrium; (H) four gastric radial pockets and a central manubrium; (I) interradial ostia; (J, K) separation of arms; *Craterolophus convolvulus*: (A) four perradial chambers; (B, C) delimitation of four interradial septa (central layer of mesoglea below complete delimitation); (D) four interradial septa with infundibula; (E) delimitation of claustra; (F) claustra and four accessory radial pockets; (G, H) delimitation of principal radial pockets, auxiliary radial pockets, and manubrium; (I) accessory and principal radial pockets separated by the claustra, and lateral auxiliary radial pockets (oval projections with gonads); (J) interradial ostia; (K, L) separation of arms. Legend: epidermis, black; gastrodermis, red; mesoglea, gray; longitudinal muscles, blue; coronal muscle, purple; gonads, green; AR, adradii; IR, interradii; PR, perradii.

According to the classification proposed by *Miranda et al. (2016*; Fig. 1*)* based mainly on molecular phylogenetic analyses, *Lucernariopsis*, *Sasakiella*, and *Kishinouyea* are synonyms of *Calvadosia*. At least in principle, species of former *Lucernariopsis* would have one chamber throughout the peduncle, whereas former *Kishinouyea* and *Sasakiella* would have four chambers at the base of the peduncle (sometimes separated by an axial canal resulting from an invagination of epidermis), which gradually merge apically to form a single chamber (*Kramp, 1961*). However, the presence of four chambers only at the base of the peduncle is not easy to observe, demanding serial transversal sections of a well-preserved and straight peduncle (the bodies of stauromedusae frequently contract during preservation). Indeed, cross sections at the bases of peduncles are rare in the descriptions of species, and most of them include only information about the median region of the peduncle (*Edmondson, 1930*; *Carlgren, 1938*; *Miranda et al., 2012*). Consequently, it is difficult to be sure that this distinction was correctly applied in former determinations. We have sectioned and analyzed the base of the peduncle of *Calvadosia vanhoeffeni*, formerly *Lucernariopsis vanhoeffeni* (Fig. 47), and although the species has been described with a one-chambered peduncle (*Browne, 1910*), it has four chambers at the peduncle base (Figs. 47D–47E). Therefore, the peduncle should be re-examined in detail for other species of the genus *Calvadosia*, in order to verify the universality and/or variability of number of chambers in the group.

A possible function for the chambers in the peduncle of stauromedusae has never been discussed, but they probably work as a hydrostatic skeleton (fluid filled cavities; see review in *Kier, 2012*), in addition to being an important part of the gastrovascular system, increasing surface area (*Hand, 1966*). In hydrostatic skeletons the force of muscle contraction is transmitted by internal pressure (*Kier, 2012*). The association of antagonist

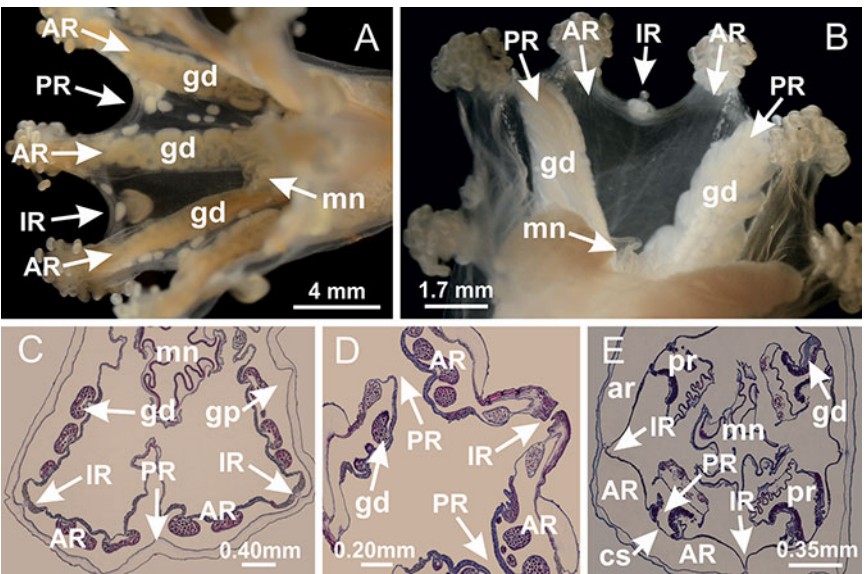

**Figure 58 Gonads and claustra.** *Haliclystus tenuis*: (A) general view of gonads associated with adradial arms in animals without claustrum; *Manania uchidai*: (B) general view of gonads at perradii in animals with claustra; *H. tenuis*: (C) cross section of calyx with gonads in the gastric radial pockets; (D) cross section of adradial arms with gonads; *M. uchidai*: (E) cross section of calyx with gonads in the principal radial pockets, at perradii. See Table 2 for abbreviations.

muscles (longitudinal and circular) with the hydrostatic skeleton can change the shape, size, and orientation of the body (*Kier, 2012*), as well as possibly improving temporary attachment—all actions particularly important to these animals.

The four chambers are formed during the metamorphosis of the stauropolyp to the stauromedusa, and their organization is complete only late in development (*Wietrzykowski, 1912*), so they should not be considered typical "polypoid" structures. For instance, the peduncle of developing *Haliclystus octoradiatus* has one chamber until the 32-tentacle stage, and then, progressively, four independent chambers are formed upwards (*Wietrzykowski, 1911*; *Wietrzykowski, 1912*). Similar observations were also described for other species of *Haliclystus*, whose juveniles have a single-chambered peduncle, later divided into four chambers that extend from the base to the top of the peduncle (*Hirano, 1986*). Accordingly, the general understanding that the peduncle region retains polypoid characters (*Stangl, Salvini-Plawen & Holstein, 2002*) is not completely correct for stauromedusae.

Therefore, the internal anatomy of the peduncle of many stauromedusae is different and more complex than the condition found in stauropolyps (*Wietrzykowski, 1912*) and in other medusozoan polyps, indicating that metamorphosis in Staurozoa (stauropolyp to stauromedusa) occurs both at the apical region (calyx) and basal region (peduncle) (*Uchida, 1929*; *Hirano, 1986*). A simple comparison between states of the stauromedusa peduncle and those of polyps of medusozoans is an oversimplification. Consequently, a detailed investigation of the metamorphosis of stauropolyps into stauromedusae would help evaluate hypotheses concerning the evolution of morphological characters in Staurozoa and in cnidarians in general.

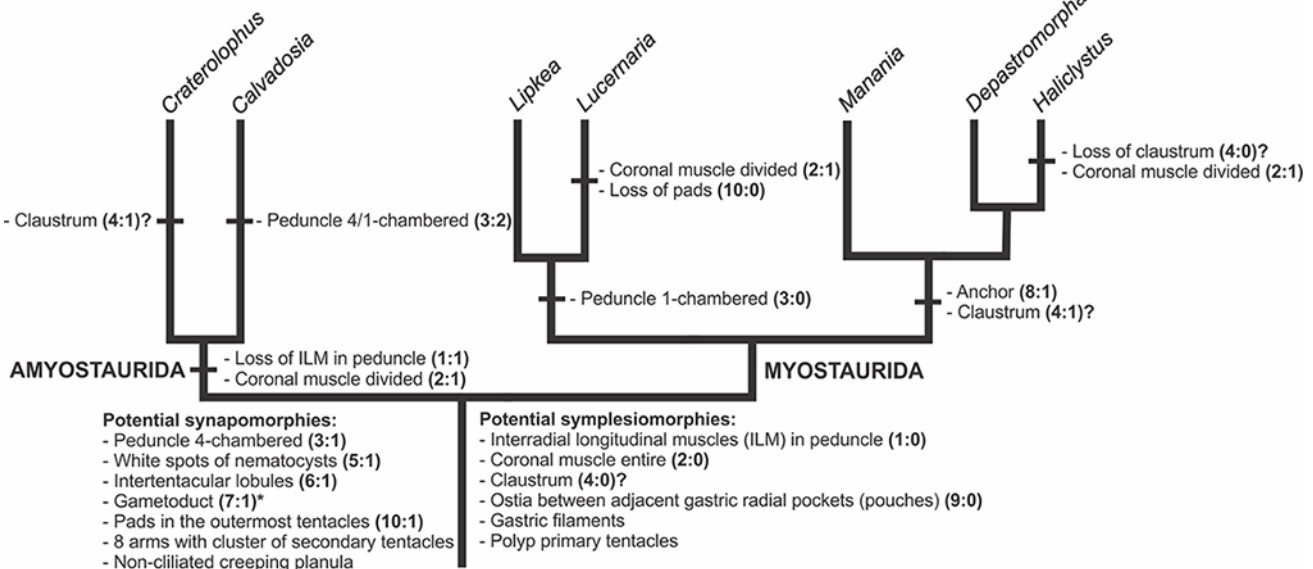

**Figure 59 Hypothesis of character evolution for staurozoan genera included in the recent molecular phylogenetic hypothesis (*Miranda et al., 2016*).** ACCTRAN optimization of morphological and life-history features (modified from *Miranda et al., 2016*). Characters 1–10 examined in this study (see Table 3 for detailed information). "?" Equivocal character (see Table 3). If claustrum is considered a symplesiomorphy of Staurozoa, it was lost in *Calvadosia*, *Haliclystus*, and in the clade *Lucernaria* + *Lipkea* (most parsimonious reconstruction). *Presence of gametoducts in other cnidarians requires further studies.

## Claustrum

The gastrovascular system of some stauromedusae has an additional level of complexity because of the presence of the claustrum (Fig. 2D; *Clark, 1863*; *Gross, 1900*; *Berrill, 1963*; *Collins & Daly, 2005*). The claustra are structures formed by the fusion of lateral evaginations of adjacent interradial septa, and each claustrum is composed of a central layer of mesoglea surrounded by gastrodermis (Figs. 2D, 20 and 53). We documented the claustra in *M. uchidai* (Fig. 20) and *C. convolvulus* (Fig. 53). Besides *Manania* and *Craterolophus*, claustra have also been recorded in *Depastrum*, *Depastromorpha*, and *Halimocyathus* (Table 3; *Clark, 1863*; *Kramp, 1961*; *Miranda et al., 2016*).

In *M. uchidai* and in *C. convolvulus*, the claustra delimit the accessory radial pockets (Figs. 20 and 53), and the principal radial pockets are formed in association with the manubrium (Figs. 20I and 20J). The accessory and principal radial pockets are separated by claustra (Figs. 2D, 20L, 20M, 53J and 54A). Consequently, there are eight gastric pockets in the calyx instead of four in the animals without the claustrum (Fig. 6; *Gross, 1900*; *Berrill, 1963*). The four external pockets, known as accessory pockets (or exogon; *Thiel, 1966*), extend into the anchors, arms, and secondary tentacles, continuing into the peduncle as perradial chambers (Fig. 57; *Berrill, 1963*). Although the accessory radial pockets occupy the same position as gastric radial pockets (of the animals without the claustrum), the four internal pockets, also known as principal (main) radial pockets (or mesogon, *Thiel, 1966*), are apparently the true gastric radial pockets of these stauromedusae because they contain the gonads and the gastric filaments (Figs. 20, 53, 54 and 57; *Clark, 1863*; *Gross, 1900*; *Berrill, 1963*).

Because of the internal organization, the principal radial pockets of the animals with claustra occupy a perradial position. Consequently, their associated gonads are also perradial (Figs. 57 and 58). The accessory radial pockets, which do not contain gonads, are associated with the arms that, likewise, do not contain gonads (Figs. 57 and 58). On the contrary, animals without the claustrum have gonads associated with the gastric radial pockets, therefore in an adradial position extending into the arms (Fig. 58). Animals with interradial paired arms (such as *C. cruciformis*, *C. corbini*, *L. quadricornis*) have the gonads still associated with the gastric radial pockets in the arms, but then positioned at the interradii/adradii.

Therefore, as a consequence of the internal architecture, the position of gonads is a useful external character for distinguishing animals with and without the claustrum (Fig. 58). Although the claustrum is a valuable character for distinguishing genera in Staurozoa, making histological sections to ascertain the presence or absence of this structure is not always possible. Externally, animals with claustra have exclusively perradial gonads located from the base of the manubrium to the perradial margin of calyx; whereas animals without the claustrum have perradial gonads only at the base of the manubrium, assuming an adradial position at calyx margin and arms, or they are associated with the interradial paired arms (Fig. 58). This can be observed in species of *Manania*, *Craterolophus*, and *Depastromorpha*, but needs to be checked for species of *Depastrum* and *Halimocyathus* (the other staurozoan genera with claustra).

According to *Berrill (1963*: 749), species with and without claustrum indicate "two biological engineering solutions to nutritional problems but does not imply a superiority of one type over the other." The claustra compartmentalize the body of stalked jellyfishes, and this could be a response to biotic or abiotic pressures, but does not seem to be directly related to other internal structures, such as the number of gastric chambers or the presence of interradial longitudinal muscles in peduncle. However, the claustra create an exclusive compartment for the gonads, separating them from the gastric chambers, anchors, arms, and secondary tentacles (differently from the animals without the claustrum), although basal communication exists before claustra delimitation is complete (Figs. 53A–53C). The gonads in *C. convolvulus* and *M. uchidai*, species with claustra, are similar, organized as a simple layer, between the gastrodermis and epidermis, and not as numerous vesicles (Figs. 22 and 54). In addition, they are constrained to the interior of the calyx, never associated with arms, and therefore they would be more protected from predators that graze part of a stauromedusa (*Prell, 1910*; *Uchida & Hanaoka, 1933*; *Davenport, 1998*; *Mills & Hirano, 2007*).

*Clark (1863)* was apparently the first author to describe the claustrum, and used the feature to divide the stalked jellyfishes into two main groups: Cleistocarpida (claustrum present) and Eleutherocarpida (claustrum absent). Since then, the claustrum has played a central role in the taxonomy of Stauromedusae (*Collins & Daly, 2005*), and most of the main proposals of classification have been based on that feature (see review in *Miranda et al., 2016*). Molecular phylogenies make it clear that neither Cleistocarpida nor Eleutherocarpida are monophyletic, dismissing the importance of the claustrum for diagnosing the primary subgroups of Staurozoa (*Collins & Daly, 2005*;
*Miranda et al., 2016*). A question derived from the homoplastic appearance of the claustrum is whether it is comprised of the same structures among the different and non-closely related staurozoan genera. We analyzed the distantly related *C. convolvulus* and *M. uchidai*, and the structures of claustra in these species are morphologically similar, i.e., lateral projections of the septa composed only of mesoglea and gastrodermis (Figs. 20 and 53). In *C. convolvulus*, besides the accessory and principal radial pockets, there are also numerous auxiliary radial pockets containing gonads, as a result of folds of septa and principal radial pockets (Figs. 53 and 54). *Collins & Daly (2005)* questioned whether a claustrum might be an ancestral character for Staurozoa (Fig. 59; Table 3). If so, the claustrum was lost several times in staurozoan evolution (Fig. 59; Table 3; *Miranda et al., 2016*). Indeed, the claustrum has also been described in the medusa stage of Cubozoa, but the gonads are associated with the exogon in box-jellyfishes (*Thiel, 1966*), and the existence of a typical staurozoan-type claustrum in Cubozoa needs to be assessed (Fig. 59; Table 3; *Miranda et al., 2016*).

## Internal layer of nematocysts, white spots of nematocysts, and exumbrellar/subumbrellar batteries of nematocysts

A continuous internal layer of nematocysts at different stages of development has been recently described for *H. antarcticus* in the subumbrellar epidermis of arms, radial pockets, and infundibula (*Miranda, Collins & Marques, 2013*). It has been hypothesized that these nematocysts are produced in the epidermis of the infundibula and migrate through the subumbrellar epidermis to the tips of the secondary tentacles in the knob, where mature nematocysts are organized in one external row and can perform their function of defense from predators, capture of prey, and adherence to substrate (Figs. 15–17; *Miranda, Collins & Marques, 2013*; see also *Gross, 1900*; *Kassianow, 1901*; *Wietrzykowski, 1912*; *Weill, 1925*; *Weill, 1935*; *Leuschel, 1932*; *Uchida & Hanaoka, 1933*; *Uchida & Hanaoka, 1934*). *Wietrzykowski (1912)* observed this layer in the epidermis of the infundibula, adjacent to the septa, beginning with the 4-tentacle stage of the stauropolyps of *H. octoradiatus*. According to him, most of the cells in these regions become nematoblasts and, as development proceeds, the epidermis contains nematocysts at different stages of development without direct contact with the external environment (*Wietrzykowski, 1912*).

We observed this internal layer of nematocysts from the epidermis at the bases of the infundibula to the knobs of the secondary tentacles (Figs. 15–17) in all species (with the possible exception of *M. uchidai*, in which this layer was not clearly discernible, but has been previously described by *Uchida & Hanaoka, 1933*: 145, identified as *M. distincta*), indicating its likely universality in Staurozoa. Additionally, this layer seems to be associated with the accumulation of unorganized nematocysts at the base of the secondary tentacles (Fig. 16L), internal to the epidermis and just opposite to the intertentacular lobules (*Miranda, Collins & Marques, 2013*). This region of accumulation was observed in all species analyzed in this study (Fig. 16), and could be important for providing the animal with a constant and dynamic supply of nematocysts.

Although *Weill (1925)* has suggested a similar scenario for the migration of nematocysts, he associated the internal layer of nematocysts with another set of structures, the white spots of nematocysts (Figs. 12 and 32). The white spots are located on the subumbrella at the perradii and/or interradii (depending on the species), frequently associated with the calyx margin, gonads, and arms. In a cross section, they look like "bubbles" in the mesoglea, between layers of subumbrellar gastrodermis and epidermis, containing abundant nematocysts (Fig. 12). The process responsible for the formation of these structures has been proposed as invagination (*Weill, 1925*) or delamination (*Leuschel, 1932*) of the subumbrellar epidermis (*Uchida & Hanaoka, 1933*). The presence of white spots of nematocysts is widespread in Staurozoa, occurring in all genera, but they were never reported as taxonomically important until *Corbin (1978)* and, more broadly, *Hirano (1997)* and *Kahn et al. (2010)* used their presence and position (perradial versus perradial and interradial) to diagnose and differentiate staurozoan species. As far as we know, white nematocyst spots are only known for Staurozoa, and their presence is a probable synapomorphy for the class (Fig. 59; Table 3).

Three hypothetical functions of the white spots can be raised. First, they could be structures for defense and attack, with the nematocysts able to escape outside the sacs by pressure through a narrow slit between ectodermal cells (*Kling, 1879*; *Kassianow, 1901*; *Uchida & Hanaoka, 1933*). Alternatively, the sacs could simply be a place for the production of nematocysts (*Uchida & Hanaoka, 1933*), an idea offered due to the inferred lack of an aperture in the white spots (*Leuschel, 1932*). The third hypothesis is that the white spots could be selective reservoirs of nematocysts produced at the base of the infundibula (*Weill, 1925*; *Weill, 1935*) and that after a period of maturation, they would migrate out to the secondary tentacles (*Weill, 1925*; *Weill, 1935*).

The white spots of the species studied herein all have the same morphology, in accordance with that described by *Uchida & Hanaoka (1933*: 146*)* for *M. uchidai* (as *M. distincta*), with a basal peripheral layer of nematoblasts adjacent to the gastrodermis, and a central group of loosely aggregated mature nematocysts (Fig. 12). The nematoblasts have opaque and darker capsules, whereas in the mature nematocysts, internal structures such as the shaft and tubule are visible. An epidermal thickening at the central region of white spots (Figs. 12 and 32) was also consistently observed.

Importantly, serial longitudinal sections of the white spots of nematocysts of *C. convolvulus* and *L. bathyphila* revealed a small opening in these sacs (Fig. 32), located in the region with the thickest epidermis, in which a well-delimited pore passing through the epidermis and adjacent mesoglea connects mature nematocysts with the outside (Fig. 32). This strengthens the hypothesis that the white spots are structures for defense and attack because there is indeed an opening through which the nematocysts could be released. Furthermore, the basal and peripheral layer of nematoblasts suggests that the nematocysts from the white spots are produced locally, contrary to the hypothesis that they would be related to maturation of nematocysts produced at the subumbrellar epidermis (*Weill, 1925*; *Weill, 1935*). Besides, there are species without white spots, such as *H. antarcticus* (cf. *Miranda, Morandini & Marques, 2009*), suggesting that the nematocysts produced in the internal subumbrellar epidermis would not need a "selective reservoir"

(*Weill, 1925*; *Weill, 1935*) in order to become mature and migrate to the secondary tentacles.

Although the function of the white spots is still not known with certainty, it has also been hypothesized that they protect the stauromedusan gonads from predators (*Kling, 1879*; *Kassianow, 1901*; *Weill, 1925*). Indeed, we observed that they are often associated with the gonads (Fig. 20M). In addition, white spots appear concomitantly with the gonads during the development of *Haliclystus octoradiatus* (32-tentacle stage; *Wietrzykowski, 1912*, Fig. XXI). The striking color pattern provided by the presence of these white spots could also be a visual signal to either attract prey or deter visual predators, as their bright reflecting white color diverges from the usually camouflaged body of stauromedusae (*Mills & Hirano, 2007*).

There is no information about whether the white spots are permanent or transitory structures, i.e., whether or not these structures are retained after (and if) nematocysts are released. However, apparently their perradial/interradial position is constant, making them useful in species differentiation (*Corbin, 1978*; *Hirano, 1997*; *Kahn et al., 2010*). During development of different species of Staurozoa, the perradial white spots generally appear first (e.g., *Haliclystus borealis*, *H. tenuis*), but in some species (e.g., *Haliclystus stejnegeri*, *H. octoradiatus*) eight adradial white spots appear at the same time (*Wietrzykowski, 1912*; *Uchida, 1929*; *Hirano, 1986*). This pattern of appearance of the white spots was used to support a close relationship between *H. tenuis* and *H. borealis* (*Hirano, 1986*), corroborated in the recent molecular phylogeny (*Miranda et al., 2016*).

In summary, based on our results and on information in the literature, there are at least two regions of nematocyst formation in stauromedusae: 1) in the epidermis of the infundibula, supplying nematocysts to the secondary tentacles (Figs. 15 and 16); and 2) at the peripheral region of white spots, supplying nematocysts exclusively to the white spots (Figs. 12 and 32). However, there is not enough information to suggest whether the nematocysts of the secondary tentacles are being formed exclusively in the basal epidermis of the infundibula or throughout the whole subumbrellar epidermis, i.e., throughout the internal layer of nematocysts.

Many species of Staurozoa also have exumbrellar batteries (warts) of nematocysts (Fig. 13), comparable to those of other cnidarians. These batteries are simple epidermal clusters of nematocysts that probably play a role in the defense of stalked jellyfishes. These batteries of nematocysts are also common on the subumbrella of *C. corbini*, and are associated with the nodular gonads, between the areas internally corresponding to two adjacent vesicles (Figs. 13L, 13M, 40H and 40I). These subumbrellar batteries could provide additional protection to the gonads, which are probably more susceptible in these animals because they are projected to the exterior (Fig. 38). The origin of the nematocysts in these batteries, as well as the origin of the nematocysts found elsewhere, such as the gastric filaments (*Miranda, Collins & Marques, 2013*), is uncertain.

## Intertentacular lobules and "U-shaped" space

The base of the tentacular cluster at the internal tip of the arms can have either several intertentacular lobules, or a deep "U-shaped" space (Table 3; *Hirano, 1997*).

The intertentacular lobules are internal projections between adjacent secondary tentacles composed of a central layer of mesoglea surrounded by a layer of gastrodermis (the gastrodermis belonging to adjacent secondary tentacles; see examples in Figs. 33C and 50C). In a cross section, it is possible to see a circular gastrodermis surrounded by mesoglea (Figs. 17E and 17M). However, some species, such as *H. tenuis*, do not have intertentacular lobules, but have a "U-shaped" space at the base of the tentacular cluster (Figs. 14 and 59; Table 3). In this region, there is a platform composed of mesoglea and gastrodermis, similar to a large lobule in a longitudinal section (Fig. 14K), which connects the exumbrellar side of the arms to the base of secondary tentacles.

These structures were rarely mentioned hitherto. Apparently, *Clark (1878)* was the first author to describe the intertentacular lobules, in the stauromedusa *Haliclystus auricula*. More than a century later, the internal anatomy of the base of secondary tentacles was used to differentiate species of *Haliclystus* (*Hirano, 1997*): *Haliclystus californiensis*, *H. tenuis*, and *H. borealis* do not have intertentacular lobules but have a "U-shaped" space, whereas *Haliclystus* "*sanjuanensis*" (*nomen nudum*), *H. auricula*, *H. octoradiatus*, and *H. stejnegeri* have intertentacular lobules (also present in *H. antarcticus*, *Miranda, Morandini & Marques, 2009*). The possible function of these structures and consequences of the different shapes of the internal base of the tentacular clusters have never been discussed, but they increase the gastrovascular area (see *Hand, 1966*) and also likely function as a point of anchorage and communication between secondary tentacles, possibly related to their movement/contraction (*Hand, 1966*; *Kier, 2012*).

Although these structures are relatively well known for *Haliclystus*, their general occurrence in Staurozoa has never been studied before. We observed intertentacular lobules of different sizes in *Craterolophus* (*C. convolvulus*), *Calvadosia* (*C. corbini*, *C. vanhoeffeni*, and *C. cruciformis*), *Lucernaria* (*L. bathyphila*, *L. quadricornis*, and *L. sainthilairei*), and *Manania* (*M. uchidai*). Therefore, intertentacular lobules seem to be widespread in Staurozoa and their presence is a probable synapomorphy for the class (Fig. 59; Table 3), whereas the "U-shaped" space may be restricted to some *Haliclystus* species (Table 3).

## Gonads and gametoduct

The gonads of Staurozoa have been considered an important character to understand the evolution of the group, and the complex ovaries with follicle cells (*Eckelbarger & Larson, 1993*) have been suggested as a potential synapomorphy for the class (*Marques & Collins, 2004*; *Collins & Daly, 2005*; *Collins et al., 2006*; *Van Iten et al., 2006*), although this was recently questioned (*Tiemann & Jarms, 2010*).

The gonads are organized inside the gastric radial pockets in species lacking claustra, from the base of the calyx to the tips of the arms (Fig. 57). In *C. corbini*, evaginations of the gastric radial pockets produce nodular gonads as external projections on the wide-open subumbrella (Figs. 38E–38I and 40). In animals with claustra (e.g., *M. uchidai*, *C. convolvulus*), the gonads are associated with the principal radial pockets (mesogon), at perradii, and do not extend into the arms. *Craterolophus convolvulus* is

unique in our observations in possessing auxiliary radial pockets, which also contain gonadal tissue (Figs. 53G–53J and 54).

Gonads are generally organized in vesicles as evaginations of gastrodermis, more (i.e., *H. tenuis*; Fig. 8) or less (i.e., species of *Lucernaria*; Figs. 27, 31 and 36) delimited. Gonadal content can also be restricted to a simple layer between gastrodermis and epidermis (i.e., *M. uchidai*; Fig. 22), without the organization of vesicles.

Male and female gonads can only be recognized with histological sections, and their internal structures have been described in a few different studies (*Uchida, 1929*; *Berrill, 1963*; *Eckelbarger & Larson, 1993*; *Eckelbarger, 1994*; *Miranda, Collins & Marques, 2013*). The gonadal content in males is organized in one peripheral layer of spermatocytes adjacent to gastrodermis and a central and internal layer of spermatozoa generally adjacent to epidermis. In females, a peripheral immature layer of oocytes is adjacent to gastrodermis and the mature oocytes occupy a central position, generally adjacent to epidermis. Each gonadal male and female structure contains numerous developing and mature spermatozoa and many immature and mature oocytes, respectively. Additionally, follicle-like accessory cells surround individual oocytes in *Haliclystus "sanjuanensis"* (*Eckelbarger & Larson, 1993*).

In this study, clear regionalization was recognized in all species other than species of *Lucernaria*. In *L. quadricornis*, the oocytes, surrounded by follicle cells, are randomly arranged in the mesoglea between gastrodermis and epidermis (Fig. 27). Oocyte organization in this genus is somewhat similar to the pattern of Scyphozoa and Cubozoa (*Eckelbarger, 1994*). The specimens of *L. bathyphila* and *L. sainthilairei* examined are apparently males, as they have structures similar to spermatocytes, but probably immature because spermatozoa could not be recognized (Figs. 31 and 36).

Gametoducts are a direct connection between the gonadal content and the gastrodermis of the vesicles (or correspondent gonadal structures), which likely carry spermatozoa or mature oocytes to the gastrovascular cavity. In females, they are associated with cells, probably the follicle-like cells of *Eckelbarger & Larson (1993)*, that surround and isolate mature oocytes at the central region of vesicles (or correspondent gonadal structures). The organization is similar in males; gametoducts are associated with cells surrounding the spermatozoa, sometimes producing different sacs, and directly connected with spermatocytes (Figs. 48 and 49) most likely of gastrodermal origin. Moreover, cilia (Figs. 27G–27I) were observed in the connection of gametoducts with the gastrodermis of vesicles (or correspondent gonadal structures), probably contributing to a more efficient transport of gametes. It is noteworthy that gametoducts were observed in all but one analyzed species. The only exception was *L. sainthilairei*, for which no gametes were observed in the specimen, raising questions about whether the permanence or transitory presence of gametoducts is related to gonadal maturity.

The presence of gametoducts in stalked jellyfishes has been described for *H. auricula* as "mouth of genital sac," "entrance of the sac," "aperture of genital sac" (*Clark, 1878*: 67, Figs. 54, 62, 74, 75 and 77), for *L. bathyphila* as "oviduct" or "oviductulus" (*Haeckel, 1882*: 61, Plate XVII, Figs. 17 and 19), and for *H. octoradiatus* as "*étroit canal*" (*Wietrzykowski, 1912*: 77, Fig. 30), but has been completely ignored for over 100 years. This is the first

study to indicate the universality of gametoducts in Staurozoa, present in all analyzed genera, both in males and females, and even in species without delimited vesicles (viz., *M. uchidai*, *C. convolvulus*), and their presence is reconstructed as a putative synapomorphy for the class (Fig. 59; Table 3). Nevertheless, more detailed studies in other cnidarian classes are necessary to better understand the evolution of these structures within Cnidaria (see also discussion below about gonopores in *Periphylla periphylla*, *Tiemann & Jarms, 2010*).

*Berrill (1963*: 742, 745*)* proposed that the "expulsion of gametes" in *Haliclystus salpinx* and in *L. quadricornis* "results from the rupture of the endodermis enclosing the gonads." Our histological studies, however, suggest that gametes are released through specialized structures, indicating a complex functional mechanism. Gametoducts allow the selective release of mature gametes, preventing damage or destruction of the gonads, and not wasting immature oocytes (peripheral) in female vesicles and spermatocytes in male vesicles (or correspondent gonadal structures).

A similar structure, the gamete-releasing pore, was described for the coronate scyphomedusa *Periphylla periphylla* (*Tiemann & Jarms, 2010*). These special pores would work as gonopores, therefore the structure of the gonad would not be destroyed, allowing a continuous gamete release (*Tiemann & Jarms, 2010*). Spermatozoa were seen associated with these pores, as well as oocytes, which were expelled through this narrow pore, becoming deformed, but then regaining their spherical shape (*Tiemann & Jarms, 2010*). In addition, there is a strong maturity gradient in gonads of both sexes, simultaneously presenting all developmental stages of gametes (*Tiemann & Jarms, 2010*), a regionalization also observed in Staurozoa. *Tiemann & Jarms (2010)* concluded that complex and specialized structures, including gonopores, strongly suggest that true sexual organs exist in Scyphozoa (interpreted broadly to include Cubozoa and Staurozoa), even though they presented observations on only a single species of coronate scyphozoan. The homology between gametoducts in Staurozoa and gonopores in *Periphylla periphylla* remains to be tested. Indeed, recent research is beginning to uncover more complex reproductive systems in different cnidarians (*Cartwright & Nawrocki, 2010*; *Schiariti et al., 2012*), especially some species belonging to the class Cubozoa (*Garm, Lebouvier & Tolunay, 2015*; *Marques, García & Lewis Ames, 2015*). Although plesiomorphic external fertilization would be expected in adult medusae (*Marques & Collins, 2004*), certain cubomedusae can present courtship behavior, specialized internal fertilization, and sperm storage (*Lewis & Long, 2005*; *Garm, Lebouvier & Tolunay, 2015*; *Marques, García & Lewis Ames, 2015*), which demonstrate the importance to highlight different cnidarian reproductive strategies in order to study their evolution.

However, a recent review concluded that Cnidaria has no specialization in the reproductive system, such as genital ducts, generally used for storing, transporting, or extruding gametic products (*Extavour, 2007*). Our results and data in *Tiemann & Jarms (2010)* refute this interpretation. That said, gonoducts in Bilateria generally take gonadal elements directly to the outside of the animal (*Dewel, 2000*; *Extavour, 2007*), in contrast to the structures of Staurozoa and *P. periphylla* (*Tiemann & Jarms, 2010*). Staurozoan gametoducts release gametes into the gastrovascular cavity, after which the

gametes are then expelled to the outside through the mouth (*Otto, 1976*). In addition, staurozoan gametoduct structures are perhaps unlikely to be homologous with bilaterian gonoducts because of their different origins, the latter generally mesodermal (*Extavour, 2007*; *Nielsen, 2008*).

Convergent evolution may have produced different morphological solutions to the problems posed by sexual reproduction (*Extavour, 2007*), although it is possible that similarities in the underlying molecular pathways may exist. Consequently, further investigation of the occurrence of gonopores or gametoducts in other cnidarians, as well as additional developmental data, particularly on gene expression, could uncover possible homologies of these specialized structures. Such knowledge will likely have important implications for understanding the evolution of reproductive systems across Eumetazoa.

## Anchors and primary tentacles

During the early metamorphosis of a stauropolyp, eight primary tentacles develop, four perradial and four interradial (*Wietrzykowski, 1912*; *Hirano, 1986*; *Kikinger & Salvini-Plawen, 1995*), which may have four different fates in the stauromedusa stage: 1) they disappear by resorption (in *L. quadricornis*, *Berrill, 1963*); 2) they metamorphose into adhesive structures called anchors (in species of *Haliclystus*, *Hirano, 1986*); 3) they remain as primary tentacles, although with a modified shape (in *Calvadosia tsingtaoensis* and *C. cruciformis*; *Ling, 1937*); 4) they are originally rather filiform, but transform into capitate tentacles and cluster together with secondary tentacles (in species of *Stylocoronella*, *Kikinger & Salvini-Plawen, 1995*).

Anchors are apparently restricted to Haliclystidae (Fig. 59; Table 3; *Miranda et al., 2016*), and their shapes have been broadly used to distinguish species of *Haliclystus* (*Miranda, Morandini & Marques, 2009*; *Kahn et al., 2010*). However, the use of this character for differentiating species can cause taxonomic problems because of ontogenetic variation (*Miranda, Morandini & Marques, 2009*).

Primary tentacles and anchors are hollow structures, composed of an internal layer of gastrodermis and an external layer of epidermis, with mesoglea between them (Figs. 9 and 10). Perradial primary tentacles/anchors are directly connected to the gastrodermis of the gastric radial pockets (or accessory radial pockets in animals with claustra) (Fig. 9). Interradial primary tentacles/anchors are connected to the gastrovascular cavity by means of small interradial ostia (Fig. 10) that connect adjacent gastric radial pockets (or accessory radial pockets in the animals with claustra).

We agree that anchors (or rhopalioids) and rhopalia, present in Staurozoa, Cubozoa, and Scyphozoa, are probably homologous marginal structures because they all result from metamorphosis of primary tentacles (*Uchida, 1929*; *Thiel, 1966*). However, whereas the rhopalia in Cubozoa and Scyphozoa have a sensorial function (*Nakanishi et al., 2010*), the rhopalioids in Staurozoa likely have a role in substrate attachment (*Larson, 1988*). The epidermis of the anchor is formed by tall supporting cells and glandular cells, providing evidence for an adhesive function that contributes to small movements of the animals, mainly when the peduncle is detached from the substrate (*Miranda, Collins & Marques, 2013*). Nematocysts are generally not observed in the anchors, but they are visible in

species with a knobbed remnant of primary tentacles, such as in *M. uchidai*, corroborating their tentacular nature. The primary tentacles of *C. cruciformis* also have an external layer of nematocysts (Fig. 9L), and they probably also help in the locomotion (adhesive function) of the stauromedusa.

## Ostia

The presence of an ostium between two adjacent gastric radial pockets has been recorded before for a small number of staurozoan species, such as *H. salpinx*, *Manania atlantica* (*Berrill, 1963*), and *H. antarcticus* (*Miranda, Collins & Marques, 2013*). The ostium connects the gastrodermis of the interradial anchor with that of the calyx (Fig. 10). In species with claustra this connection occurs in the accessory radial pockets. The main hypothesis to explain the presence of these ostia is that they might permit a nutritional extension into the four interradial anchors (*Berrill, 1963*: 742). However, they are small that they might not allow for effective circulation of fluids between two adjacent radial pockets (*Berrill, 1963*). Based on this hypothesis, species without anchors would not need an ostium connecting adjacent gastric radial pockets if ostia really only serve to allow for the connection of the anchor/primary tentacle to the calyx (*Miranda, Collins & Marques, 2013*).

However, all species analyzed, both with and without anchors/primary tentacles, have an ostium connecting two adjacent gastric radial pockets (Fig. 10). Therefore, two main hypotheses emerge: (1) the ostia, that probably provided gastrodermal extension to the primary interradial tentacles in the stauropolyp stage, were simply retained after the disappearance of primary tentacles in the stauromedusa stage (species without anchors/primary tentacles); (2) ostia are important in the communication of radial pockets, and probably have other function(s) besides providing nutritional extension to the four interradial anchors/primary tentacles.

Comparable structures, such as the septal ostium or connecting canal, have been observed in Scyphozoa (*Korschelt & Heider, 1895*; *Chapman, 1966*) and Cubozoa (*Conant, 1898*: 18, 19), but further detailed studies are necessary to assess their homology across these taxa (see also Fig. 59; Table 3).

## Pad-like adhesive structures

Pad-like structures can be present individually in the outermost secondary tentacles of the tentacular cluster (*Larson & Fautin, 1989*), or as a broad structure on the tip of each arm (Fig. 24; *Larson, 1980*; *Miranda et al., 2012*). We found individual pad-like adhesive structures in the outermost secondary tentacles in *C. convolvulus*, *M. uchidai*, *C. cruciformis*, and *C. vanhoeffeni*. These structures are a simple lateral projection in the tentacular stem, with an epidermal thickening (Figs. 24L–24N). The only examined species with pads at the tips of arms was *C. corbini* (Figs. 24A–24K). This more complex structure is an exumbrellar lateral projection with thick epidermis, mesoglea, and internal hollow gastrodermal canals, connected to the gastrovascular cavity of arms.

Individual adhesive glandular pads at the outermost secondary tentacles are widespread in the class (*Miranda et al., 2016*), a putative synapomorphy for the class

(Fig. 59; Table 3), whereas the broad pad-like adhesive structure on the tip of each arms is probably a synapomorphy of a clade within *Calvadosia* (*Miranda et al., 2016*, Fig. 16). The pad-like adhesive structures of the outermost secondary tentacles and at the tips of the arms were hypothesized to be homologous by *Corbin (1978)*, and the broad adhesive pad-like structure on the tip of each arm was suggested to be the result of fusion of several outermost secondary tentacles (*Larson, 1980*). Indeed, cross sections of the pads on the tips of arms show many hollow gastrodermal canals, which could be interpreted as the "merging" of several stem canals of the secondary tentacles. However, the thin gastrodermis of the pads in the arms contrasts with the thick and glandular gastrodermis of the secondary tentacles (Figs. 17, 24A–24K), but this could be a secondary differentiation related to a different function assumed by the pads.

Adhesive pads help the animals to adhere to its substrate (*Larson, 1980*). They have been hypothesized to serve to temporarily reattach the stauromedusae when the peduncle is detached (*Larson, 1988*). Their internal anatomy, with epidermis composed of tall glandular and supporting cells, like all other adhesive regions (viz. secondary tentacles, anchors, peduncle and pedal disk) of a stauromedusa, corroborates this hypothesis (*Miranda, Collins & Marques, 2013*).

## CONCLUSIONS

We have presented the internal anatomy of nine species of Staurozoa. Histological and taxonomic characters, such as the longitudinal interradial muscles, claustra, and chambers in peduncle were reviewed based on the recently proposed phylogenetic hypothesis for the class (*Miranda et al., 2016*). Characters rarely cited in the literature were also assessed. We found a general occurrence of intertentacular lobules in Staurozoa (putative synapomorphy for the class), contrary to the restricted occurrence of "U-shaped" space in some species of *Haliclystus*. Ostia between radial pockets are present in all observed species, even in those without interradial anchors/primary tentacles. Additionally, the functional and evolutionary aspects of key internal structures were discussed, such as the complex gonads in Staurozoa, and possibly in other cnidarians, with regionalization of mature and immature gametes and the presence of gametoducts (putative synapomorphy for the class, although more information for other cnidarian classes is necessary), both in males and females, thereby elucidating gamete release. Two different nematogenesis regions were observed in most species examined: 1) in the subumbrellar epidermis of infundibula, which supplies nematocysts to secondary tentacles, and 2) at the peripheral region of white spots, supplying nematocysts exclusively to the white spots. The new evidence that white spots have an opening through which the nematocysts can be released to the environment is consistent with the hypothesis that the white nematocyst spots (a putative synapomorphy of Staurozoa) are structures for defense of the gonads and attack, warranting more direct studies of their function. Future histological studies should include genera not addressed in this study (*Depastromorpha*, *Depastrum*, *Halimocyathus*, *Kyopoda*, *Lipkea*, and *Stylocoronella*) in order to access important information on the universality of many characters discussed here, such as the anatomy of the claustrum.

The challenge to homologize structures of Staurozoa and other cnidarian classes is due to the fact that a single stauromedusa has characters found in both medusae and polyps of other medusozoans. Therefore, comprehensive histological surveys associated with robust phylogenetic frameworks are important for providing a better understanding of the evolution of Cnidaria. In this context, comparative analyses including different life history stages are also essential to address the morphological evolution within the phylum.

## ACKNOWLEDGEMENTS

The authors are grateful to André C. Morandini (IB-USP) for discussions on the morphology and anatomy of Medusozoa and to Enio Mattos (IB-USP), José Eduardo A. R. Marian (IB-USP), and Amanda Ferreira e Cunha (IB-USP) for their kind assistance with the histological procedures and analyses. We are also grateful to James Reimer, Marymegan Daly, and an anonymous reviewer who helped to improve the quality of the manuscript.

### Funding

This study was supported by: FAPESP 2010/07362-7 (LSM), 2015/23695-0 (LSM), 2010/52324-6 (ACM), 2011/50242-5 (ACM), 2013/50484-4 (ACM); CNPq 142270/2010-5 (LSM), 165066/2014-8 (LSM), 474672/2007-7 (ACM), 563106/2010-7 (ACM), 562143/2010-6 (ACM), 477156/2011-8 (ACM), 305805/2013-4 (ACM), 445444/2014-2 (ACM); CAPES/PDSE: 16499/12-3 (LSM). The funders had no role in study design, data collection and analysis, decision to publish, or preparation of the manuscript.

### Grant Disclosures

The following grant information was disclosed by the authors:
FAPESP: 2010/07362-7 (LSM), 2015/23695-0 (LSM), 2010/52324-6 (ACM), 2011/50242-5 (ACM), 2013/50484-4 (ACM).
CNPq: 142270/2010-5 (LSM), 165066/2014-8 (LSM), 474672/2007-7 (ACM), 563106/2010-7 (ACM), 562143/2010-6 (ACM), 477156/2011-8 (ACM), 305805/2013-4 (ACM), 445444/2014-2 (ACM).
CAPES/PDSE: 16499/12-3 (LSM).

### Competing Interests

The authors declare that they have no competing interests.

### Author Contributions

- Lucília S. Miranda conceived and designed the experiments, performed the experiments, analyzed the data, contributed reagents/materials/analysis tools, wrote the paper, prepared figures and/or tables, reviewed drafts of the paper.
- Allen G. Collins analyzed the data, contributed reagents/materials/analysis tools, wrote the paper, reviewed drafts of the paper.
- Yayoi M. Hirano wrote the paper, reviewed drafts of the paper.
- Claudia E. Mills wrote the paper, reviewed drafts of the paper.

- Antonio C. Marques conceived and designed the experiments, analyzed the data, contributed reagents/materials/analysis tools, wrote the paper, reviewed drafts of the paper.

### Data Deposition

The raw data is included in the manuscript.

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
