# Peer review of "Comparative internal anatomy of Staurozoa (Cnidaria), with functional and evolutionary inferences"

_PeerJ, doi:10.7717/peerj.2594_

## Round 0.1 · original submission · Major Revisions

I have heard back from two reviewers and also read through the MS for myself. Both reviewers (and myself) are impressed with the amount of work that has gone into this manuscript - it is amazing in its detail and scope. That said, there are a few things that the reviewers suggest that I agree with that will help make this manuscript more accessible and informative for readers:

1. Reviewer 1 wishes to see a table comparing histological and taxonomic characters, which is similar to what reviewer 2 suggested (matrix of characters). I leave the form up to you, but I agree something along these lines would be helpful to readers navigating your work.

2. Reviewer 2 correctly mentions many of the characters' evolution is discussed but not tested in any way. Please consider how you can do this, or if not, change your MS accordingly. Personally, I hope you take up this challenge - it could add much to your paper.

I have also prepared a PDF with a few small comments of my own. Note: I cannot seem to attach the file as it is too big (!!), will figure out another way to get to you.

I hovered between major and minor revision, but given as this is a 164-page tome, I chose major revisions. I look forward to seeing a revised version.

Reviewer 1 ·

Basic reporting

No comments

Experimental design

No comments

Validity of the findings

No comments

Additional comments

General comments

The paper by Miranda et al. entitled “Comparative internal anatomy of Staurozoa (Cnidaria), with functional and evolutionary inferences.” is a great work on functional and evolutionary aspects of body plan of a broad range of staurozoan species from the view of the Miranda’s paper about Staurozoa (Miranda et al, 2016). The authors investigated the body plans of nine species of Staurozoa and compared these internal characters. They found many histological identification keys for taxonomy and ecological functions and evolutionary aspects of internal structures. They cleared that internal and external morphological study is very important for evolutional and functional biology with molecular phylogenetics. This paper will be of interest to jellyfish taxonomy and phylogenetic researchers.
However, this paper is too much pages of Figures…. Readers may confuse to find figures. I hope you must select important figures for your statements and then the other figures should be as supplemental figures.

Specific comments

L677~681
To state the paragraph, you need some supportive evidence or paper that reported previously.


L752~754
Delete the sentence.

L844~846
Why did you think so?
Did juvenile individuals have white spot?
Or
Only adult individuals have one?

L1093~1096
The function of white spot would be reasonable, however, you must observe this phenomena to state that function.

Finally, I would like you to add a Table which compare histological and taxonomic characters, longitudal muscles, coronal muscles,chambers in the pedancle, claustrum, nematocysts, Intertentacular lobules, U-shaped space, Gonad, gametoduct, anchors, primary tentacle, Ostia, and pad-lile adhessive structures.

·

Basic reporting

The manuscript is a bit hard to wade through (as is any anatomical treatise). Navigating the manuscript would be easier for me (and more useful for others) if the authors included an explicit matrix of characters--this would clearly and unambiguously connect the characters and states and make explicit (and testable) homology propositions. The absence of such an explicit statement of homology among elements of the anatomy makes evaluating and interpreting the text difficult and is problematic in light of the optimization that was performed as part of the discussion (but that is not described in the methods).

The manuscript engages in significant discussion of the evolution of anatomical attributes but lacks any explicit method for coding the anatomical features described, optimizing them on a tree, or evaluating alternative interpretations of either homology OR character transformation. This is a major limitation of the work, which is otherwise very strong and significant in both the level of detail and breadth.

Why is the species name of Haliclystus “sanjuanensis” is quotes?

Hand (1966) discusses the role of longitudinal musculature in other lineages of Cnidaria. This may be insightful for the authors' discussion of the musculature and chambers of the peduncle.

The discussion (to me) veers between descriptions of anatomy that lack synthesis (summarize the results) and assertions of function or evolutionary history that seem insufficiently justified (see points above re: explicit methods for optimization). This may reflect the state of knowledge or the nature of the evidence, and so may be justified, but the criteria by which conclusions are drawn or by which alternative interpretations are assessed is not discussed, which makes the differences more stark and harder to understand.

Authors' declaration (line 834 and again in the conclusions) notwithstanding, the discovery of an aperture does not corroborate the hypothesis that white spots serve a defensive/aggressive function.

The discussion of reproductive structures is excellent. The structures and functions described seem to give lie to the characterization of cnidarians as lacking organs. The impact of this is discussed with respect to metazoan systems, but not within Cnidaria.The impact of this finding is (to me) under-sold.

In general, comparisons with the complex polyps of anthozoans are lacking (and understandable, to some extent, given phylogeny), but that group has a rich literature of anatomical interpretation and functional study for longitudinal muscles, septae, etc, that may prove useful for understanding the anatomies of staurozoa.For example, regionization of games (with more mature games in closer proximity to the GVC) is common among Anthozoa, as are ostia (in the mesenteries).

The Conclusions section is a summary, not a conclusion. Unless required by the journal, I would cut that section (nothing new in it, duplicative of abstract, increases length)

Experimental design

I think a more explicit discussion of the ways in which alternative hypotheses are considered and evaluated would strengthen the manuscript. I think that coding and homologizing the characters across the taxa would be useful in this and would make transformation series more transparent. The summary of anatomy and a broad consideration of these in a phylogenetic context is interesting, but it is not as rigorous as it could be.

This point is the basis for my characterization of the manuscript as needing major revisions.

Validity of the findings

The quantity of the data collected and the detail that is provided are exceptional. The findings are basically descriptions of the anatomy of these animals, and as such, they are both well-supported and well documented. The figures are gorgeous!

---

## Round 0.2 · accepted · Accept

Both reviewers are very happy with the revised version. You should be commended for your hard work!

Reviewer 1 ·

Basic reporting

No Comments

Experimental design

No Comments

Validity of the findings

No Comments

Additional comments

This manuscript is improved. Especially, added Table 3 and Figure 59 are greatly helpful to reading and understanding your works. This paper is also great work to bear out your molecular phylogenetic work with internal morphology works in detail. Then, almost all question and comments from reviewers were cleared.
Therefore, this paper must be published for jellyfish researches ASAP.

It is very happy to read and referee this manuscript.
Thank you.

·

Basic reporting

The edits have improved the manuscript and it meets (or exceeds) all basic reporting standards.

Experimental design

Meets standards.

Validity of the findings

New table and statement on methods provides better context for conclusions. No further changes required.

Additional comments

This is an outstanding piece of work.